# Predictive nonlinear modeling of malignant myelopoiesis and tyrosine kinase inhibitor therapy

Jonathan Rodriguez[1,2†], Abdon Iniguez[1,2†], Nilamani Jena[3], Prasanthi Tata[3], Zhong-Ying Liu[3], Arthur D Lander[2,4,5,6], John Lowengrub[2,5,6,7]*, Richard A Van Etten[2,3,5]*

[1]Graduate Program in Mathematical, Computational and Systems Biology, University of California, Irvine, Irvine, United States; [2]Center for Complex Biological Systems, University of California, Irvine, Irvine, United States; [3]Department of Medicine, University of California, Irvine, Irvine, United States; [4]Department of Developmental and Cell Biology, University of California, Irvine, Irvine, United States; [5]Chao Family Comprehensive Cancer Center, University of California, Irvine, Irvine, United States; [6]Department of Biomedical Engineering, University of California, Irvine, Irvine, United States; [7]Department of Mathematics, University of California, Irvine, Irvine, United States

*For correspondence:
lowengrb@math.uci.edu (JL);
vanetten@hs.uci.edu (RAVE)

†These authors contributed equally to this work

Competing interest: The authors declare that no competing interests exist.

**Abstract** Chronic myeloid leukemia (CML) is a blood cancer characterized by dysregulated production of maturing myeloid cells driven by the product of the Philadelphia chromosome, the BCR-ABL1 tyrosine kinase. Tyrosine kinase inhibitors (TKIs) have proved effective in treating CML, but there is still a cohort of patients who do not respond to TKI therapy even in the absence of mutations in the BCR-ABL1 kinase domain that mediate drug resistance. To discover novel strategies to improve TKI therapy in CML, we developed a nonlinear mathematical model of CML hematopoiesis that incorporates feedback control and lineage branching. Cell–cell interactions were constrained using an automated model selection method together with previous observations and new in vivo data from a chimeric *BCR-ABL1* transgenic mouse model of CML. The resulting quantitative model captures the dynamics of normal and CML cells at various stages of the disease and exhibits variable responses to TKI treatment, consistent with those of CML patients. The model predicts that an increase in the proportion of CML stem cells in the bone marrow would decrease the tendency of the disease to respond to TKI therapy, in concordance with clinical data and confirmed experimentally in mice. The model further suggests that, under our assumed similarities between normal and leukemic cells, a key predictor of refractory response to TKI treatment is an increased maximum probability of self-renewal of normal hematopoietic stem cells. We use these insights to develop a clinical prognostic criterion to predict the efficacy of TKI treatment and design strategies to improve treatment response. The model predicts that stimulating the differentiation of leukemic stem cells while applying TKI therapy can significantly improve treatment outcomes.

## Editor's evaluation

This is an important study that investigates the impact of tyrosine kinase inhibitors (TKIs) in chronic myeloid leukemia. Through a combination of preclinical in vivo measurements, clinical data, and computational modeling, the authors present solid evidence regarding the heterogeneous effects of TKIs in patients and how the response to treatment may be improved. This study is of interest to those working in the fields of mathematical oncology and cancer biology.

## Introduction

Chronic myeloid leukemia (CML) is a myeloproliferative neoplasm of the hematopoietic system, which normally produces billions of mature myeloid and erythroid cells on a daily basis, is tightly regulated, and accommodates massive increases in the production of individual cell types in response to physiological and pathological stresses. The hematopoietic system is organized hierarchically as a collection of progressively more differentiated cells starting from a hematopoietic stem cell (HSC) located in the bone marrow (BM) and ending with postmitotic terminally differentiated myeloid and lymphoid cells (*Rieger and Schroeder, 2012*; *Liggett and Sankaran, 2020*).

CML is characterized by an overproduction of myeloid cells including mature granulocytes (neutrophils, basophils, and eosinophils) and their immediate precursors (metamyelocytes, myelocytes, and promyelocytes), and of myeloid progenitors (*Jamieson et al., 2004*) including multipotential progenitors (MPPs) and committed progenitors (common myeloid progenitors [CMP], granulocyte-macrophage progenitors [GMPs], and megakaryocyte-erythroid progenitors [MEPs]). Untreated, the disease has three distinct phases (*Chereda and Melo, 2015*). In the initial 'chronic' phase, the differentiation of myeloid progenitors is essentially normal, resulting in excessive levels of mature postmitotic neutrophils and their immediate precursors. In later stages of the disease (accelerated phase and blast crisis), differentiation is reduced and expansion of immature progenitors is observed. Additional clonal karyotypic abnormalities are typically only observed during the accelerated and blast crisis phases (*Hehlmann et al., 2020*).

CML has one of the simplest cancer genomes. It is driven by a single genetic abnormality arising somatically in an HSC, the Philadelphia (Ph) chromosome, the result of a balanced translocation between chromosomes 9 and 22 that creates a fusion of the genes for *BCR* and *ABL1*. The product of the *BCR-ABL1* fusion gene is a dysregulated cytoplasmic protein-tyrosine kinase, BCR-ABL1. CML thus represents a natural model of dysregulated granulocytopoiesis (*Quintás-Cardama and Cortes, 2009*).

Cell biological studies have shown that Ph$^+$ cells expressing markers of normal HSC are capable of engrafting immunodeficient mice (*Sirard et al., 1996*; *Lewis et al., 1998*), implying that these cells are leukemia-initiating or leukemic 'stem' cells (LSCs). More mature committed progenitors in CML, like normal progenitors, lack sustained self-renewal capacity and cannot stably engraft immunodeficient mice nor generate hematopoietic colonies in vitro upon serial replating (*Huntly et al., 2004*). The proportion of LSCs in the BM is highly variable across CML patients at diagnosis and can range from a few percent to nearly 100% (*Petzer et al., 1996*; *Diaz-Blanco et al., 2007*; *Abe et al., 2008*; *Thielen et al., 2016*), perhaps reflecting different periods of time patients spend in chronic phase before they are diagnosed, different rates of disease progression, or both.

There is persuasive experimental evidence of significant feedback regulation of different cell compartments in the dynamics of myeloid cell production in both normal and CML hematopoiesis, including signaling between the normal and CML cells (*Jiang et al., 1999*; *Devireddy et al., 2005*; *Vicente-Dueñas et al., 2009*; *Naka et al., 2010*; *Reynaud et al., 2011*; *Zhang et al., 2012*; *Krause et al., 2013*; *Walenda et al., 2014*; *Welner et al., 2015*). For instance, experiments in a mouse model of CML provided evidence that IL-6, produced by leukemic neutrophils, blocked MPP differentiation toward a lymphoid fate, implying feedback from the myeloid lineage onto MPPs (*Reynaud et al., 2011*). Surprisingly, our knowledge of the details of feedback regulation in hematopoiesis is still incomplete, especially for granulopoiesis, where even late-stage feedback interactions are poorly understood. For example, two cytokines, granulocyte colony-stimulating factor (G-CSF) and granulocyte-macrophage colony-stimulating factor (GM-CSF), can pharmacologically increase neutrophil production, but mice lacking both cytokines maintain baseline neutrophil levels and can still increase neutrophil production in response to infection (*Basu et al., 2000*). In many cases, it is not known which cell types are providing and receiving the feedback, what signals are used, and what aspects of proliferative cell behavior they influence (i.e., proliferation rates, renewal probability, or progeny fate choice).

In spite of this knowledge deficit, CML can be treated quite effectively using selective small-molecule tyrosine kinase inhibitors (TKIs) of the *BCR-ABL1* kinase. TKIs such as imatinib, dasatinib, and nilotinib, which inhibit proliferation and increase apoptosis of Ph$^+$ cells, have dramatically lowered CML death rates (*Gambacorti-Passerini et al., 2011*). The response to TKI therapy in CML is monitored primarily by determining the level of *BCR-ABL1* mRNA transcripts in peripheral blood, normalized to a control RNA and expressed as a percentage on an International Scale (*Arora and Press,*

*2017*). *BCR-ABL1* transcript levels, an approximation of the proportion of circulating malignant cells at any given time, generally decrease exponentially in patients responding to TKI therapy resulting in at least two distinct slopes when plotted semi-logarithmically—an initial rapid decline attributed to TKI-induced killing of more mature myeloid cells, and a subsequent slower decline postulated to represent lower death rates of more primitive leukemic stem/progenitor cells (*Michor et al., 2005*). Clinical resistance to TKI therapy in CML is a significant problem and is classified as acquired resistance (increasing *BCR-ABL1* transcript levels following a substantial decrease) or primary resistance (lack of an adequate initial response). Many patients with acquired resistance have developed mutations in the BCR-ABL1 kinase domain that mediate pharmacological resistance to the TKI (*Ernst and Hochhaus, 2012*). By contrast, 10–15% of newly diagnosed CML patients fail to achieve an 'early molecular response,' defined as the level of *BCR-ABL1* transcripts being less than 10% at 3 mo (*Hanfstein et al., 2012*; *Marin et al., 2012*). Clinical data indicate that switching TKIs may not benefit these patients (*Yeung et al., 2012*; *Yeung et al., 2015*), suggesting that this group is destined to do poorly regardless of the specific inhibitor used. *BCR-ABL1* mutations are generally not present in this group of patients (*Zhang et al., 2009*; *Pietarinen et al., 2016*), and thus the mechanism(s) underlying this primary resistance is unclear. We hypothesized that these variable patient responses to TKI therapy arise from nonlinearity introduced by non-cell-autonomous interactions between normal and CML cells. To test this hypothesis, we developed a novel mathematical model of CML hematopoiesis and TKI treatment that incorporates lineage branching and interactions between normal and CML cells through feedback and feedforward regulation.

Mathematical modeling of leukemia has a long history aimed at understanding disease progression and improving treatment response using single and combination targeted therapies and immunotherapy (*Whichard et al., 2010*; *Pujo-Menjouet, 2015*; *Brunetti et al., 2021*; *Kuznetsov et al., 2021*; *Roeder and Glauche, 2021*). Further, recent efforts have been made to integrate mathematical modeling in clinical decision-making to design personalized therapies (*Hoffmann et al., 2020*; *Engelhardt and Michor, 2021*). Many models of leukemia have utilized simplified lineage architectures and minimal feedback (*Roeder et al., 2006*; *Komarova and Wodarz, 2007*; *Horn et al., 2008*; *Foo et al., 2009*; *Hähnel et al., 2020*; *Pedersen et al., 2021*). While these models can be made to fit the multiphasic disease response data of CML to TKI treatment, the simplicity of the models can make these fitted parameters of limited clinical value. More physiologically accurate, nonlinear models that account for cell–cell signaling and lineage branching are expected to improve clinical relevance. Mathematical models that incorporate feedback signaling have been developed in normal (*Engel et al., 2004*; *Marciniak-Czochra et al., 2009*; *Mahadik et al., 2019*, *Mon Père et al., 2021*) and diseased (*Wodarz, 2008*; *Sachs et al., 2011*; *Krinner et al., 2013*; *Stiehl et al., 2014*; *Stiehl et al., 2015*; *Crowell et al., 2016*; *Woywod et al., 2017*; *Jiao et al., 2018*; *Stiehl et al., 2018*; *Zenati et al., 2018*; *Park et al., 2019*; *Sharp et al., 2020*) hematopoiesis. Because of the vast number of possible ways in which feedback models of normal hematopoiesis and leukemia can be configured, mathematical models tend to greatly simplify the lineage architectures and the feedback interactions among the cell types. For example, *Manesso et al., 2013* developed a hierarchical ordinary differential equation (ODE) model of normal hematopoiesis containing multiple cell types and branch points (16 cell types and 4 branch points) in the lineage tree. Limiting the feedback loops to involve only local, negative regulation (e.g., regulation by self and immediate progenitor/progeny in the lineage tree) results in about $10^6$ models, which enabled the use of a stochastic optimization algorithm to obtain parameters consistent with homeostasis and a requirement for a rapid return to equilibrium following system perturbations.

In the context of leukemia, the model architectures are typically much simpler. Generally, models of leukemia introduce a parallel mutant lineage with the same structure as that used to model the normal hematopoietic cells but with different parameters. For example, Wodarz developed an unbranched lineage ODE model of normal and leukemia stem and differentiated cells in which feedback from the differentiated cells controlled whether the stem cells divided symmetrically or asymmetrically, and demonstrated this provides a mechanism for blast crisis in CML to occur without additional mutations (*Wodarz, 2008*). Krinner et al. incorporated positive and negative feedback regulation of differentiation and proliferation in an unbranched lineage model that combined a discrete agent-based model for the stem cell compartment with an ODE system for the progenitor and differentiated cells to provide a detailed view of the stem cell dynamics and to test the effect of therapies (*Krinner et al.,*

*2013*). *Stiehl et al., 2015* developed an unbranched lineage ODE model of normal and leukemic cells in which only negative feedback regulation of stem and progenitor cell self-renewal fractions was considered, and this was further limited to arise only from factors produced by the postmitotic, mature normal and leukemic cells. Later work extended this approach to investigate clonal selection and therapy resistance (*Stiehl et al., 2014*), the role of cytokines on leukemia progression (*Stiehl et al., 2018*), combination treatment strategies (*Banck and Görlich, 2019*), and niche competition (*Stiehl et al., 2020*). Clonal competition was also considered in an ODE feedback model of CML (*Woywod et al., 2017*) and a stochastic model with feedback (*Dinh et al., 2021*). Simpler unbranched lineage models of normal and leukemic cells in which only the normal cells respond to feedback but normal and leukemic cells compete for space in the BM have been used to investigate regimes of coexistence of normal and leukemic cells (*Crowell et al., 2016*; *Jiao et al., 2018*) and design combination therapies using optimal control algorithms (*Sharp et al., 2020*).

Here, we develop a nonlinear ODE model of normal and CML hematopoiesis using a general approach that integrates an automated method, design space analysis (DSA; *Fasani and Savageau, 2010*), with data gleaned from previously published experiments, and from two new in vivo experiments presented here that separately decrement the number of stem cells and terminally differentiated myeloid cells in the BM of mice. This approach enables us to systematically select among plausible model architectures and signaling interactions without a priori knowledge of which cells are providing and receiving signaling stimuli. We start with a model for normal hematopoiesis that accounts for stem, multipotent progenitor cells, and two types of terminally differentiated cells representing the myeloid and lymphoid lineage branches. This approach allows us to reduce the potential model space from about 60,000 models to a single model class and reveals the existence of feed-forward and feedback mechanisms. We then extend the model to incorporate CML hematopoiesis by introducing a parallel lineage of CML cells with the same model architecture but with different parameters. The model captures the dynamics of CML at various stages of the disease and exhibits variable response to TKI treatment consistent with that observed in clinical data. The model suggests biomarkers of primary resistance, identifies the underlying mechanisms governing the response to TKI therapy, and suggests new treatment strategies.

## Results

### Model of normal hematopoiesis

The primary challenge in developing mathematical models of normal and CML hematopoiesis is sorting through the combinatorial explosion of models that occurs when cell–cell signaling interactions are taken into account. Consider the model hematopoietic system shown in *Figure 1A*, which accounts for hematopoietic stem (**HSC; S**), multipotent progenitor (**MPP; P**), and two types of postmitotic, terminally differentiated cells—myeloid (**TD$_m$**) and lymphoid (**TD$_l$**). The HSC self-renew with fraction (e.g., probability) $p_0$ or differentiate with fraction 1-$p_0$. That is, the fraction of HSC that remain as HSC after division is $p_0$. The MPPs self-renew with fraction $p_1$ and differentiate into either lymphoid or myeloid cells with fractions $q_1$ and 1-$p_1$-$q_1$, respectively. The HSC and MPPs divide with rates $\eta_1$ and $\eta_2$ and the myeloid and lymphoid cells die at rates $d_m$ and $d_l$, respectively. The ODEs that govern the dynamics of the cells are given in 'Methods.' Assuming that there is either positive or negative regulation of the self-renewal and differentiation probabilities and division rates of any cell type from any other cell type results in 59,049 models, counting each combination of regulated signaling as a separate model.

To select the most physiologically accurate models, we first filtered the models using an automated approach (DSA) developed by Savageau and co-workers (*Savageau et al., 2009*; *Fasani and Savageau, 2010*; *Lomnitz and Savageau, 2016*) that enables models to be distinguished based on their range of qualitatively distinct behaviors, without relying on knowledge of specific values of the parameters. This method relies on identifying boundaries in parameter space that separate qualitative behaviors of a particular model, which is much more efficient than searching for model behaviors directly. The boundaries can be approximated from a sequence of inequalities that identify regions where one term on the right-hand side of each ODE (e.g., the rate of change) dominates all others in the sources (positive terms) and another dominates the sinks (negative terms). This is known as a dominant subsystem (S-system) of the model. The number of S-systems in each model depends on the

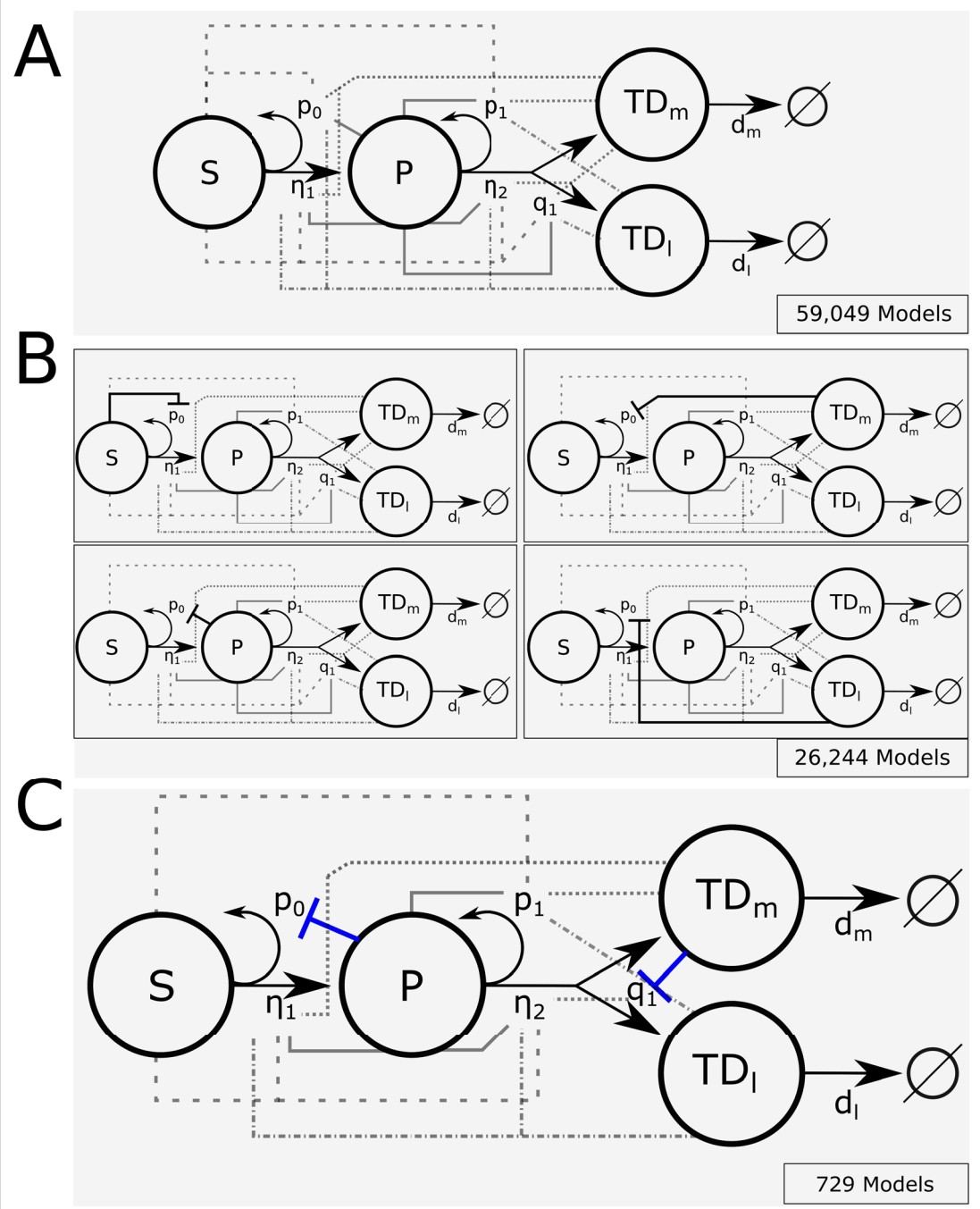

**Figure 1.** Branched lineage model of normal hematopoiesis with feedback regulation. (**A**) Branched lineage model consisting of hematopoietic stem cells (HSC; S), multipotent progenitor cells (MPP; P), and postmitotic, terminally differentiated myeloid (TD$_m$) and lymphoid (TD$_l$) cells. Modulation of the HSC and MPP self-renewal fractions (p$_0$ and p$_1$), division rates ($\eta_1$ and $\eta_2$), and fate switching probability (q$_1$) through feedback can arise from any cell type. The different line styles correspond to regulation by a particular cell type (dashed for S, solid for P, dot-dashed for TD$_l$, and dotted for TD$_m$). (**B**) Using Design Space Analysis, four candidate model classes are identified that differ in how HSCs are regulated. (**C**) Using biological data from the literature as discussed in the text, we reduced the model space by hypothesizing that factors secreted by terminally differentiated myeloid cells direct the fate of MPPs (e.g., IL-6) and those by MPPs suppress HSC self-renewal (e.g., CCL3).

number of combinations of positive and negative terms in the rates of change. If the equilibria of the S-systems, which are determined analytically, are not self-consistent (e.g., consistent with the assumed dominance of terms reflected in the inequalities) or the equilibria are not stable, then the S-system is rejected. If all the S-systems of a particular model are rejected, then that model is removed from

further consideration. Models with at least one self-consistent and stable S-system are viable candidates for further analysis. DSA can be easily automated to make the analysis of very large numbers of equations feasible. Details are provided in 'Methods' and 'Appendix 1' (Section 1). The result of this procedure is the elimination of all but the four model classes shown in *Figure 1B*, which require negative regulation of the stem cell self-renewal fraction but differ by where this regulation arises. The models within the classes share at least one S-system and have common qualitative behaviors. The differences between models in a class lie in whether or not there is positive, negative, or no regulation on the rest of the parameters from any of the cell types. This reduces the number of possible models to 26,244.

Previous work has implicated several feedback mechanisms active in both normal and malignant hematopoiesis. Interleukin-6 (IL-6) is produced by differentiated myeloid cells and acts to bias MPPs toward a myeloid fate (*Reynaud et al., 2011*; *Welner et al., 2015*). Such negative feedback circuits, known as fate control, have been shown to provide an effective strategy for robust control of cell proliferation and reduction of oscillations in branched lineages (*Buzi et al., 2015*). The chemokine CCL3 (also known as macrophage inhibitory protein α [MIP-1α]), produced in BM by basophilic myeloid progenitors (*Baba et al., 2016*), acts to inhibit the proliferation and self-renewal of normal HSC (*Broxmeyer et al., 1989*; *Staversky et al., 2018*), but CML HSC are relatively resistant to its action (*Eaves et al., 1993*, *Baba et al., 2013*). In hypothesizing these regulatory networks, we arrived at a single model class as shown in *Figure 1C*. In this class, there are 729 model candidates, which differ only in how the HSC and MPP cell division rates and the MPP self-renewal fraction are regulated. These above results suggest that IL-6 is a candidate feedback factor expressed in the myeloid compartment ($TD_m$) with the ability to negatively regulate the fraction $q_1$ of MPPs that differentiate into lymphoid cells. CCL3 is a candidate factor mediating negative feedback from the MPP population onto HSC self-renewal. To further constrain the remaining models, we performed cell biological experiments in mice to glean information about cell–cell interactions by separately perturbing the stem cell and myeloid cell compartments.

## Depletion of HSC increases HSC and MPP proliferation

As described in 'Methods,' healthy C57BL6/J (B6) mice were treated with low-dose (50 cGy) ionizing radiation, previously shown to be selectively toxic to HSC in the BM (*Stewart et al., 1998*). The BM stem/progenitor compartment was analyzed by flow cytometry in untreated mice, and on days 1, 3, and 7 post-irradiation, using the gating strategy in *Figure 2A*. These time points and the number of mice analyzed at each time point were informed by a Bayesian hierarchical framework for optimal experimental design of mathematical models of hematopoiesis (*Lomeli et al., 2021*). In particular, the Bayesian framework suggests combining early time points (soon after radiation was applied) with late time points because the early time points provide more information about division rates, while the late time points provide more information about the feedback parameters. One day after treatment, we observed an acute approximately twofold decrease in the relative size of the HSC compartment in the irradiated mice (*Figure 2B*), accompanied by approximately threefold increase in proliferation rates for both HSC and MPPs (*Figure 2C*). There was no significant change in MPP population, however, and the system returned to equilibrium by day 7. These results suggest that the HSC population exerts negative feedback on their own division rate ($\eta_1$) and inhibits the division of MPPs through a negative feedforward loop on $\eta_2$.

## Depletion of mature myeloid cells increases the MPP population

B6 mice were treated with the anti-granulocyte antibody RB68C5 (50 μg), and their BM was analyzed 1 d after treatment (see 'Methods'). This treatment resulted in an ~20% decrease in mature BM myeloid cells, as measured by CD11b expression (*Figure 2D and E*), and was accompanied by a concomitant increase in the size of the phenotypic MPP compartment (*Figure 2E*) and a decrease in the HSC compartment (*Figure 2E*). These results suggest that there is a negative feedback loop from the myeloid cells onto the MPP self-renewal fraction $p_1$.

Taking all these results into consideration, we arrive at the feedback-feedforward model shown in *Figure 2F*. The negative feedback loops shown in blue correspond to those suggested by previous experimental data, while the negative feedback and feedforward loops in red are suggested by the cell depletion experiments presented here. See 'Methods' for a detailed description of the corresponding

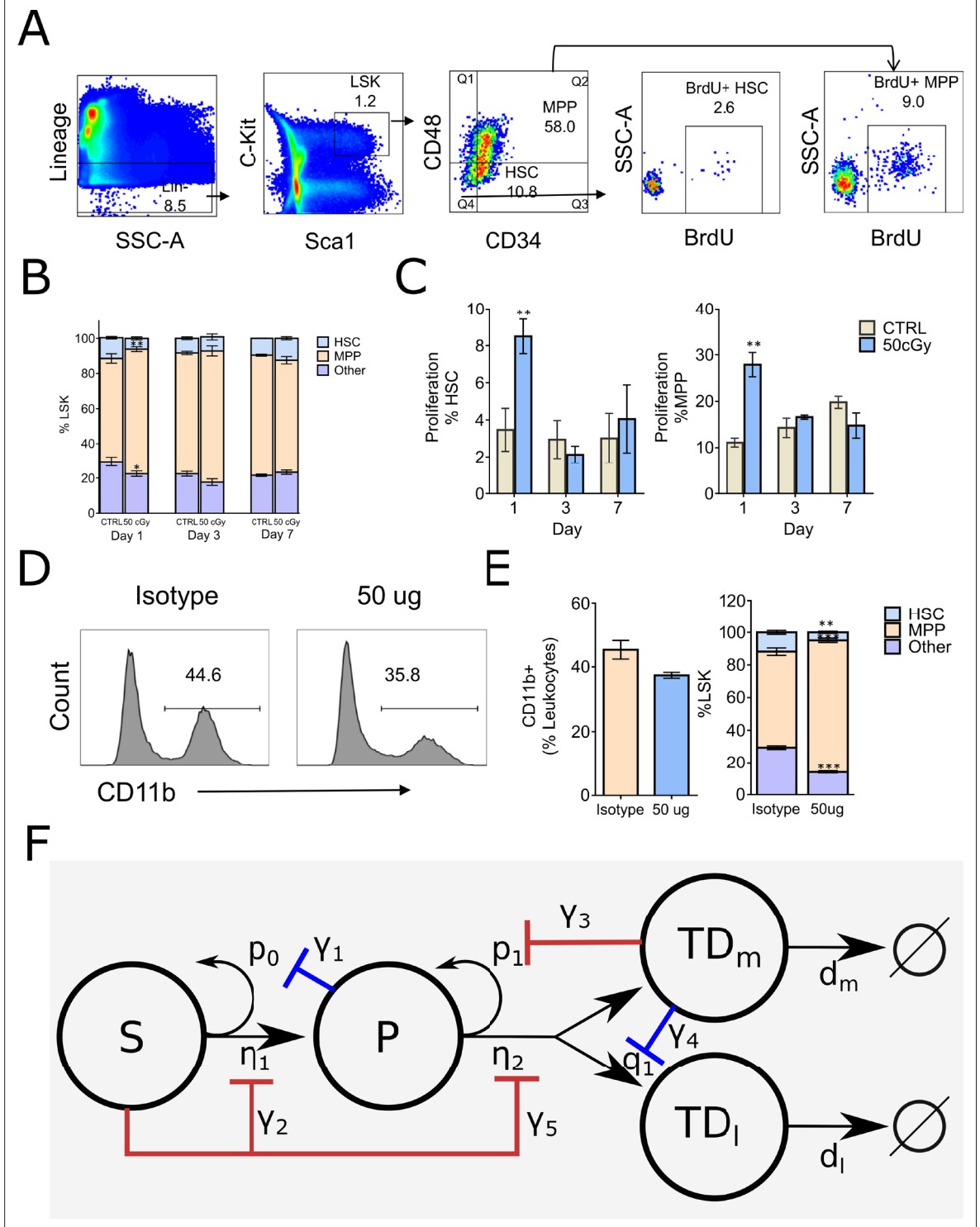

**Figure 2.** Fluorescence-activated cell sorting (FACS) analysis of mouse Lin–Sca-1+c-Kit+ (LSK) bone marrow stem/progenitor cells and the proposed branched lineage hematopoiesis model. (**A**) Gating schema for phenotyping hematopoietic stem cells (HSC, defined as LSK CD34–CD48–) and multipotential progenitors (MPP, defined as LSK CD34+ CD48+), and BrdU incorporation in their respective compartments. (**B**) Distributions of HSC (blue), MPP (orange), and other (purple) compartments on days 1, 3, and 7 in the bone marrow (BM) of control (CTRL) B6 mice and mice that received 50 cGy radiation. (**C**) Frequency of HSC and MPP proliferation in CTRL (gray bars) and irradiated (blue bars) mice measured by BrdU incorporation on days 1, 3, and 7. Data are shown as mean ± SEM. *p<0.05. (**D**) Representative histograms depicting the frequency of myeloid cells as measured by CD11b expression in mice 24 hr after intravenous administration of isotype control (iso) or RB68C5 (50 µg) antibody. (**E**) Left panel: bar graph showing the frequency of CD11b+ cells in BM of mice that were treated with isotype control antibody (Iso; orange bar, n = 3 ) or RB68C5 antibody (50 µg; blue bars,

*Figure 2 continued on next page*

*Figure 2 continued*

n = 3). Right panel: HSC (blue), MPP (orange), and other cell type (purple) frequencies from mice that received isotype or RB6-8C5 antibody. Data are shown as mean ± SEM. *p<0.05. (**F**) Proposed feedforward-feedback model of hematopoiesis with associated feedback strengths denoted with $\gamma_1$–$\gamma_5$. The negative feedback loops shown in blue correspond to those suggested by previous experimental data (*Reynaud et al., 2011*; *Staversky et al., 2018*), while the negative feedback and feedforward loops in red are supported by our cell depletion experiments in (**A–E**).

ODEs. Although these validation data were derived from mice, we hypothesize that similar cell–cell signaling occurs in humans.

## Parameter estimation for feedback-feedforward model of hematopoiesis

To determine biologically relevant parameters for the feedback-feedforward model in *Figure 2F*, a grid-search algorithm was employed. The full ODE model is given in 'Methods' and Appendix 1 (Section 2). The 12 model parameters (proliferation and death rates, self-renewal and branching fractions, feedback/feedforward gains) were sampled using a random uniform distribution for each parameter. See 'Methods,' Appendix 1 (Section 3), and *Appendix 1—tables 2 and 3* for details and a full parameter list. Once parameter values were chosen, the model was simulated for long times. If a parameter set resulted in steady state values consistent with the range of values previously reported for a dynamic human hematopoiesis model (*Manesso et al., 2013*), that parameter set was accepted. Out of $\sim 10^6$ possible parameter combinations, a total of 1493 parameter sets were accepted (*Appendix 1—figure 4*). We further restricted the candidate parameter sets by considering only those with sufficiently large feedforward gains on the MPP division rate ($\gamma_5 > 0.01$) in order to focus on the novel feedforward dynamics. This reduced the number of eligible parameter sets to 563, and their distributions are shown in *Appendix 1—figure 5*. Each of these parameter sets can be thought of as representing the 'normal' condition of a virtual patient by having different individual parameters, for example, due to genetic, epigenetic or environment factors, that nevertheless result in a 'normal' homeostatic hematopoietic system. The different parameter sets thus model a range of variability across individual CML patients. The values of the parameters used are given in Appendix 1, Section 3.

## Sensitivity analyses of hematopoiesis model

DSA can be used to determine qualitative model behaviors and how sensitive the model is to perturbations of key parameters. Here, we focused on the feedback gains $\gamma_1$ and $\gamma_3$ on the HSC and MPP self-renewal probabilities, respectively (see Appendix 1, Section 1.3 for details, and for sensitivity analyses for other parameters, see *Figure 3—figure supplement 1*). As indicated in *Figure 3A*, DSA identifies four regions (design space) in the $\gamma_1$ and $\gamma_3$ plane which the dynamics are governed by different S-systems. Using a parameter set in each design space region (indicated by white dots) as a base value, we performed a parameter sweep in which we vary $\gamma_1$ and $\gamma_3$ in a range within 0.9–1.1 times the magnitude of their original values. In *Figure 3B*, the evolution of each of the cell populations is shown, starting from an initial condition in which there are only a small number of HSC. The different graphs correspond to the parameter sets (*Appendix 1—table 4*) in the four regions of the design space although the dynamics are shown for the full ODE solutions. The solid curves denote results from the original (white dot) parameter set, and the shading denotes the range of behaviors when the parameters are varied. The black and blue curves correspond to the HSC and MPPs, respectively, while the dark-green and light-green correspond to the terminally differentiated myeloid and lymphoid cells. While the system tends to equilibrium for all parameter combinations, the approach to equilibrium is different. The dynamics in regions i and ii are monotonic while those in regions iii and iv are not (e.g., the equilibria in regions i and ii are stable nodes, while those in regions iii and iv are stable spirals). Further, the larger the $\gamma_1$, the faster the approach toward equilibrium. The cell numbers and proportions in each design space region are different as well. In regions i and ii, the HSC dominate while in regions iii and iv the differentiated myeloid cells dominate the population. Further, the number of cells in regions i and iii is larger than those in regions ii and iv. The equilibrium cell populations in region iii correspond more closely to the physiological populations identified by *Manesso et al., 2013*. *Figure 3—figure supplement 2* depicts the effective parameters in region iii as it develops toward the physiological steady state.

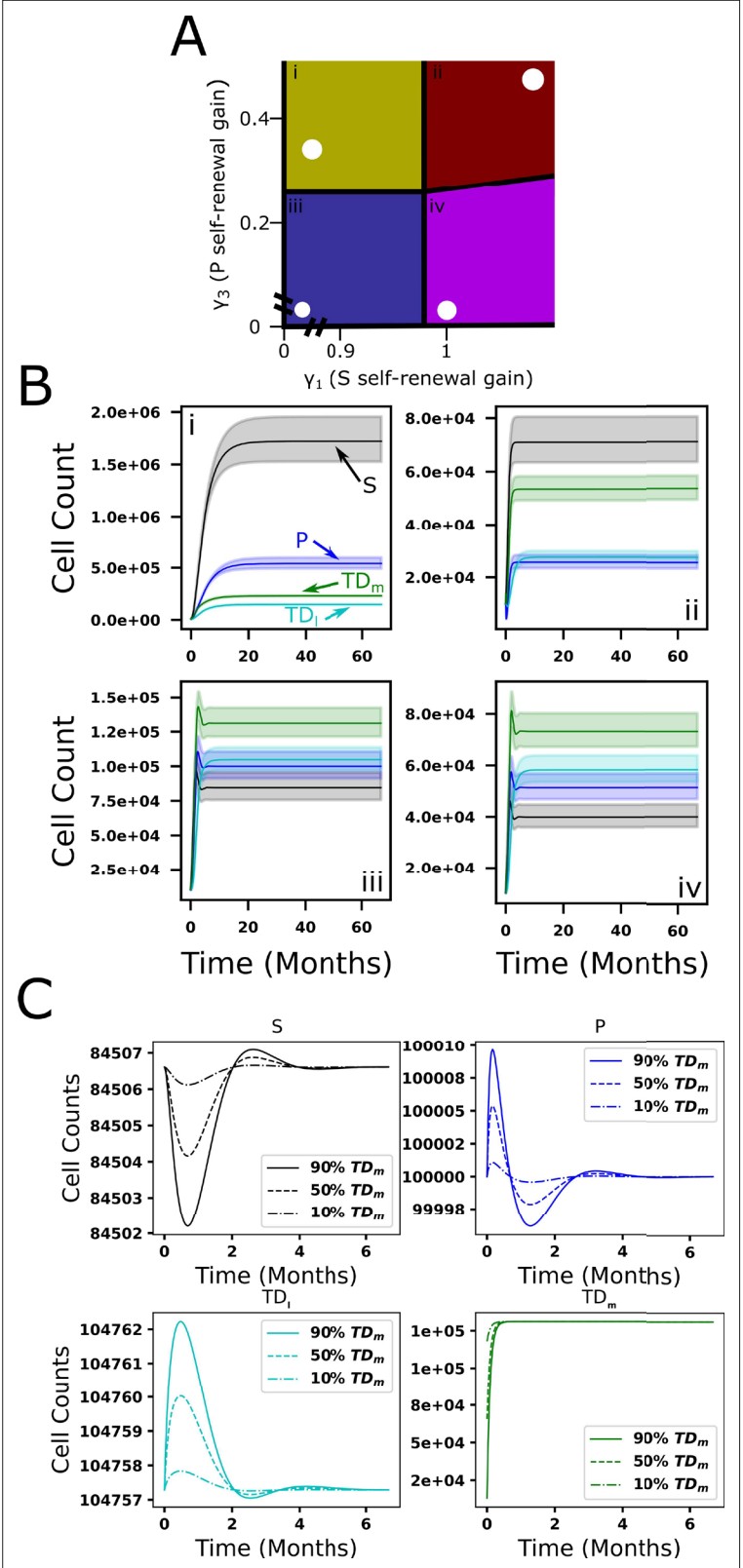

**Figure 3.** Qualitative behavior of feedforward-feedback model and parameter sensitivity. (**A**) The colored regions (i–iv) represent areas of design space in which there are distinct qualitative behaviors as a function of the feedback gains γ₁ and γ₃ of the hematopoietic stem cell (HSC) and multipotent progenitor (MPP) self-renewal fractions, respectively. White dots denote specific parameter combinations. (**B**) The dynamics for each cell compartment

*Figure 3 continued on next page*

*Figure 3 continued*

within each of the four design space regions (i–iv). Solid lines represent ordinary differential equation (ODE) solutions using the specific parameter combinations (black dots in **A**) while the lightly colored regions represent the range of ODE solutions resulting from perturbations in γ₁ and γ₃ in a range within 0.9–1.1 times their original values. The blue and black curves correspond to the HSC and MPPs, respectively, the green and turquoise curves correspond to the myeloid and lymphoid cells. (**C**) The return to equilibrium following partial depletion of mature myeloid cells (10%, 50%, 90%) using the parameter combination (white dot) in region iii.

The online version of this article includes the following figure supplement(s) for figure 3:

**Figure supplement 1.** The dynamics from the steady state with perturbations of each parameter, as labeled, ranging from 90% to 110% of the original parameter value from *Appendix 1—table 4*.

**Figure supplement 2.** Effective parameters for proliferation, self-renewal, and branching as the solutions to the model for the normal hematopoietic system approach steady state.

We next investigated the sensitivity of the model to perturbations about the equilibrium cell population. In *Figure 3C*, we present the results obtained by reducing the number of terminally differentiated myeloid cells from their equilibrium value by 10% (dot-dashed), 50% (dashed), and 90% (solid) and with parameters from design space region iii (*Appendix 1—table 4*). By initially depleting the myeloid cells, which is similar to the experiment in *Figure 2D and E*, the hematopoietic system is shifted away from its steady state. While the presence of the negative feedback loops introduces small magnitude oscillations of the HSC, MPPs, and lymphoid cells, the myeloid cell dynamics are monotonic and the system robustly returns to its steady state over times that are consistent with those established in previous experiments (*Reynaud et al., 2011*) for similar perturbation studies.

## Extension of the hematopoiesis model to CML

Following previous modeling studies, we modeled CML by introducing a parallel lineage of mutant leukemic cells (denoted by the superscript L) but with the behavior of that lineage coupled at many points to the behavior of non-mutant cells, and vice versa. In particular, the model for normal and CML cells shares the same lineage structure and feedback architecture with both normal and mutant cell types providing a source of regulating factors, and although all the leukemic parameters (*Appendix 1—table 3*) could be different from their normal counterparts (*Appendix 1—table 2*), we begin by assuming the only difference between the two lineages is a decrease in the feedback strength for leukemic HSC (**HSC$^L$; S$^L$**), as indicated by $p_0^L$ in the schematic in *Figure 4A*. This makes the leukemic cells less responsive to negative feedback and enables leukemic cells to gain a competitive advantage for growth. One candidate mediator of this negative feedback is CCL3, previously shown to inhibit self-renewal and division of normal HSC but HSC$^L$ are less sensitive to its inhibitory regulation (*Eaves et al., 1993*, *Dürig et al., 1999*, *Baba et al., 2016*; *Staversky et al., 2018*). An example of CML hematopoiesis is shown in *Figure 4B*, where it is seen that, after the introduction of a few HSC$^L$ at equilibrium of the normal hematopoietic system, the CML cells (dashed curves) repopulate the BM at the expense of normal cells (solid curves). Because of negative feedback, the system will eventually reach a new equilibrium consisting solely of leukemic cells. See *Appendix 1—table 5* for the leukemic parameter values, and *Figure 4—figure supplements 1–5* for parameter sensitivity studies of systems containing both normal and CML cells.

We then perturbed each of the leukemic parameters within 10% of the values in *Appendix 1—table 5* and found that only the leukemic stem cell self-renewal parameters—the maximal HSC$^L$ self-renewal fraction $p_{0,max}^L$ and the feedback gain $\gamma_1^L$ on the HSC$^L$ self-renewal fraction—have the potential to significantly influence the results. The results are insensitive to changes in the other leukemic cell parameters (see Appendix 1, Section 9, *Figure 6—figure supplements 2–4*). These results are characteristic of even larger changes in the base parameters.

In this and subsequent parameter investigations, we constrained $p_{0,max}^L$ to be less than or equal to $p_{0,max}$ , the maximal self-renewal fraction of the normal HSC, motivated by the paucity of evidence that $p_{0,max}^L$ is larger than $p_{0,max}$ , coupled with experimental data suggesting that $p_{0,max}^L$ is less than or equal to $p_{0,max}$ . For example, CML long-term culture initiating cells (LTC-IC; thought to be phenotypically similar to stem cells) decrease significantly in in vitro cultures while the number of normal LTC-IC is unchanged, consistent with a relative decrease in self-renewal probability of the CML cells (*Udomsakdi et al., 1992*). In vivo, HSC self-renewal can be assessed directly through transplantation studies.

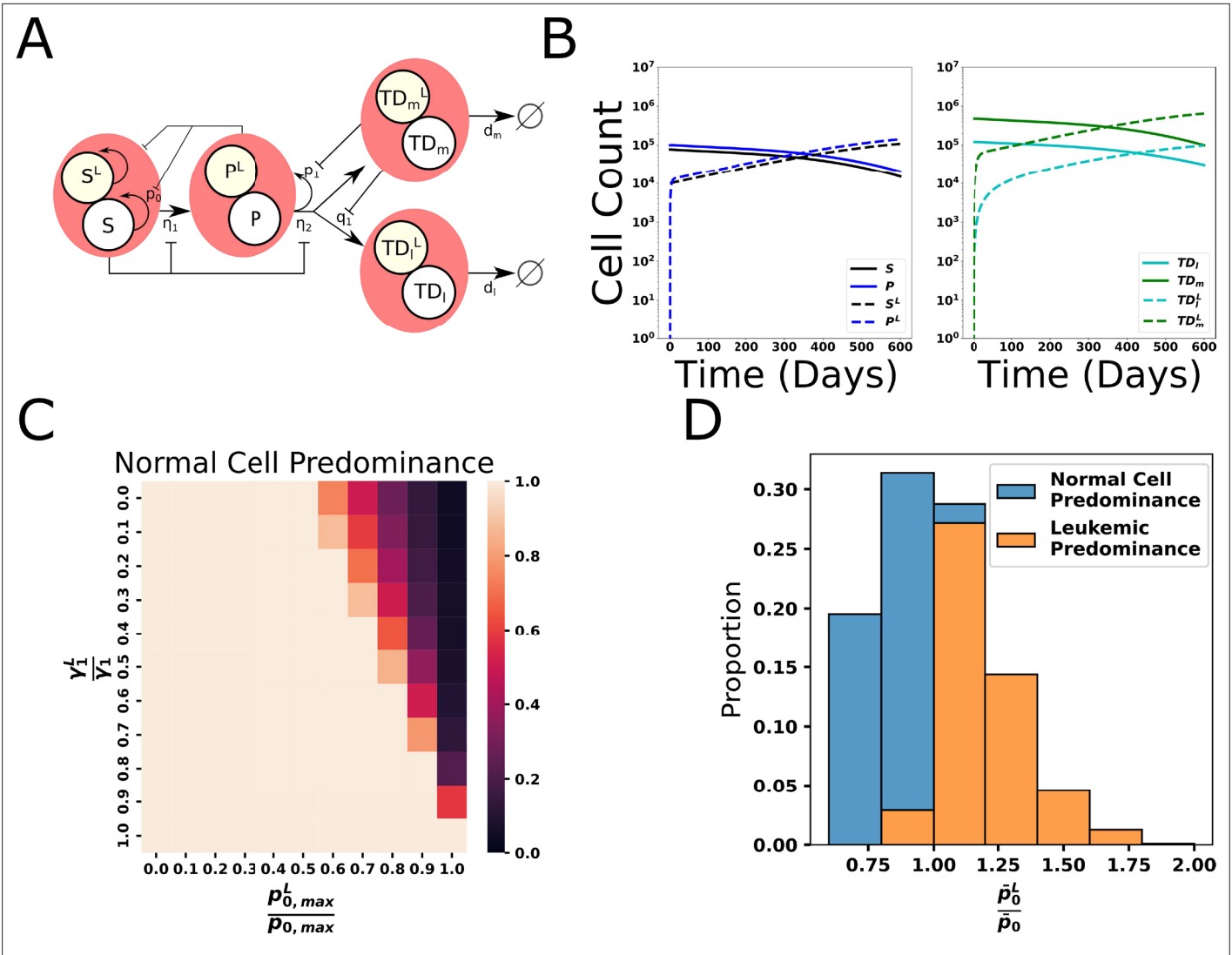

**Figure 4.** Extension of the model of hematopoiesis to chronic myeloid leukemia (CML). (**A**) Schematic of two branched lineages consisting of normal and CML cell compartments. The two lineages share the same feedback architecture. The difference between the two lineages is the leukemic hematopoietic stem cell (HSC) self-renewal is less affected by negative feedback, denoted by $p_0^L$ (see text). (**B**) Dynamics of hematopoiesis upon introduction of CML cells. We begin with having normal hematopoiesis at equilibrium. At time 0, $10^4$ leukemic stem cells (HSC$^L$, S$^L$) cells are introduced to the system and subsequently expand over time at the expense of the normal cells, which decrease. (**C**) Sensitivity analyses of the outcomes of CML hematopoiesis with values corresponding to the proportion of parameter sets where less than 50% of terminal cells are leukemic. (**D**) The fitness of the leukemic stem cells relative to the normal stem cells, as measured by the ratio of their characteristic self-renewal fractions ($\bar{p}_0^L / \bar{p}_0$) determines whether CML will progress and leukemic cells will take over the system after CML stem cells are introduced.

The online version of this article includes the following figure supplement(s) for figure 4:

**Figure supplement 1.** The dynamics from steady state upon introduction of leukemic stem cells with perturbations of each parameter, as labeled, ranging from 90% to 110% of the original parameter value in *Appendix 1—table 5*.

**Figure supplement 2.** The dynamics from steady state upon introduction of leukemic stem cells with perturbations of each parameter, as labeled, ranging from 90% to 110% of the original parameter value in *Appendix 1—table 5*.

**Figure supplement 3.** The dynamics from steady state upon introduction of leukemic stem cells with perturbations of each parameter, as labeled, ranging from 90% to 110% of the original parameter value in *Appendix 1—table 5*.

**Figure supplement 4.** The dynamics from steady state upon introduction of leukemic stem cells with perturbations of each parameter, as labeled, ranging from 90% to 110% of the original parameter value in *Appendix 1—table 5*.

**Figure supplement 5.** The dynamics from steady state upon introduction of leukemic stem cells with perturbations of each parameter, as labeled, ranging from 90% to 110% of the original parameter value in *Appendix 1—table 5*.

*Figure 4 continued on next page*

*Figure 4 continued*

**Figure supplement 6.** Variations in the other leukemic parameters and their associated gamma for all eligible parameter sets.

In this regard, CML HSC engraft immunodeficient mice variably and inefficiently compared to normal human HSC (*Wang et al., 1998*) while HSC from *BCR-ABL1* transgenic mice exhibit an engraftment defect upon secondary transplantation into syngeneic recipients (*Schemionek et al., 2010*). Both results are suggestive of a relative decrease in self-renewal capacity of *BCR-ABL1*+ stem cells.

Next, we performed a sweep through leukemic stem cell self-renewal parameters $p_{0,max}^L$ and $\gamma_1^L$ for each of the eligible parameter sets for normal hematopoiesis (see below). We found that for the terminally differentiated cell proportion to be at least 50% leukemic (darker regions), there are biological constraints upon the combination of $p_{0,max}^L$ and $\gamma_1^L$ (*Figure 4C*). As the heat map shows, in order for CML to dominate hematopoiesis (e.g., terminally differentiated cell proportion >50% leukemic), the CML stem cells should have $p_{0,max}^L$ sufficiently close to $p_{0,max}$ and $\gamma_1^L$ should be sufficiently small. As the ratio $p_{0,max}^L / p_{0,max}$ decreases from 1, the system requires smaller feedback gains $\gamma_1^L$ to compensate and allow for CML to develop. Further, there are threshold values of the parameters required for CML hematopoiesis to prevail. Namely, the system is dominated by normal cells (CML cells do not 'take over') when $p_{0,max}^L / p_{0,max}$ is sufficiently large or when $\gamma_1^L/\gamma_1$ is sufficiently small.

To further examine these biological constraints, we calculated characteristic effective self-renewal fractions for normal and leukemic stem cells, defined as $\overline{p}_0^L = p_{0,max}^L / \left(1 + \gamma_1^L \overline{N}\right)$ and $\overline{p}_0 = p_{0,max} / \left(1 + \gamma_1 \overline{N}\right)$, where $\overline{N} = 10^5$, a characteristic value for the size of the MPP population based on MPP steady state values (*Manesso et al., 2013*). The relative fitness of the CML cells defined by the ratio of characteristic values of the HSC$^L$ and HSC self-renewal fractions: $\overline{p}_0^L / \overline{p}_0$. Here, all eligible parameter sets representing the states of the normal system are considered and the leukemic parameters $p_{0,max}^L / p_{0,max}$ and $\gamma_1^L/\gamma_1$ are varied from 0.6 to 1.0 and 0.1–0.6, respectively. In *Figure 4D*, we examined the relative fitness of leukemic cells through the distribution of the ratio of characteristic values colored by leukemic cells outcompeting normal (orange) and normal cells maintaining majority (blue). As expected, the larger the relative fitness, the more likely that CML will take over the system and dominate hematopoiesis. For further analysis of the leukemic parameter combinations for CML hematopoiesis and under treatment, see *Figure 4—figure supplement 6*, *Figure 6—figure supplements 2–4*, *Figure 7—figure supplement 2*, *Figure 8—figure supplement 2*, Appendix 1, Sections 9–11, and *Appendix 1—figures 11–17* for details.

## Validation of the CML model

To test whether our mathematical model recapitulates known features of CML biology, we simulated a published transplant experiment in a transgenic mouse model of CML that recapitulates the main features of human CML (*Reynaud et al., 2011*). In this experiment, either HSC$^L$ or leukemic MPPs (MPP$^L$) were implanted into sublethally irradiated mice (*Figure 5A*). Transplantation of HSC$^L$ enables engraftment and myeloid cell production that leads to CML. On the other hand, transplanting MPP$^L$s does not allow for long-term engraftment but results in a larger fraction of donor-derived lymphoid cells after 35 d (*Figure 5B*). This study presented evidence that IL-6 produced by differentiated myeloid cells reprograms these MPP$^L$ progenitors toward a myeloid fate (*Reynaud et al., 2011*). As described in 'Methods' and Appendix 1, Section 4, we modeled this experiment by reducing the number of cells in equilibrium to mimic the effects of sublethal radiation. We explored a range of possible reductions of HSC$^L$, MPP$^L$, and differentiated myeloid and lymphoid cells and tracked the outcomes when 4000 HSC$^L$ or MPP$^L$ were introduced after the decrements from equilibrium. We then discarded those parameter sets that did not yield results consistent with a simple majority of myeloid cells for HSC$^L$ transplant and a simple majority of lymphoid cells for MPP$^L$ transplant (*Reynaud et al., 2011*). In particular, 85 parameter sets were discarded, leaving a total of 478 parameter sets remaining. Characteristic results are presented in *Figure 5C and D* when the reductions for HSC$^L$, MPP$^L$, and terminally differentiated cells were 55, 35, and 10%, respectively, from their equilibrium values. See *Figure 5—figure supplement 1* for results using other decrements, the removed parameter set criteria (*Figure 5—figure supplement 2*), and *Figure 5—figure supplement 3* for the final parameter distributions. When HSC$^L$ are transplanted (solid curves), the

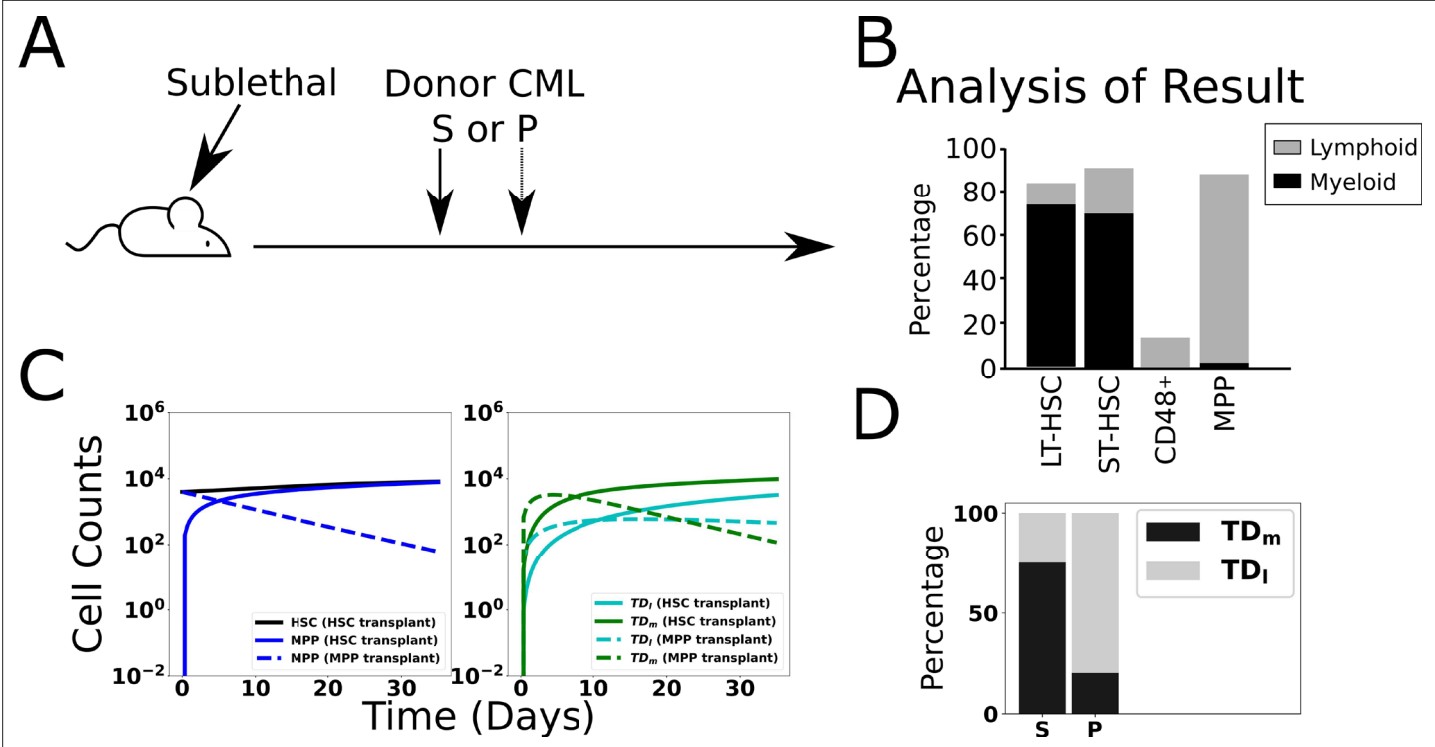

**Figure 5.** Validation of model through simulated transplant. Results of a transplant experiment from **Reynaud et al., 2011**. Schematic (**A**) depicting the experimental pipeline and results (**B**), adapted from Figure 2A-C in **Reynaud et al., 2011**. When HSC^L are transplanted into sublethally irradiated mice, chronic myeloid leukemia (CML)-like leukemia is induced and the myeloid cells expand. When leukemic multipotent progenitor cells (MPP^L, P^L) are transplanted, they do not stably engraft and transiently produce a larger fraction of differentiated lymphoid cells. (**C**) Simulated time evolutions of donor-derived HSC^L, MPP^L, and terminally differentiated lymphoid (TD_l), and myeloid (TD_m) cells when HSC^L (solid) or MPP^L (dashed) cells are transplanted. (**D**) Bar chart showing model predictions of the percentages of donor-derived myeloid and lymphoid cells after 35 d when HSC^L or MPP^L are transplanted, which is consistent with the experimental data in (**B**; see text).

The online version of this article includes the following figure supplement(s) for figure 5:

**Figure supplement 1.** Heat map depicting the outcomes of transplant experiments in the presence of decrements of 50–70% HSC^L and 30–50% MPP^L from their equilibrium values (see text for details).

**Figure supplement 2.** The dynamics of each parameter set that does not match the experimentally observed behavior of the transplant experiments from **Reynaud et al., 2011**.

**Figure supplement 3.** Distributions of the remaining 478 parameters after removal determined through the depletion sweep.

donor-derived MPP^L (**Figure 5C**, left) rapidly increased as did the terminally differentiated myeloid and lymphoid cells (**Figure 5C**, right). Consistent with the experiments, there is a larger fraction of donor-derived myeloid cells than lymphoid cells after 30 d (**Figure 5D**). In contrast, when MPP^L are introduced (dashed curves), their population decreases (**Figure 5C**, right) because the MPP^L do not stably engraft. Concomitantly, there is burst of donor-derived myeloid and lymphoid cells at early times (**Figure 5C**, right) as the transplanted MPP^L differentiate.

The early time dynamics of the myeloid and lymphoid cells depend on the specific values of the MPP^L self-renewal ($p_1$) and fate control ($q_1$) fractions, whose values in turn depend on the number of myeloid cells through negative feedback regulation. In particular, if $1 - p_1 > 2q_1$, then more myeloid than lymphoid cells will be produced at early times, as in **Figure 5C** (right), whereas more lymphoid cells will be produced if $1 - p_1 < 2q_1$. In both cases, because the MPP^L's do not stably engraft and instead differentiate into lymphoid and myeloid cells, we observe that there is a larger fraction of donor-derived lymphoid cells after 30 d (**Figure 5D**), consistent with the experiments. This occurs because there is a decreasing flux of differentiating cells since there is no stable engraftment and the lymphoid cells are longer-lived (smaller death rate) than the shorter-lived myeloid cells, which have a larger death rate.

## Leukemic stem cell load influences TKI therapy outcomes

We next explored the effects of TKI therapy on CML in the model. While the overall size of the phenotypic HSC compartment is not increased in CML patients (*Jamieson et al., 2004*), the proportion of HSC$^L$ in the BM can vary widely across newly diagnosed CML patients from a few percent to nearly 100% (*Petzer et al., 1996*; *Diaz-Blanco et al., 2007*; *Abe et al., 2008*; *Thielen et al., 2016*). We therefore investigated how the HSC$^L$ load in the BM affects therapy outcomes. We used one eligible parameter set (see *Appendix 1—table 4*), out of all 478 parameter sets all of which are capable of characterizing the normal state of our simplified model of the hematopoietic system and one choice of leukemic parameters (see *Appendix 1—table 4*) in which the only difference between normal and leukemic cells is that the HSC$^L$ are one half as sensitive to negative feedback regulation compared to the normal HSC ($\frac{\gamma_1^L}{\gamma_1} = 0.5$). The initial condition was obtained by simulating the development of CML, analogous to that shown in *Figure 4B*, prior to initiating therapy. TKI treatment was initiated at three different times to achieve varying leukemic stem cell load (6, 18, and 36 mo) and was simulated by introducing a death rate of HSC$^L$ and MPP$^L$ proportional to their proliferation rates, with the HSC$^L$ proliferation rate lower than that of normal HSC (*Jørgensen et al., 2006*). While some studies have shown that primitive CML stem/progenitor cells are relatively resistant to killing by TKIs in vitro (*Graham et al., 2002*; *Corbin et al., 2011*), clinical studies suggest that long-term TKI therapy can decrement the CML stem cell compartment, at least in some patients (*Etienne et al., 2017*; *Chen et al., 2019*), consistent with mathematical modeling of patient *BCR-ABL1* transcript data (*Tang et al., 2011*). This supports the concept that TKIs possess a measure of leukemia stem cell killing ability, and we therefore included this effect in our model. The TKI treatment parameters were the same across the three cases. See 'Methods' and Appendix 1 for details and *Appendix 1—tables 4 and 5* for parameter values. Thus, these cases can be thought of as representing the response of one virtual patient to TKI therapy implemented at different times after disease initiation.

At an early treatment time with lower (<90%) initial HSC$^L$ fractions (HSC$^L$, *Figure 6A*), the numbers of MPP$^L$ (blue-dashed), leukemic terminally differentiated lymphoid (light-green-dashed), and myeloid (dark-green-dashed) decrease rapidly at the early stages of treatment and are accompanied by a rapid increase in HSC$^L$ due to the loss of negative feedback from the MPP$^L$. This loss of negative feedback from the MPP$^L$ also results in a rapid increase in the number of normal HSC (black solid curves) that subsequently drives an increase in the normal MPPs (blue solid). The increased number of HSC and HSC$^L$ decreases their division rates due to the autocrine negative regulatory loop as well as the division rates of the MPPs and MPP$^L$ through feedforward negative regulation. This decreases the flux into the terminally differentiated cell compartments (both normal and leukemic), thereby decreasing their numbers at early times. At later times, both the HSC$^L$ and MPP$^L$ gradually decrease in response to TKI-induced cell death, which drives an accompanying decrease in the leukemic differentiated cells. A small, transient increase in MPP$^L$ is observed before the gradual decline. This is driven primarily by the increase in flux into the MPP$^L$ compartment by differentiating HSC$^L$, although there is also a small contribution from the feedforward regulation of the MPP$^L$ division rate, which lowers the effectiveness of TKI therapy on the MPP$^L$. Both of these effects are reduced as the HSC$^L$ numbers are decreased by TKI therapy. This in turn increases the effectiveness of TKI therapy in killing leukemic cells at later times and enables the normal cells (solid curves) to rebound toward their pre-leukemic equilibrium values.

At intermediate treatment time with larger (90–99%) HSC$^L$ fractions (*Figure 6B*), the responses of the leukemic and normal cells to TKI treatment at early times are qualitatively similar to those observed in the previous case although the effects are more pronounced. The increase in HSC$^L$ is much larger than the previous case because there are fewer normal cells to compete with in the BM. This significantly decreases the HSC/HSC$^L$ and MPP/MPP$^L$ division rates through the negative feedback/feedforward regulation and correspondingly the rates of TKI-induced cell death. Accordingly, at later times the MPP$^L$ population rebounds, driven by the flux of differentiating HSC$^L$, and eventually the system reaches a state in which both normal and leukemic cells coexist. The stem cell compartment is dominated by HSC$^L$ which are largely quiescent, while the multipotent progenitor and terminally differentiated cell compartments have a higher fraction of normal cells. This is consistent with experimental results from mouse models (*Reynaud et al., 2011*) and our own unpublished data. In this scenario, *BCR-ABL1* transcript levels in the peripheral blood are ~1–9%, but the patient would not respond further to TKI treatment and hence would not reach MR3; this has been observed clinically including one of the patients in our study (see below). The small flux of differentiating normal and

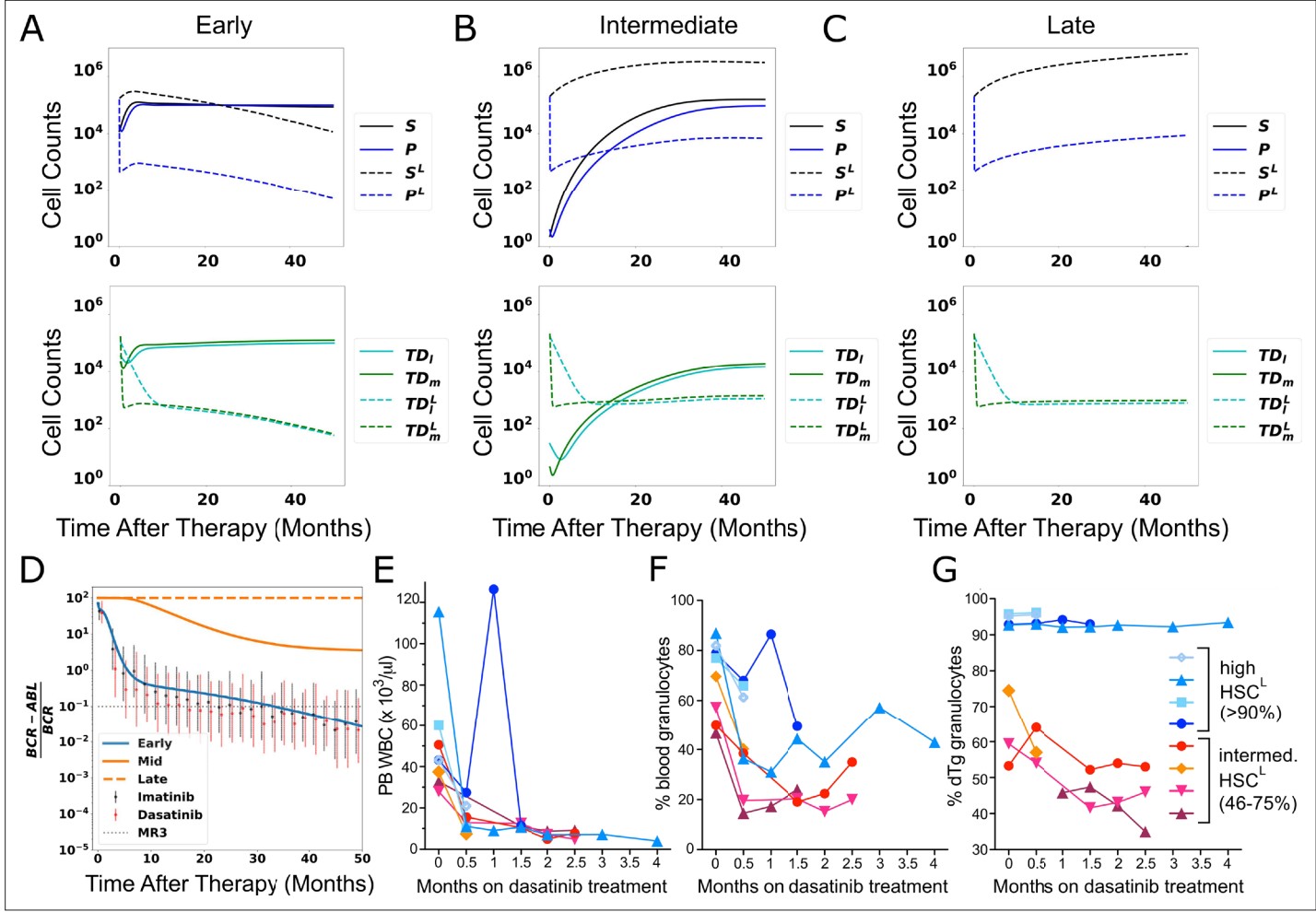

**Figure 6.** The response of chronic myeloid leukemia (CML) to tyrosine kinase inhibitor (TKI) therapy. (**A–C**) Simulated cell dynamics of normal and leukemic cells in response to TKI therapy that is started at different time points in CML development (**A**, early times; **B**, intermediate times; **C**, late times). See text. (**D**) Simulated molecular response curves corresponding to the application of TKIs for each of simulations in (**A–C**). The simulated molecular response from (**A**) (blue) compares well with clinical data (symbols) measuring treatment responses to two different TKIs (imatinib, dasatinib) averaged across a cohort of patients (*Glauche et al., 2018*). The simulated molecular responses from (**B**) and (**C**) (orange solid and dashed curves) are indicative of primary resistance. (**E–G**) Experiments in chimeric mice (see text) that show that the size of the leukemic stem cell clone correlates with decreased response to TKI therapy. Peripheral blood (PB) leukocyte counts (**E**), percentage of PB granulocytes (**F**), and PB *BCR-ABL1*[+] (leukemic) granulocyte chimerism (**G**) are shown in cohorts of mice treated with dasatinib. Blue symbols are chimeras bearing >90% *BCR-ABL1*[+] HSC[L], red-orange symbols are chimeras bearing 46–75% *BCR-ABL1*[+] HSC[L].

The online version of this article includes the following figure supplement(s) for figure 6:

**Figure supplement 1.** Effective parameters for proliferation, self-renewal, and branching after leukemic stem cells are added to the normal system at steady state, and after tyrosine kinase inhibitor (TKI) therapy begins for the cases shown in *Figure 6* in the main text.

**Figure supplement 2.** The dynamics from steady state upon introduction of leukemic stem cells with perturbations of each leukemic parameter, as labeled, ranging from 90% to 110% of the original parameter value (*Appendix 1—table 4* for normal cells, *Appendix 1—table 5* for leukemic cells), except for $p_{0,max}^{L}$, which is never varied above the $p_{0,max}$ value.

**Figure supplement 3.** The dynamics from steady state upon introduction of leukemic stem cells with perturbations of each leukemic parameter, as labeled, ranging from 90% to 110% of the original parameter value (*Appendix 1—table 4* for normal cells, *Appendix 1—table 5* for leukemic cells).

**Figure supplement 4.** The dynamics from steady state upon introduction of leukemic stem cells with perturbations of each leukemic parameter, as labeled, ranging from 90% to 110% of the original parameter value (*Appendix 1—table 4* for normal cells, *Appendix 1—table 5* for leukemic cells).

leukemic stem and progenitor cells, combined with the negative feedback loops on the self-renewal and branching fractions, supports nearly steady populations of differentiated cells.

When given years to develop and a late time to treatment, the HSC[L] fraction is nearly 100% (*Figure 6C*), and there are so few normal stem cells that the leukemic cells easily maintain nearly 100%

of each cellular compartment even in the presence of TKI therapy. Aside from a short-lived, transient decrease in MPP$^L$ (and differentiated leukemic cell) numbers, the leukemic cells are largely unresponsive to TKI therapy because the feedback/feedforward negative regulation of stem and progenitor cell division rates makes these rates so low that the TKIs are largely ineffective in killing the leukemic stem and progenitor cells. As in the previous case, the negative feedback regulation and the small fluxes of differentiating leukemic stem and progenitor cells enables the system to approach a steady state containing only leukemic cells.

In *Figure 6D*, we plot the simulated *BCR-ABL1* transcript levels over time for the three scenarios. As described in 'Methods,' the transcripts are modeled using a relative ratio of leukemic and normal terminally differentiated myeloid and lymphoid cell numbers. The solid blue curve corresponds to CML using the treatment time from *Figure 6A*, which responds to TKI therapy. Just as in the clinical data (symbols), the response to TKI therapy produces a biphasic exponential decrease in *BCR-ABL1* transcripts, which decreases below $10^{-1}$, representing a so-called major molecular response (MMR or MR3), which represents a major goal of therapy in CML as the risk of relapse and leukemia-related death is virtually nonexistent once this milestone is achieved (*Hochhaus et al., 2017*). Consistent with previous interpretations, the rapid initial decrease in *BCR-ABL1* transcripts is due to TKI-induced cell death of MPP$^L$ and the increase in normal HSC and MPPs, which induce corresponding changes in the myeloid and lymphoid cells (*Figure 6A*). The long-term, slower depletion of leukemic cells and the stable normal cell populations result in the second phase of the biphasic response. The simulated results compare well with clinical data from the DAISISON study of imatinib versus dasatinib in patients with newly diagnosed CML (*Cortes et al., 2016*) where the data corresponds to mean *BCR-ABL1* transcripts, with standard deviations, adapted from *Glauche et al., 2018* for patients who received imatinib (blue) or dasatinib (red).

The two other curves in *Figure 6D* correspond to the treatment times from *Figure 6B* (solid orange) and C (dashed orange). In these cases, the *BCR-ABL1* transcripts do not decrease below the MR3 threshold, indicating that neither of these virtual patients responds adequately to TKI therapy. There is a partial response in patient from *Figure 6B* as the transcripts initially decrease due to TKI-mediated death of MPP$^L$, but this effect soon saturates because the leukemic stem cells are able to drive the regrowth and persistence of leukemic progenitor and differentiated cells. For the virtual patient with parameters from *Figure 6C*, there is essentially no change in the *BCR-ABL1* transcripts when therapy is applied. These behaviors are consistent with those observed in CML patients with primary resistance to TKI therapy (*Zhang et al., 2009*; *Yeung et al., 2012*; *Pietarinen et al., 2016*).

## HSC$^L$ load influences the response to TKI therapy in a mouse CML model

The fundamental difference between these three virtual patients is the number of leukemic stem cells at the start of therapy, which occurs because treatment is initiated at different times following the development of CML (early—6 mo after CML initiation ~93% initial HSC$^L$ fraction, intermediate—18 mo after CML initiation ~99% initial HSC$^L$ fraction, late—36 mo after CML initiation ~99.99% initial HSC$^L$ fraction). Our results suggest that the higher the HSC$^L$ fraction at the start of therapy, the less effective the therapy. This follows from the feedback/feedforward regulation where increased numbers of HSC and HSC$^L$ decrease their own proliferation rates as well as those of the MPPs and MPP$^L$ (see *Figure 6—figure supplement 1* and *Appendix 1—figure 6* for further explorations of feedback/feedforward regulation of parameters). This reduces the effectiveness of TKI therapy as evidence suggests TKIs preferentially target dividing leukemic cells (*Graham et al., 2002*; *Corbin et al., 2011*) and suggests a mechanism why some patients are destined to do poorly with TKI therapy.

To test this hypothesis, we created BM chimeric mice containing both normal and leukemic (*BCR-ABL1*$^+$) HSC by transplantation of BM from conditional *BCR-ABL1* transgenic mice (*Koschmieder et al., 2005*) into unirradiated congenic recipient mice. Following stable engraftment, *BCR-ABL1* expression is induced in transgenic HSCs by withdrawal of doxycycline (see 'Methods'). These chimeras represent a novel and physiologically accurate in vivo model of early CML development that reflects interactions between normal and CML cells in a BM microenvironment unperturbed by radiation (*Rodriguez et al., 2022*). Two cohorts of chimeric mice bearing either a high HSC$^L$ burden (94 ± 1.5% of the HSC population) or an intermediate HSC$^L$ burden (58 ± 12%) were treated with dasatinib (25 mg/kg daily by oral gavage). Both cohorts showed a hematological response to TKI therapy, with

decreased peripheral blood leukocyte counts (*Figure 6E*) and a decreased percentage of circulating granulocytes (*Figure 6F*). By contrast and consistent with the predictions of the quantitative model, while mice bearing smaller populations of HSC$^L$ showed a decrease in the percentage of circulating *BCR-ABL1*$^+$ granulocytes in response to TKI therapy, mice with the highest HSC$^L$ burden showed virtually no decrease in circulating leukemic cells (*Figure 6H*). Because the level of circulating granulocytes reflects the proportion of BM HSC (*Wright et al., 2001*; data not shown), these results demonstrate that TKI therapy is unable to decrement the HSC$^L$ compartment in mice with predominantly *BCR-ABL1*$^+$ HSC at the start of treatment.

## HSC self-renewal as an additional determinant of TKI response

While analyses of clinical data also show that patients with lower leukemic stem cell burden are more likely to respond to TKI treatment (*Thielen et al., 2016*), some patients with a high percentages of leukemic stem cells at the start of treatment are nonetheless still capable of responding to TKI therapy (e.g., see Figure 3 in *Thielen et al., 2016*). This suggests that leukemic stem cell burden alone does not predict the molecular response to TKIs. To investigate this further, we tested the response to TKI treatment for each of our 478 parameter sets. In *Figure 7A*, we present the results using only one choice of leukemic parameters (see *Appendix 1—table 5*). Other choices of leukemic parameters give similar results (see *Appendix 1—figure 13*). The model outcomes bear a striking resemblance to the clinical data of *Thielen et al., 2016*. The leukemic stem cell fraction does influence TKI response, but treatment outcomes are seen to vary among virtual patients within the same initial leukemic stem cell load. We then asked what characteristics (e.g., parameter sets) distinguish whether a virtual patient achieves a MR3 response within 50 mo. We also varied the HSC$^L$ parameters, taking into account several studies suggesting that CML stem cells are at least 5–10 times less sensitive to CCL3-mediated inhibition of self-renewal (*Eaves et al., 1993*, *Chasty et al., 1995*; *Wark et al., 1998*; *Dürig et al., 1999*); for example, $\frac{\gamma_1^L}{\gamma_1}$ should be less than 0.2. Further, since 10–15% of patients do not respond to TKI treatment even in the absence of *BCR-ABL1* mutations (*Hanfstein et al., 2012*; *Marin et al., 2012*), we estimate $p_{0,max}^L / p_{0,max} \approx 0.8$ from *Appendix 1—figure 11* to roughly match this proportion of nonresponding patients. Taken together, this suggests that the effective leukemic stem cell fitness would be $\frac{\bar{p}_0^L}{p_{0,max}} \approx 0.7$. We thus varied the HSC$^L$ parameters accordingly. In *Figure 7B*, we plot the results for $\frac{\bar{p}_0^L}{p_{0,max}} \geq 0.7$ as a bivariate histogram for $\frac{\bar{p}_0^L}{p_{0,max}}$ and $p_{0,max}$ with proportion of response (blue) and nonresponse (orange) for every parameter set. Surprisingly, we found that although the fitness $\frac{\bar{p}_0^L}{p_{0,max}}$ impacts response, the parameter that clearly distinguished responders from nonresponders was the maximal self-renewal fraction $p_{0,max}$ for normal stem cells, shown in *Figure 7B* (marginal y-axis). See *Figure 7—figure supplement 1* for the distributions as a function of the other parameters using the single leukemic parameter set from *Figure 7A* (see *Appendix 1—table 5*), and *Appendix 1—figures 15–16* in Appendix 1, Section 10 for different bivariate distributions corresponding to different choices of the minimum fitness $\frac{\bar{p}_0^L}{p_{0,max}}$. In particular, larger values of $p_{0,max}$ and $\gamma_1$ are correlated with a decreased response to TKI therapy after leukemia develops. Although these parameters are associated with normal HSC, the self-renewal fraction $p_0^L$ of HSC$^L$ and feedback strength $\gamma_1^L$ depends on these parameters since we assumed the fitness of the CML stem cells $\frac{\bar{p}_0^L}{p_{0,max}}$ is larger than a minimum threshold. Therefore, increasing the self-renewal fraction of the normal stem cells has the effect of also increasing the fitness of the CML stem cells.

To understand further the differences between response and nonresponse to TKI therapy, we took the parameter set from *Figure 6A* as a representative patient for response and selected an arbitrary nonresponsive parameter set (*Appendix 1—table 6*) to be a representative patient for nonresponse. In *Figure 7C*, we show that the effective $p_0^L$ (the fraction of HSC$^L$ self-renewal after feedback) for nonresponders (orange) is larger after TKI therapy is applied than for responders (blue). In particular, as TKI treatment kills the leukemic progenitors, this increases the effective self-renewal fraction for both normal and leukemic stem cells because of the release of negative feedback. When the maximum self-renewal $p_{0,max}$ is larger, the leukemic stem cells experience an acute increase in self-renewal, resulting in their dominance over normal stem cells that then leads to a decreased response to TKIs.

Clinical data provide support for this mechanism of resistance. Patients with clonal hematopoiesis, in which there is a dominant clone driving hematopoiesis, exhibit predominantly normal hematopoiesis

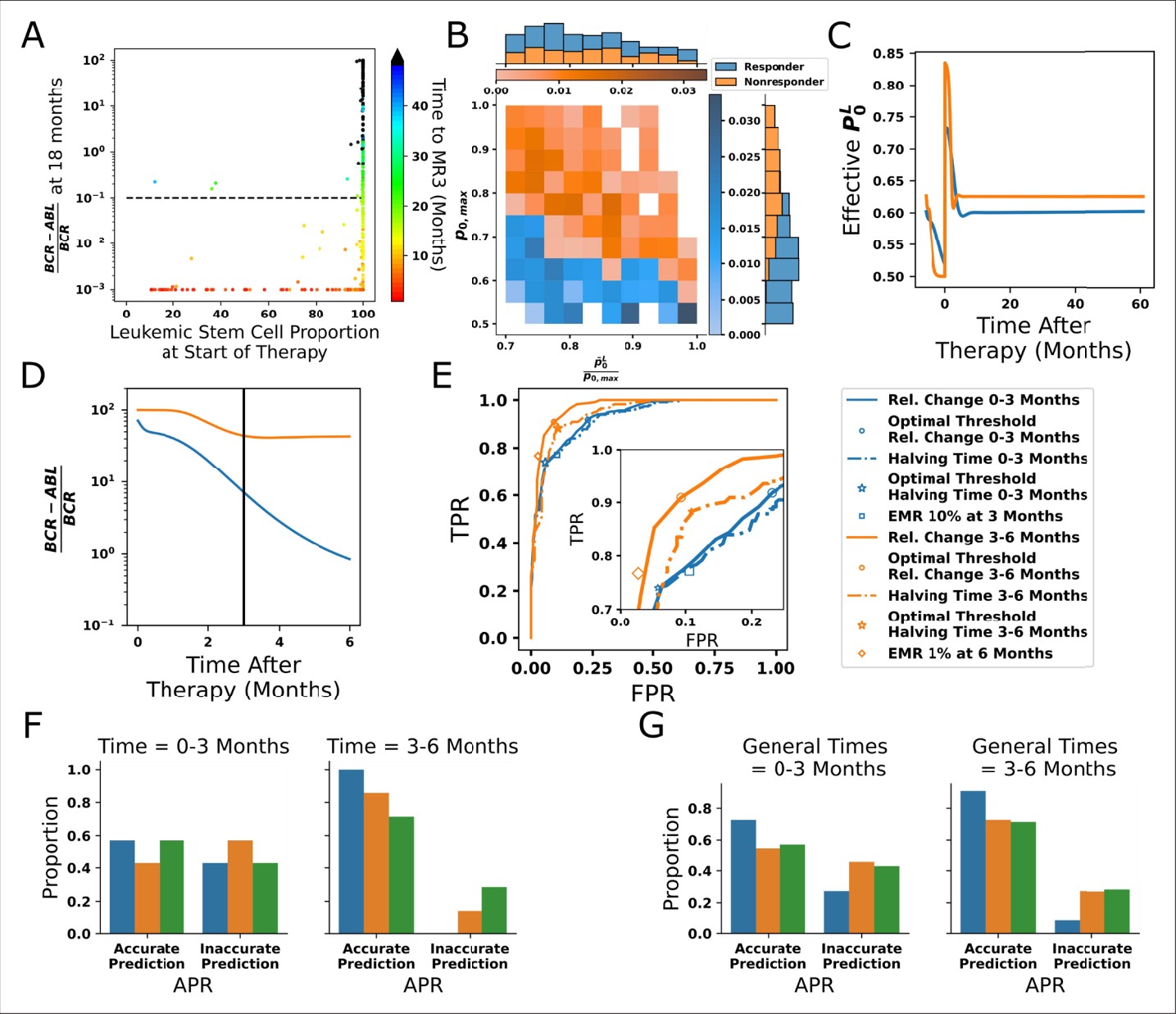

**Figure 7.** Leukemic stem cell load alone does not predict response to tyrosine kinase inhibitor (TKI) therapy. (**A**) Scatter plot of the distribution of simulated *BCR-ABL1* transcripts at 18 mo after start of treatment as a function of the HSC$^L$ proportion at the start of TKI therapy for each of the 478 parameter sets. The time to reach MR3 (*BCR-ABL1* transcripts less than 0.1%) is indicated by the color. (**B**) The proportion of parameter sets that achieve MR3 within 50 mo (responders, blue) and those that do not (nonresponders, orange) shown as a joint distribution of the parameters $p_{0,max}$ and $\bar{p}_0{}^L/p_{0,max}$ using the minimum fitness threshold $\frac{\bar{p}_0{}^L}{p_{0,max}} \geq 0.7$, which reveals the maximal self-renewal fraction of the normal HSC, $p_{0,max}$ (y-axis marginal distribution), distinguishes response across parameter combinations. (**C**) Dynamics of the effective leukemic stem cell self-renewal fraction for the parameter set used in *Figure 6A–D* (blue) and an arbitrary representative non-responsive parameter set (orange) during chronic myeloid leukemia (CML) development and before initiation of therapy (t < 0), and after application of TKI therapy (t > 0). (**D**) Early time dynamics (e.g., t = 0–3 mo; left of the vertical line) of the transcript levels reveal that it is difficult to distinguish responders (blue) from nonresponders (orange). At later times (e.g., t = 3–6 mo; right of the vertical line), the two populations are easier to distinguish. (**E**) Receiver operating curves (ROC) obtained from the 478 parameter sets using our new prognostic criterion based on the relative changes in transcript levels (solid) and the transcript halving time (dashed) for the first 3 mo (blue) and the second 3 mo (orange) after therapy. The prognostic thresholds (symbols) are identified by optimizing true and false positives rates. Early molecular response (EMR) at 3 mo (10% transcript levels) and 6 mo (1% transcript levels) are shown by the blue square and orange diamond, respectively. Inset: expanded view of the 'elbow' region of the ROC curves to display differences between the prognostic tests. Accuracy is improved using the 3–6 mo time window, and our new prognostic criterion outperforms the EMR and halving time prognostics in this time window. (**F**) The accuracy of the prognostic criteria applied to CML patient data (n = 7) treated using the same TKI dosing for the 6-month period after therapy is started.

*Figure 7 continued on next page*

*Figure 7 continued*

The results are consistent with the synthetic data in (**E**). (**G**) The prognostic criteria applied to patient data (n = 7) in which therapy could be changed but those changes were maintained for 6 mo (see text). Although the data are limited, the results are consistent with those in (**E**) and (**F**) suggesting increased accuracy using the 3–6-month window, and that the prognostic criterion based on relative change may yield more accurate predictions than EMR and halving time in the 3–6-month time frame.

The online version of this article includes the following figure supplement(s) for figure 7:

**Figure supplement 1.** Parameter distributions by response to tyrosine kinase inhibitor (TKI) treatment using synthetic data.

**Figure supplement 2.** Comparison of the proportion of cases that successfully respond to therapy as a function of the ratio of characteristic values of leukemic and normal stem cells: $\bar{p}_0^L/\bar{p}_0$.

but frequently have mutations in the genes, such as *TET2*, *DNMT3A*, and *ASXL1*, that are known to increase stem cell self-renewal (*Steensma, 2018*). Clinical data shows that CML patients whose blood cells have mutations in *TET2* and *ASXL1*, some of which may exist prior to development of CML, frequently exhibit a poor response to TKI therapy (*Kim et al., 2017*; *Marum et al., 2017*). Taken together, these data suggest that patients with higher stem cell self-renewal fare worse when their CML is treated using TKIs than patients with lower stem cell self-renewal.

## Predicting long-term response to TKI treatment

Several measures of the response of CML patients to TKI therapy have been developed, based on *BCR-ABL1* transcript levels in peripheral blood. Here we test a new, model-driven criterion for predicting patient response and compare the results with several other criteria currently used in the clinic. A major focus has been on the predictive value of the decline in transcripts over the first 3 mo of treatment, principally the so-called 'early molecular response' or EMR (defined as *BCR-ABL1* transcripts <10% at 3 mo and <1% at 6 mo), where patients with >10% transcripts had significantly lower probability of achieving cytogenetic remission and decreased overall survival (*Hanfstein et al., 2012*; *Marin et al., 2012*). Subsequently, there was an effort to improve the predictive power by focusing on the velocity of reduction in transcripts (*Branford et al., 2014*; *Hanfstein et al., 2014*; *Pennisi et al., 2019*). Because the best predictor of patient response to TKIs, the self-renewal fraction of normal stem cells, is very difficult to measure clinically, we searched for an alternative criterion that could accurately predict patient response and could still be measured using the data collected in standard practice. Therefore, we focused on alternative time frames and calculation methods for assessing *BCR-ABL1* transcript levels (*Figure 7D*). It is important to be able to predict the long-term TKI response early after starting treatment in order to enable changes in therapy. However, since both responders (blue) and nonresponders (orange) may show significant decreases in the transcript levels in the first months of treatment, it was difficult to distinguish between the two at relatively early time points. By contrast, responses in the 3–6-month time frame make it easier to identify the different behaviors of responders and nonresponders (*Figure 7D*).

By calculating the relative changes of the transcript levels from 3 to 6 mo, we developed a prognostic formula: $PF\left(3, 6\right) = \left(BCRABL1\left(6\right) - BCRABL1\left(3\right)\right)/BCRABL1\left(6\right)$. We found that optimizing for sensitivity (TPR, the true positive rate) and specificity (1-FPR, with FPR being the false positive rate) resulted in a prognostic threshold of $PF \approx -3.2$, with sensitivity of $\approx 0.91$ and specificity of $\approx 0.91$ (*Figure 7E*, orange curves) compared to the optimal sensitivity and specificity of the velocity-based prognostic ($\approx 0.89$ and $\approx 0.88$, respectively) and $\approx 0.77$ and $\approx 0.98$ for EMR 1% with our parameter sets. This demonstrates that this prognostic tool had higher sensitivity and specificity than previously developed predictive criteria in separating responders ($< -3.2$) from nonresponders ($> -3.2$), where response is defined as achievement of MR3 within a clinically relevant timeframe of 18 mo. We also tested the various prognostics at the 0–3-month interval as is the current clinical practice, but that resulted in lower predictive power (*Figure 7E*, blue curves). These results highlight the importance of including the 3–6-month TKI response in predicting the long-term outcome of treatment, instead of considering only the first 3 mo. See *Appendix 1—figures 7 and 17* in Sections 8 and 12, respectively, for further discussion, comparison of additional prognostic criteria, and the effect of leukemic parameters.

We then applied our prognostic criterion to anonymized CML patient data (see 'Methods') to determine clinical significance and utility. The prognostic tests shown in *Figure 7E* were calculated for both the first 3 mo and the subsequent 3–6-month period after the start of therapy, for the patients

who were treated with the same TKI and dosage for the full 6 mo. All the prognostic tests achieved a more accurate prediction of patient outcome using the 3–6-month data compared to the same test applied to the first 3 mo (*Figure 7F*). To expand clinical utility, the prognostics were calculated for cases where TKI therapy was changed (due either to toxicity or an inadequate response) but then maintained for a subsequent 6-month period, which were added to the data from *Figure 7F*. The aggregated data (*Figure 7G*) reaffirms the improved accuracy in prediction using the 3–6-month transcript data compared to that from 0 to 3 mo. Over the first 3 mo, all the prognostic criteria performed similarly. Although the number of patients was small, the results suggest that our prognostic criteria may perform better than the EMR and velocity-based prognostics that are in current clinical use. For comparisons between the prognostic criteria and time frames with patient data, see Appendix 1, Section 7, *Appendix 1—figures 8–10*.

## Improving response to therapy: Combining TKIs with interventions that promote differentiation

Our model suggests that combination therapy to modulate the stem cell self-renewal rate, in addition to directly targeting the leukemic HSC and MPPs with TKI therapy, might counteract TKI treatment resistance mediated by high stem cell self-renewal. Such pro-differentiation therapy could be accomplished through either direct stimulation of differentiation or through suppression of self-renewal. In our modeling experiments, we explored the impact of this approach through the suppression of self-renewal (see 'Methods' for details). To begin the exploration of the combined TKI-differentiation therapy, we performed this combination therapy on each of our 478 parameter sets, which represents a population of CML patients with person-to-person variability. We then recorded which parameter sets achieved MR3 within 50 mo for each strength of the differentiation therapy ($\Delta$), where $\Delta$ is a dimensionless constant greater than 0 that constantly suppresses stem cell self-renewal (both normal and CML) in the setting of combination therapy (see 'Methods' and Appendix 1, *Equations 40 and 41*). Using these data, *Figure 8A* depicts the proportion of parameter sets achieving MR3 given a strength of differentiation therapy of $\Delta$. As differentiation therapy strength increases from zero, the proportion of parameter sets that achieve MR3 increases before leveling off between $\Delta = 0.2$–$0.3$, with maximum efficacy occurring at a strength of differentiation $\Delta$ of about 0.24. The efficacy of combination therapy then begins to decline rapidly, and with too great a strength of differentiation treatment, the combination therapy becomes inferior to TKI therapy alone.

To investigate how combination therapy effectively targets resistance, and the mechanism of the decreased efficacy of combination therapy in achieving MR3 when $\Delta$ is large, we returned to examining parameter distributions. *Figure 8B* depicts the same distribution of $p_{0,max}$ as in *Figure 7B*, but overlaid with a second histogram (hatched regions) to denote the effect of the differentiation therapy at the point of maximum efficacy (see *Figure 8—figure supplement 1* for all the parameter distributions). The two types of hatching reveal important factors that determine under which conditions combination therapy improves or impairs response. The orange hatching represents transition from response to nonresponse by combination therapy; this occurs in individuals with the lowest $p_{0,max}$. In these cases where stem cell self-renewal is already close to the ideal effective self-renewal fraction of 0.5, differentiation therapy pushes too many normal cells into differentiation, causing the normal cell populations to deplete themselves and decreasing the efficacy as $\Delta$ increases beyond 0.24. In contrast, the blue hatching shows the desired scenario of nonresponding individuals with high $p_{0,max}$ becoming responders and achieving MR3 within 50 mo due to the combination therapy.

To understand further the mechanisms underlying the efficacy of combination therapy, we explored treatment dynamics (changes in *BCR-ABL1* transcript levels) and the rates of change of both normal and leukemic stem cell populations for the nonresponsive individual from *Figure 7D*. By applying two different strengths of differentiation therapy ($\Delta = 0.24$ and $0.5$) in combination with TKI therapy, for this individual both strengths are able to achieve MR3 at ~18 mo (*Figure 8C*) in contrast to TKI monotherapy, which resulted in a failure to reach MR3 (*Figure 7B*, orange). *Figure 8D and E* show how the rates of change in the size of the normal and leukemic stem cell compartments vary with respect to time for the three different $\Delta$ values. For TKI therapy alone ($\Delta = 0$), rates of growth of both the normal and leukemic stem cell populations show an increase as a result of the loss of negative feedback due to TKI-induced killing of $MPP^L$, but the leukemic stem cells experience a much greater numerical increase and outcompete normal cells, resulting in a system that exhibits resistance to

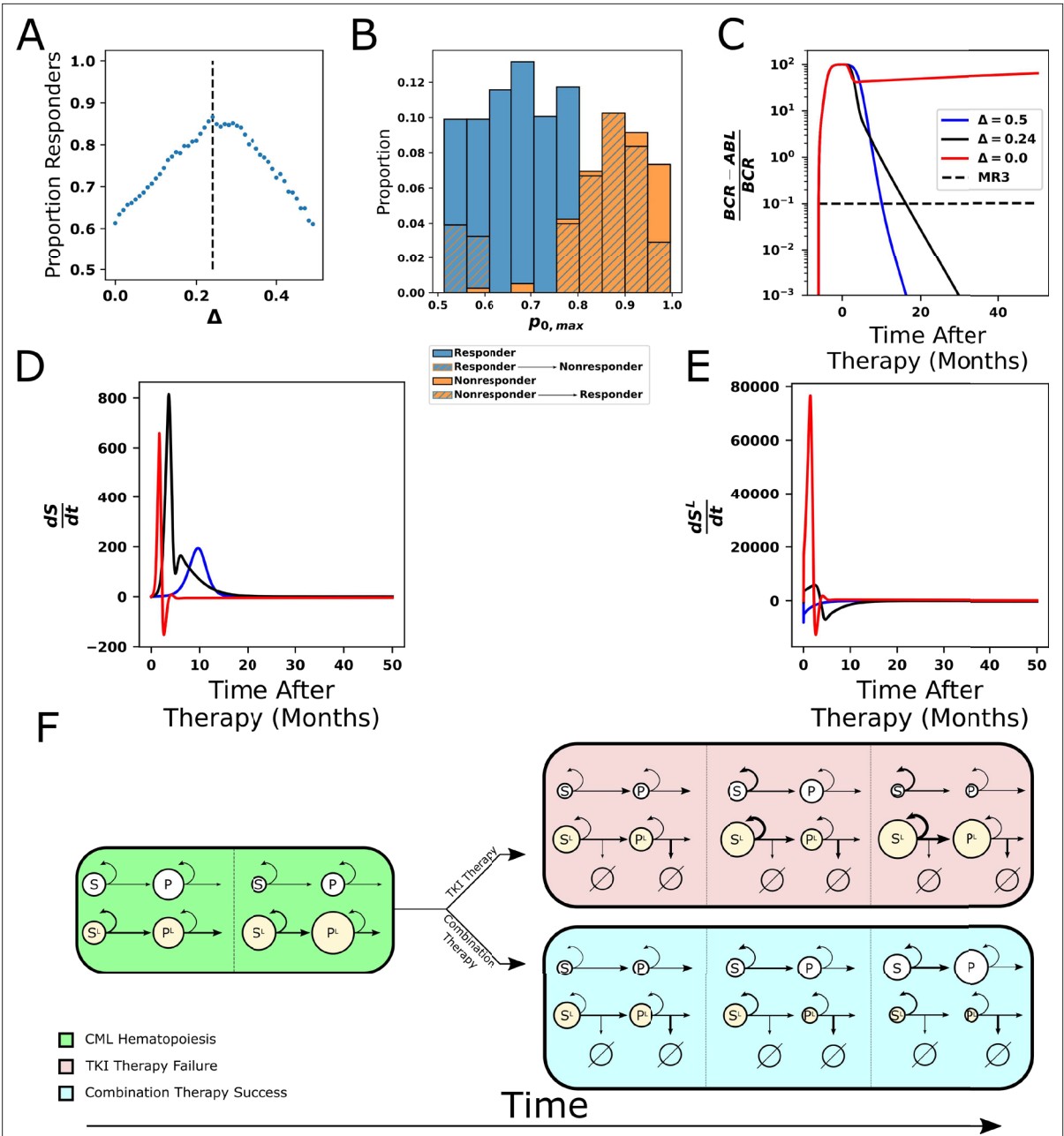

**Figure 8.** Combining tyrosine kinase inhibitor (TKI) therapy with differentiation promoters enhances response to treatment. (**A**) The proportion of the 478 parameter sets that achieve MR3 under combined TKI and differentiation therapy depends nonmonotonically on the strength Δ of the differentiation therapy, with the peak response (86.8%) occurring at Δ = 0.24. (**B**) The maximum stem cell self-renewal fraction for a single $\bar{p}_0^L/p_{0,max}$ in *Figure 7B* (marginal y) with hatching indicating the effects of the combination of TKI and differentiation therapy with Δ = 0.24. Blue hatching indicates nonresponders (who did not achieve MR3) that become responders (achieve MR3) while orange hatching indicates responders that become nonresponders upon combined treatment. Differentiation promoters allow nonresponders to TKI therapy with large self-renewal fractions to reach MR3. The opposite outcome, loss of MR3 in a TKI responder, primarily occurs only at the smallest self-renewal fractions. (**C**) Time evolution of *BCR-ABL1* transcripts during combination therapy, with Δ = 0.24 (black) and Δ = 0.5 (blue), using the parameter set from *Figure 5B* that does not achieve MR3 using TKI monotherapy (red). (**D, E**) The time derivatives of the number of normal (**D**) and leukemic (**E**) stem cells during combination therapy. The differentiation promoter attenuates the rapid increases in the rates of change at early times after therapy starts in both normal and leukemic cells, but the attenuation is much larger in the leukemic cells. This results in the growth of normal cells, while leukemic cells experience restricted growth or outright depletion depending upon the differentiation therapy strength. (**F**) Simplified diagram representing the key interactions between the cells and the impact on outcomes of TKI and combination therapy. Green: chronic myeloid leukemia (CML) hematopoiesis depicting the loss of normal stem cells and progenitors and the increase in leukemic stem cells and progenitors. Red fill: TKI treatment failure. The TKI-induced death of leukemic progenitors

*Figure 8 continued on next page*

*Figure 8 continued*

relieves negative feedback and increases stem cell self-renewal, resulting in increases in both normal and leukemic stem cells, and eventually their progeny (panel 2). The increases are larger for the leukemic cells because their self-renewal fraction is bigger. Increases in the leukemic progenitor compartment (panel 3) drive down the self-renewal fraction of normal stem cells proportionally more than for the leukemic stem cells. The increases in $HSC^L$ also drive down proliferation rates, which makes the leukemic cells less responsive to TKI treatment. Altogether, this makes the leukemic cells more fit than the normal cells and results in therapy failure. Blue fill: treatment by combined TKI and pro-differentiation therapy reduces stem cell self-renewal relative to TKI monotherapies, equalizes the normal and leukemic self-renewal fractions, which limits leukemic stem cell growth and limits decreases in proliferation rates, making the $HSC^L$ and $MPP^L$ more susceptible to TKI-induced death (panel 2). This allows repopulation of the bone marrow by normal stem cells and progenitors to occur (panel 3).

The online version of this article includes the following figure supplement(s) for figure 8:

**Figure supplement 1.** Changes in the parameter distributions from *Figure 7—figure supplement 1* when combined differentiation and tyrosine kinase inhibitor (TKI) therapy is administered.

**Figure supplement 2.** Variations in $\frac{p_{0,max}^L}{p_{0,max}}$ and $\frac{\gamma_1^L}{\gamma_1}$ for all parameter sets with combination therapy show improvement of response with combination therapy compared to *Appendix 1—figure 11* (bottom right).

the TKI therapy. Under conditions of maximum efficacy ($\Delta = 0.24$), the normal stem cell population rate of change still increases rapidly but to a maximum level below that for TKI monotherapy before decreasing more rapidly to zero as the system re-equilibrates. In contrast, under the same conditions the rate of change of the leukemic stem cell population is greatly reduced and becomes negative after normal stem cells begin to outcompete the leukemic stem cells. Under conditions of stronger differentiation therapy ($\Delta = 0.5$), although the accumulation rate of the normal stem cells is substantially reduced, the growth rate of the $HSC^L$ is immediately negative. This enables the normal HSC to easily outcompete the leukemic cells and restore the system to the normal state. Effectively, for large values of HSC self-renewal and corresponding feedback gains, the differentiation promoter acts to bring the self-renewal fraction of the normal HSC closer to that of the $HSC^L$, which then enables the TKI therapy to disadvantage the leukemic cells, and allow for repopulation and dominance by normal cells. For the effect of combination therapy on different combinations of leukemic parameters, see *Figure 8—figure supplement 2* and Appendix 1, Section 11 for further analysis.

## Discussion

In this work, we developed a nonlinear mathematical model of normal and CML hematopoiesis that incorporated feedback control, lineage branching, and signaling between normal and CML cells. Using ODEs, we modeled the dynamics of the stem, multipotent progenitor, and terminally differentiated cell populations. To filter through the combinatorial explosion of models that occurs when cell–cell signaling interactions are taken into account, we focused first on normal hematopoiesis. We used DSA (*Savageau et al., 2009*; *Fasani and Savageau, 2010*; *Lomnitz and Savageau, 2013*; *Lomnitz and Savageau, 2016*), an approach that enables models to be distinguished based on their range of qualitatively distinct behaviors without relying on knowledge of specific values of the parameters, to perform an automated search for regions of stability in thousands of proposed models and efficiently eliminate unphysiological, unstable models. When combined with previous observations and new in vivo data to further constrain cell–cell interactions, we arrived at a new feedback-feedforward model (*Figure 2F*).

Using cell perturbation experiments in mice, we validated several features of the model, including feedback from differentiated myeloid cells on MPP self-renewal, and feedforward regulation by stem cells on proliferation of stem and MPP compartments. We postulate that these regulatory loops may also regulate human blood cell production. While there are some known differences between mouse and human hematopoiesis (*Parekh and Crooks, 2013*), many signaling pathways are conserved between species. For example, the role of IL-6 in regulating lymphoid differentiation (e.g., $\gamma_4$ in *Figure 2F*) has been validated in mice (*Reynaud et al., 2011*) and human samples (*Welner et al., 2015*), while CCL3 mediates negative feedback from progenitors onto stem cell self-renewal (e.g., $\gamma_1$ in *Figure 2F*) in both mice (*Staversky et al., 2018*) and humans (*Broxmeyer et al., 1989*). TGF-β, produced by HSC, differentiated myeloid cells, and BM stroma, is a candidate factor regulating negative feedback of HSC onto their own division rate and that of the MPPs (*Zhao et al., 2014b*; *Naka and Hirao, 2017*), while IL-6 may inhibit MPP self-renewal and increase myeloid differentiation (*Zhao*

*et al., 2014a*), at least under stress conditions. The role of these candidate hematopoietic regulators could be tested directly in our mouse model via a genetic approach. Moving forward, it will be important to validate results from mouse model systems in human studies whenever possible.

We used a grid-search algorithm to determine a set of approximately 500 biologically relevant parameter sets for our new model. While we could have used other approaches such as the Latin hypercube algorithm to sample our multidimensional parameter space (*Read et al., 2018*), we chose to perform a gridsearch because of the ease of implementation and the fact that our goal was not to exhaustively search the full parameter space, but rather to obtain a set of biologically relevant parameter values consistent with normal hematopoietic homeostasis. In particular, using each parameter set in the model yields steady states that are consistent with normal ranges of hematopoietic cells. These parameter sets model a population of individuals with normal cell counts but person-to-person variability of parameters due, for example, to genetic, epigenetic and/or environmental differences.

We then extended the model to incorporate CML hematopoiesis by introducing a mutant lineage with the same structure as the normal system. We incorporated one of the central features of CML pathophysiology, that the leukemic stem cell clone, hypothesized to arise from a single HSC that acquires a Ph chromosome, has a competitive advantage over normal HSC and with time comes to dominate the stem cell compartment (*Dingli et al., 2010*; *Thielen et al., 2016*; *Holyoake and Vetrie, 2017*; *Majeti et al., 2022*). This competitive advantage could be a consequence of positive feedback (autocrine or paracrine) on the $HSC^L$ population or negative feedback with different strengths for normal and leukemic stem cells. Candidate mediators of such positive and negative feedback include interleukin-3 (*Jiang et al., 1999*) and CCL3 (*Baba et al., 2016*), respectively. Our current model incorporated differential negative feedback of MPPs on HSC (*Figure 4A*) with the $HSC^L$ being less sensitive to the negative feedback than are the normal HSC, which is consistent with CCL3 (*Eaves et al., 1993*, *Baba et al., 2013*). This one difference provided leukemic cells with a competitive advantage for growth, and in the absence of treatment, the leukemic cells will take over the BM at the expense of normal cells (*Figure 4B*). Upon exploration of the leukemic parameter space, we found that only the leukemic cell parameters for the leukemic stem cells ($HSC^L$)—the maximal $HSC^L$ self-renewal fraction $p_{0,max}^L$ , the feedback gain $\gamma_1^L$ on the $HSC^L$ self-renewal fraction, and the TKI-induced $HSC^L$ death rate $TKI_{HSCL}$—have the potential to significantly influence the results. The results are insensitive to changes in the other leukemic cell parameters (see *Figure 4—figure supplement 6*, *Figure 6—figure supplements 2–4*, and *Appendix 1—figure 11*).

When combined with TKI therapy, the feedback/feedforward model exhibited variable responses to TKI treatment, consistent with those observed in CML patients. That is, although our 500 parameter sets were consistent with normal hematopoietic cell counts, the responses to TKI treatment were highly variable, with some sets responding to treatment while others did not. The model predicted that a contributor to primary TKI resistance is the overall proportion of HSC that are leukemic, consistent with experimental data in mice (*Figure 6G*) as well as patient data (*Thielen et al., 2016*). However, leukemic stem cell burden alone does not predict the molecular response to TKIs, as observed both clinically (*Thielen et al., 2016*) and in our data (*Figure 7A*), since some patients with high $HSC^L$ fractions in their BM nonetheless still respond to TKIs.

The model suggested that a key predictor of reduced response to TKI treatment is an increased tendency of normal hematopoietic stem cells to self-renew, which in turn influences self-renewal of the leukemic stem cells since they were estimated to be sufficiently fit with respect to the normal stem cells. This is also consistent with clinical data that suggest that CML patients whose normal and leukemic cells share mutations in genes such as *TET2* and *ASXL1*, which are known to increase stem cell self-renewal (*Steensma, 2018*), tend to have inferior outcomes under TKI therapy (*Kim et al., 2017*; *Marum et al., 2017*). This is illustrated in *Figure 8F* (red panel), where the high initial $HSC^L$ population and the subsequent decline of progenitor cells reveals the effect that high stem cell self-renewal has on driving TKI resistance. In our model, the presence of a *TET2* or *ASXL1* mutation in both normal and leukemic stem cells that led to a proportional increase in self-renewal in both populations would tend to cause resistance to TKI therapy, provided that the $HSC^L$ are sufficiently fit in the presence of the mutations, which we would expect. The self-renewal-driven resistance we describe herein challenges the prevailing paradigm that TKI resistance is proliferation-driven and a consequence of $HSC^L$ quiescence (*Graham et al., 2002*; *Corbin et al., 2011*).

Because stem cell self-renewal is hard to quantify experimentally, we developed a clinical prognostic criterion to predict TKI response based on the relative changes in the *BCR-ABL1* transcripts over a 3-month period. Using the synthetic data from our normal and leukemic parameter sets, we found that using changes in transcripts from 3 to 6 mo was very effective in predicting the long-term outcome of treatment (e.g., reaching MR3 within 18 mo). In contrast, using transcript data from 0 to 3 mo resulted in less accurate predictions. This observation also holds for prognostic criteria based on EMR and transcript halving time, which are currently used in the clinic. We then tested the prognostic criteria on data obtained from small number of anonymized CML patients and found the same conclusions hold. Our results suggest that the relative change prognostic criterion more accurately predicts patient response than EMR and the halving time, although more data are needed to confirm this. Our cohort of patients was small due to the variable nature of patient treatment and inconsistent data collection, for example, patients were frequently switched from one TKI to another or one dosage to another (sometimes multiple times), and the patients' *BCR-ABL1* transcript levels were not always consistently recorded. However, we believe that this pilot study demonstrates the feasibility of our approach. Moving forward, we aim to apply our approach to larger datasets and hope to convince others to do the same.

Two strategies can be postulated to overcome TKI resistance. One approach could be to decrease stem cell self-renewal either by inhibiting self-renewal directly (e.g., by augmenting TET2 function using ascorbate; *Agathocleous et al., 2017*; *Cimmino et al., 2017*) or by promoting differentiation (e.g., using retinoids; *Drumea et al., 2008*). By applying combined TKI and pro-differentiation therapy, the self-renewal fractions of the normal and leukemic stem cells can be decreased and brought closer together, which ultimately disadvantages the leukemic cells because of TKI-induced cell death (*Figure 8F*, blue panel). An alternative or complementary approach would be to increase stem cell proliferation via pro-proliferative stimuli such as IFN-alpha (*Essers et al., 2009*) to increase efficacy of TKIs in killing $HSC^L$.

It is apparent that the feedback/feedforward interactions incorporated in our model, which are necessarily somewhat restricted, may be further constrained by spatial characteristics of the BM microenvironment. Nonetheless, our model still displays consistent and biologically relevant behaviors, and although further refinement of the model behaviors is possible, based upon our findings the key behaviors (feedback mechanisms and importance of stem cell self-renewal) would be expected to remain much the same. To explore experimentally observed phenomena not captured by our current model such as treatment-free remission, where a low level of $HSC^L$ persists in the absence of TKI pressure without myeloid cell expansion, improvement of the model is necessary. For example, it may be necessary to incorporate features of the BM microenvironment such as stem cell–niche interactions (*MacLean et al., 2014*; *Lai et al., 2022*) and interactions with immune cells (*Hähnel et al., 2020*). The inclusion of a quiescent stem cell state and additional cellular compartments (such as committed progenitors) coupled with appropriately constrained cell–cell signaling would also make the model more physiologically accurate.

In summary, the feedback/feedforward model we have presented here, while a simplified version of normal and CML hematopoiesis, makes novel and testable predictions regarding the origins of non-genetic primary resistance, which patients will respond to TKI treatment and suggests a combination therapy that can overcome primary resistance. Although preliminary evidence was presented to support model predictions, future work should focus on designing targeted experiments and collecting patient outcomes to generate data to more thoroughly test the model.

## Methods
### Mathematical model of hematopoiesis
The classical depiction of hematopoiesis is a hierarchy of cell types starting with the hematopoietic stem cell at the top, followed by progenitors and ultimately ending with mature cells located in the peripheral blood. Therefore, we model hematopoiesis using a lineage ODE model (*Roeder et al., 2006*; *Komarova and Wodarz, 2007*; *Horn et al., 2008*; *Foo et al., 2009*; *Lander et al., 2009*; *Marciniak-Czochra et al., 2009*; *Manesso et al., 2013*; *Buzi et al., 2015*; *Hähnel et al., 2020*; *Pedersen et al., 2021*) to describe cellular growth dynamics. The modeling allows us to follow the similar hierarchical structure by creating an order of differentiation. Our branched lineage model of

hematopoiesis is simplified and only models HSCs, progenitor cells (MPPs), and two types of terminally differentiated cells (myeloid and lymphoid cells). The model can easily include additional cell types, such as committed progenitor cell types, which will provide an additional level of detail. The model consists of two dividing cell types consisting of S (HSC) and P (MPP) cells with a division rate associated to the cells ( $\eta_1$ and $\eta_2$, respectively). The S cells have the ability to self-renew with fractions (p₀) or differentiate (1 p₀). The P cells have the ability to self-renew with fraction (p₁) or differentiate into either TD$_l$ (lymphoid) or TD$_m$ (myeloid) cells (q₁ or 1-p₁-q₁, respectively). Both S and P cells do not die within the model. The terminal cells form the majority of the hematopoietic system and consist of TD$_l$ and TD$_m$ cells. TD$_m$ and TD$_l$ cells are postmitotic and die at rates d$_m$ and d$_l$, respectively. The following equations (*Equations 1–4*) describe the dynamics of the system:

$$x'_S = (2p_0 - 1)\eta_1 \, x_S$$
$$x'_P = 2(1 - p_0)\eta_1 \, x_S + (2p_1 - 1)\eta_2 \, x_P$$
$$x'_{TD_L} = 2q_1\eta_2 x_P - d_l x_{TD_l}$$
$$x'_{TD_M} = 2\left(1 - p_1 - q_1\right)\eta_2 x_P - d_m x_{TD_m}$$

Further expanded forms of the equations are shown in Appendix 1, *Equations 26–29*, with the addition of feedback regulation for each of the parameters.

## Design space analysis

We use an automated method developed by Savageau and collaborators (*Savageau et al., 2009*; *Fasani and Savageau, 2010*; *Lomnitz and Savageau, 2013*; *Lomnitz and Savageau, 2016*) that separates models by distinct qualitative behaviors at steady state. The strategy is to deconstruct the model of interest at steady state to focus on cases where one production term and one loss term dominate, which gives a dominant subsystem (S-System). This implies that particular inequalities hold in order to ensure the production and loss terms chosen are larger than the others. The inequalities are evaluated at the S-system's steady state to assess self-consistency. If the inequalities are satisfied, the system is self-consistent and the regions where equality holds form boundaries that pertain to a particular qualitative behavior associated with the system. The interior region (where strict inequality holds) is termed a domain in design space. If all the S-systems associated with a model do not have any equilibria that are self-consistent or equilibria that are stable, then the model is rejected. The benefits of this method are that it does not require prior knowledge about parameter values, and it can enumerate the different types of qualitative dynamics a certain system may have. By eliminating subsets of parameters for which the equilibrium is unstable, this approach will automatically select models that are robust to parameter variation due to stability. When we applied this method to the ODE system in *Figure 2F* (Appendix 1, *Equations 26–29*), we found that only the four model classes shown in *Figure 1B* were accepted. See Appendix 1, Section 1, for further details.

## Parameter estimation

To approximate biologically relevant parameters for the model a grid-search algorithm was employed. Parameters were sampled using a random uniform distribution for each parameter (see Appendix 1, Section 3.1). Once parameter values were chosen, the model was simulated to steady state. If a parameter set resulted in steady-state values consistent with the order of magnitude in *Manesso et al., 2013*, the parameter set was accepted, otherwise it was rejected. Specifically, these inequalities had to be satisfied $10^4 <$ HSC < MPPs with MPPs fixed at $10^5$ and MPPs < TD$_l$ < T$_{Dm}$. For $10^6$ iterations, a sample of 1493 parameter sets were accepted. The distribution for these parameter sets is shown in *Appendix 1—figure 4*. To further explore the effect of the feedforward interaction, these parameter sets were reduced to the 563 sets with γ₅ > 0.01. The distribution for these sets is shown in *Appendix 1—figure 5*. The parameter sets used in *Figures 3–7* are provided in *Appendix 1—table 4*.

## Modeling CML development

To model CML development in the presence of normal hematopoietic cells, we introduce a new leukemic cell type for each compartment. Each compartment is then composed of both normal and leukemic subcompartments, which exhibit feedback together as a single compartment. We assume

the only difference between the two cell lineages is the feedback strength for leukemic HSC self-renewal. This small difference gives the leukemic lineage a competitive advantage for growth, consistent with the ability for leukemic HSC to initiate CML (*Reynaud et al., 2011*; *Holyoake and Vetrie, 2017*) and the differential response of the normal and leukemic cells to CCL3 (*Eaves et al., 1993*, *Baba et al., 2013*), which negatively regulates stem cell self-renewal. The full equations used in the model are shown in Appendix 1, *Equations 30–37*.

## Modeling transplant experiments

The model was tested by simulating the transplant experiments (*Figure 4C and D*) of *Reynaud et al., 2011* where $HSC^L$ or $MPP^L$ were implanted into sublethally irradiated mice and terminal cell counts were measured after 35 d. We used two parallel lineages of leukemic cells with identical parameters to mirror the two leukemic cell populations of the experiment. To mimic the effects of sublethal radiation, we reduced the cell populations from their equilibrium values by variable amounts. The $HSC^L$ depletions varied between 50 and 70% and the $MPP^L$ depletions varied between 30 and 50% while both terminal cells were depleted by 10%. After depletion, an additional 4000 cells of either stem or progenitor types were transplanted in accordance with the experiment. We then discarded the 85 parameter sets that were not consistent with the results from *Reynaud et al., 2011*, leaving 478 eligible parameter sets. The results shown in *Figure 4* used depletions of 55% for $HSC^L$, 35% for $MPP^L$s. See Appendix 1, Section 4 and *Figure 5—figure supplements 1–3* for results using other decrements, the discarded parameter sets, and the final parameter distributions.

## Modeling TKI therapy

To account for the treatment by TKIs, additional proliferation-dependent death terms are added to the equations for leukemic stem cells and leukemic progenitor cells shown in Appendix 1, *Equations 38 and 39* (parameter values are given in *Appendix 1—figure 5*). These represent the ability of TKIs to induce cell death in the leukemic cells. Both cell types have unique death rates, to reflect TKIs having different efficacy in killing stem cells and progenitors. The death rates were selected using a single parameter set to ensure a reasonable biphasic curve for *BCR-ABL1* transcript levels compared to patient transcript levels from *Glauche et al., 2018*. The same death rates were then used across every parameter set to ensure consistency. In addition to these changes upon initiation of TKI therapy, the leukemic stem cell division rate is reduced. This reduction models the ability of TKIs to drive leukemic stem cells to quiescence (*Jørgensen et al., 2006*).

To approximate the *BCR-ABL1* transcript levels, we used a method based upon (*Michor et al., 2005*). We use the cell counts of both normal and leukemic terminal cells for both myeloid and lymphoid lineages. The terminal cells are used as in our model they are the closest to peripheral blood in which transcript levels are measured clinically. This results in the following measure for *BCR-ABL1* transcript levels: $\frac{BCR-ABL1}{BCR} = \frac{TD_L^L + TD_M^L}{TD_L^L + TD_M^L + 2(TD_L + TD_M)}$ .

## Modeling combined TKI and differentiation therapy

Combination therapy consists of simultaneously employing TKI therapy, described in 'Methods' and Appendix 1, Sections 2–3, and the addition of a new differentiation therapy. To model differentiation therapy, we altered the form of $p_0$ by including a new constant repressive force that affects both normal and leukemic self-renewal, resulting in $p_{0,new} = \frac{p_{0,max}}{1+\gamma_1 P+\Delta}$ and $p_{0,new}^L = \frac{p_{0,max}^L}{1+\gamma_1^L P+\Delta}$, where $P = x_{P^L} + x_P$ and $\Delta$ is the differentiation therapy strength. We then performed combination therapy using our existing parameters, swept through differentiation therapy strengths and recorded which parameter sets achieved MR3 within 50 mo. We then determined that a differentiation therapy strength of D = 0.24 resulted in the highest proportion of parameter sets that achieved MR3 response. The full equations for combined TKI and differentiation therapy are shown in Appendix 1, *Equations 40 and 41*.

## CML patient data

Data from newly diagnosed CML patients (n = 21) treated with TKI therapy at UCI Health were obtained under an honest broker mechanism from the electronic health record under Exemption 4 for human subjects research.

## Mice

C57BL/6J female mice (Jackson Laboratories), 6–12 wk of age were used for irradiation and myeloid depletion experiments. Conditional *BCR-ABL1* double transgenic mice (*Koschmieder et al., 2005*) were obtained from Dr. Emmanuel Passegue (Columbia University). All protocols in mice were approved by the Institutional Animal Use and Care Committee of University of California, Irvine.

## Irradiation of mice

To achieve selective depletion of HSCs, a 50 cGy dose of irradiation from X-ray source (Precision X-rad 320) was applied. Control mice did not receive irradiation. The distribution of time points at which observations were made (days 1, 3, and 7 post-irradiation), and the number of mouse replicates to use at each time point (between 2 and 7, totaling 13 mice), were informed by our Bayesian hierarchical framework for optimal experimental design (*Lomeli et al., 2021*).

## Myeloid cell depletion

RB6-8C5, an anti-Gr1 antibody (catalog # BE0075, BioXCell) or isotype control (catalog # BE0090, BioXCell) was injected intravenously, 50 µg per mouse, and mice sacrificed 24 hr later.

## BrdU injections

In irradiation experiments, mice were pulsed with BrdU by IP injection of 200 µl of 10 mg/ml BrdU in DPBS. BrdU flow kit (552598) from BD Biosciences was used for detection of BrdU labeling in hematopoietic cells by flow cytometry.

## Flow cytometry analysis of cell populations

BM cells from femur and tibia of control and dosed mice were isolated by flushing bones. Following lysis of red blood cells (RBC lysis buffer, eBiosciences), leukocytes were stained with CD34 antibody for 1 hr and subsequently incubated with a cocktail of biotinylated antibodies directed against lineage markers (CD3, Gr-1, B220, Ter119) and stem/progenitor markers (c-Kit, Sca-1, CD48) for 30 min. Streptavidin (SA)-conjugated fluorochrome was utilized to detect biotinylated antibodies. Following fixation, permeabilization, and DNase digestion, anti-BrdU antibody was used to assess BrdU incorporation. Events were acquired on FACS Arial II and analyzed with Flowjo v.10 software.

## Antibodies

Monoclonal antibodies for flow cytometry were biotinylated mouse lineage panel (559971, BD Biosciences), PE-CF594 Streptavidin (562318, BD Biosciences), anti-mouse CD48 (561242, BD Biosciences), anti-mouse CD34 eFluor450 (48-0341-82, eBiosciences), anti-mouse Sca-1-PE (108108, BioLegend), anti-mouse c-Kit-APC (17-1171-82, eBiosciences), and FITC BrdU flow kit (559619, BD Biosciences).

## Generation and TKI treatment of chimeric *BCR-ABL1* mice

The full details of the CML mouse model will be published elsewhere (Jena et al., in preparation). Briefly, BM cells from conditional *BCR-ABL1* double transgenic mice (CD45.2$^+$) (Koschmieder, Gottgens et al. 2005) (40 million cells) were transplanted intravenously into unirradiated C57BL/6J recipients (CD45.1$^+$CD45.2$^+$) maintained on doxycycline to suppress *BCR-ABL1* expression. Two months post-transplant, doxycycline was removed to allow induction of CML-like leukemia. Chimerism was assessed by percentage of CD45.1$^-$ CD45.2$^+$ granulocytes in peripheral blood. To generate chimeric mice with high (>90%) leukemic stem cell burden, the donor and recipient pair was reversed, with double transgenic mice transplanted with normal B6 BM. In mice with established CML-like leukemia (peripheral blood leukocytes > 20,000/µl and >40% circulating granulocytes), TKI treatment was initiated with dasatinib (25 mg/kg daily by oral gavage).

## Data availability

Code used to generate the figures, determine the S-Systems and files containing parameter sets are found at the following GitHub: https://github.com/jonatdr/CML_Treatment (copy archived at *Jonatdr, 2023*).

## Acknowledgements

The authors thank the reviewers for their careful analysis of the manuscript and their suggestions, which significantly improved it. ADL, JSL, and RAV acknowledge the National Institutes of Health for partial support through grant nos. 1U54CA217378-01A1 for a National Center in Cancer Systems Biology at UC Irvine and P30CA062203 for the Chao Family Comprehensive Cancer Center at UC Irvine. In addition, ADL and JSL acknowledge support from DMS-1763272 and the Simons Foundation (594598QN) for an NSF-Simons Center for Multiscale Cell Fate Research, RAV acknowledges support from R01 CA090576. JSL also acknowledges NSF grants DMS-1936833 and DMS-1953410. JR and AI each acknowledge support from NSF graduate research fellowships.

## Additional information

### Funding

| Funder | Grant reference number | Author |
| --- | --- | --- |
| National Institutes of Health | 1U54CA217378-01A1 | Arthur D Lander<br>Richard A Van Etten<br>John Lowengrub |
| National Institutes of Health | P30CA062203 | Arthur D Lander |
| National Institutes of Health | R01 CA090576 | Richard A Van Etten |
| National Science Foundation | DMS-1763272 | Arthur D Lander<br>John Lowengrub |
| National Science Foundation | DMS-1936833 | John Lowengrub |
| National Science Foundation | DMS-1953410 | John Lowengrub |
| National Science Foundation | GRFP 16-588 | Abdon Iniguez |
| Simons Foundation | 594598QN | Arthur D Lander<br>John Lowengrub |
| National Institute of General Medical Sciences | GM136624 | Jonathan Rodriguez |

The funders had no role in study design, data collection and interpretation, or the decision to submit the work for publication.

### Author contributions

Jonathan Rodriguez, Conceptualization, Software, Formal analysis, Validation, Investigation, Visualization, Methodology, Writing - original draft, Writing – review and editing; Abdon Iniguez, Conceptualization, Data curation, Software, Investigation, Methodology, Writing - original draft; Nilamani Jena, Data curation, Investigation, Methodology; Prasanthi Tata, Data curation, Investigation, Visualization, Methodology; Zhong-Ying Liu, Resources, Supervision, Project administration; Arthur D Lander, Conceptualization, Supervision, Funding acquisition, Project administration; John Lowengrub, Conceptualization, Supervision, Funding acquisition, Investigation, Project administration, Writing – review and editing; Richard A Van Etten, Conceptualization, Supervision, Funding acquisition, Investigation, Methodology, Project administration, Writing – review and editing

### Author ORCIDs

Jonathan Rodriguez ⬩ http://orcid.org/0000-0002-6414-0526
Arthur D Lander ⬩ http://orcid.org/0000-0002-4380-5525
John Lowengrub ⬩ http://orcid.org/0000-0003-1759-0900
Richard A Van Etten ⬩ http://orcid.org/0000-0003-0000-635X

### Ethics

This study was performed in strict accordance with the recommendations in the Guide for the Care and Use of Laboratory Animals of the National Institutes of Health. All of the animals were handled according to approved institutional animal care and use committee (IACUC) protocol (AUP-19-159) of the University of California, Irvine.

### Decision letter and Author response

Decision letter https://doi.org/10.7554/eLife.84149.sa1
Author response https://doi.org/10.7554/eLife.84149.sa2

---

## Additional files

### Supplementary files
• MDAR checklist

### Data availability

Modelling code and parameter set data are available in a Github repository. Patient data is unavailable publicly as it could be used to potentially identify the patients. Deidentified raw patient transcript data will be made available to qualified researchers (academic or industry) upon request to Dr. Van Etten at vanetten@hs.uci.edu.

The following dataset was generated:

| Author(s) | Year | Dataset title | Dataset URL | Database and Identifier |
|---|---|---|---|---|
| Rodriguez J, Iniguez A | 2023 | Grid-search | https://github.com/jonatdr/CML_Treatment/tree/main/Grid-search | GItHub, Grid-search |

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

## Appendix 1

### 1 Design space analysis

#### 1.1 Introduction

In this section, we provide details on the application of design space analysis (DSA), developed in *Savageau et al., 2009*; *Fasani and Savageau, 2010*; *Lomnitz and Savageau, 2013*; *Lomnitz and Savageau, 2016* for chemical reaction networks, to a simplified version of the cell lineage model considered in the main text (see *Figure 1*). This enables us to analyze nonlinear dynamical systems near steady state to identify the regions in parameter space where common qualitative behaviors occur. Applying this analysis allows us to (1) ignore specific parameter values, (2) obtain analytical steady states, (3) reduce the search area in parameter space by searching boundaries separating regions of common behaviors, and (4) easily automate this process. In this section, we review how one can construct a design space given an ODE model. We then apply this analysis to an ODE model of cell lineages and show how this model can be used to define regions of stability in parameter space.

#### 1.2 Boundaries of design space for general models

In order to apply DSA, the ODE must be a generalized mass action system, shown below in *Equations 1 and 2*,

$$\frac{dX_i}{dt} = \sum_{k=1}^{r} \alpha_{ik} \prod_{j=1}^{n+m} X_j^{g_{ijk}} - \sum_{k=1}^{r} \beta_{ik} \prod_{j=1}^{n+m} X_j^{h_{ijk}}, \tag{1}$$

$$X_i(0) = X_{i0}, \tag{2}$$

for $i = 1, n$. Here, $n$ corresponds to the number of dependent variables and $m$ corresponds to the number of independent variables. The $\alpha_{ik}$ and $\beta_{ik}$ parameters correspond to rate constants of the differential equation and $r$ corresponds to the number of associated rate constants.

DSA takes advantage of the above form by creating a system of deconstructed ODEs where one source term and one sink term in the differential equation dominate, known as a subsystem (S-system). The following is the generalized form of an S-system:

$$\frac{dX_i}{dt} = \alpha_{ip} \prod_{j=1}^{n+m} X_j^{g_{ijp}} - \beta_{iq} \prod_{j=1}^{n+m} X_j^{h_{ijq}} \tag{3}$$

$$X_i(0) = X_{i0} \tag{4}$$

where $p$ and $q$ correspond to the number of positive and negative terms of the differential equation, respectively. We are interested in solving these solutions at steady state, and therefore solve the system by setting the time derivative to zero. We take advantage of the form shown in *Equation 3* and take the log of the system:

$$\log(\alpha_{ip}) + \sum_{j=1}^{n+m} g_{ijp} \log(X_j) = \log(\beta_{iq}) + \sum_{j=1}^{n+m} h_{ijq} \log(X_j) \tag{5}$$

thus, making this a linear solve in log space. In defining the S-systems, we must make assumptions about the model and its parameters. To satisfy the S-systems, we impose inequality constraints to satisfy the dominating source and sink terms of the S-systems by the following:

$$\alpha_{ip} \prod_{j=1}^{n+m} X_j^{g_{ijp}} > \alpha_{i\bar{p}} \prod_{j=1}^{n+m} X_j^{g_{ij\bar{p}}} \text{ for i = 1,..,n;} \bar{p}\text{=1,...,p-1,p+1,...,r} \tag{6}$$

$$\beta_{iq} \prod_{j=1}^{n+m} X_j^{h_{ijq}} > \beta_{i\bar{q}} \prod_{j=1}^{n+m} X_j^{h_{ij\bar{q}}} \text{ for i = 1,..,n;} \bar{q}\text{=1,...,q-1,q+1,...,r} \tag{7}$$

We can then log transform these inequalities to obtain

$$\log\left(\alpha_{ip}\right) + \sum_{j=1}^{n+m} g_{ijp} \log\left(X_j\right) > \log\left(\alpha_{i\bar{p}}\right) + \sum_{j=1}^{n+m} g_{ij\bar{p}} \log\left(X_j\right) \tag{8}$$

$$\log\left(\beta_{iq}\right) + \sum_{j=1}^{n+m} h_{ijq} \log\left(X_j\right) > \log\left(\beta_{i\bar{q}}\right) + \sum_{j=1}^{n+m} h_{ij\bar{q}} \log\left(X_j\right) \tag{9}$$

These inequalities become our boundaries in parameter space in which qualitative behavior is shared once the $X_j$'s are evaluated at steady state, where the steady states are obtained from solving the log-linear S-system. Not all S-systems will have a unique solution or satisfy the inequality constraints. The S-systems that do not satisfy these constraints will be discarded. The analysis is summarized in *Appendix 1—figure 1*.

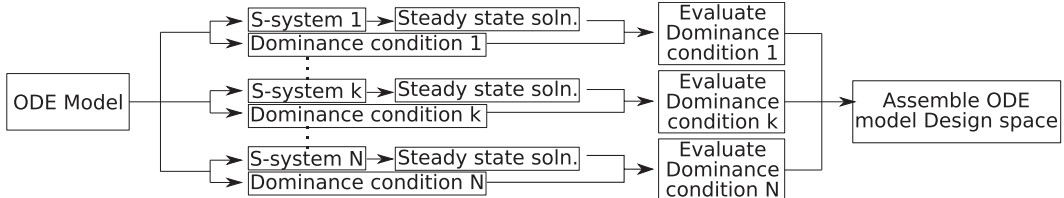

**Appendix 1—figure 1.** Flow chart for design space analysis (DSA). Given an ordinary differential equation (ODE), we can obtain the design space by obtaining all S-systems, steady states, and evaluated dominance conditions. S-systems that do not satisfy the dominance condition are not included in the design space.

## 1.3 Analysis of a four-cell lineage model

We next apply this analysis to a lineage model with four cell types (see *Appendix 1—figure 2*). The model consists of two dividing cell types consisting of $S$ (HSC) and $P$ (MPP) cells with a division rate associated to the cells ($\eta_1$ and $\eta_2$, respectively). The $S$ cells have the ability to self-renew with fraction ($p_0$) or differentiate ($1 - p_0$). The $P$ cells have the ability to self-renew with fraction ($p_1$) or differentiate into either $TD_l$ (lymphoid) or $TD_m$ (myeloid) cells ($q_1$ or $1 - p_1 - q_1$, respectively). $TD_m$ and $TD_l$ cells only have the ability to die at rates $d_m$ and $d_l$, respectively. We add negative feedback on the self-renewal fraction of the stem cells from the differentiated cells. *Appendix 1—figure 2* shows a schematic of the lineage with the parameters. The corresponding differential equations are

$$x_S' = \left(2p_0 - 1\right) \eta_1 x_S \tag{10}$$

$$x_P' = 2\left(1 - p_0\right) \eta_1 x_S + \left(2p_1 - 1\right) \eta_2 x_P \tag{11}$$

$$x_{TD_L}' = 2q_1 \eta_2 x_P - d_l x_{TD_l} \tag{12}$$

$$x_{TD_M}' = 2\left(1 - p_1 - q_1\right) \eta_2 x_P - d_m x_{TD_m} \tag{13}$$

where $p_0 = \bar{p}_0 / \left(1 + \gamma x_P\right)$, and $\bar{p}_0$ is defined as the maximum stem cell self-renewal fraction and $\gamma$ is the feedback strength.

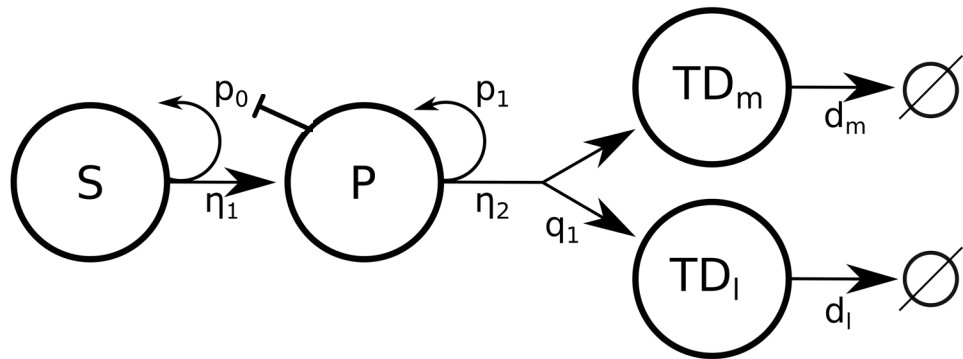

**Appendix 1—figure 2.** Lineage schematic depicting a stem cell and terminally differentiated cell with negative feedback onto stem cell self-renewal probability $p_0$.

We begin by rewriting the equations in the form shown in *Equation 1*.

$$x'_S = \left(2\bar{p}_0 x_{new}^{-1} - 1\right)\eta_1 x_S \tag{14}$$

$$x'_P = 2\left(1 - \bar{p}_0 x_{new}^{-1}\right)\eta_1 x_S + (2p_1 - 1)\eta_2 x_P \tag{15}$$

$$x'_{TD_L} = 2q_1\eta_2 x_P - d_l x_{TD_l} \tag{16}$$

$$x'_{TD_M} = 2\left(1 - p_1 - q_1\right)\eta_2 x_P - d_m x_{TD_m} \tag{17}$$

$$0 = 1 + \gamma x_P - x_{new}. \tag{18}$$

Note that we introduce a new variable $x_{new} = 1 + \gamma x_P$, which is necessary to achieve the form of *Equation 1*. We next find all combinations in which one source term and one sink term dominates the system of differential equations.

From *Equations 14–18*, we obtain 24 S-system combinations, a subset of which is shown in *Appendix 1—table 1*. We continue the analysis with S-system 2 from *Appendix 1—table 1*. Using S-system 2, we set the time derivatives to zero and rearrange the equations such that we obtain $A\bar{x} = \bar{b}$:

$$
\begin{bmatrix}
0 & 0 & 0 & 0 & 1 \\
1 & -1 & 0 & 0 & 0 \\
0 & 1 & -1 & 0 & 0 \\
0 & 1 & 0 & -1 & 0 \\
0 & -1 & 0 & 0 & 1
\end{bmatrix}
\begin{bmatrix}
\log\left(\bar{x}_S\right) \\
\log\left(\bar{x}_P\right) \\
\log\left(\bar{x}_{TD_l}\right) \\
\log\left(\bar{x}_{TD_m}\right) \\
\log\left(\bar{x}_{new}\right)
\end{bmatrix}
=
\begin{bmatrix}
\log\left(2\bar{p}_0\right) \\
\log\left(\frac{\eta_2}{2\eta_1}\right) \\
\log\left(\frac{d_l}{2q_1\eta_2}\right) \\
\log\left(\frac{d_m}{2\eta_2}\right) \\
\log\left(\gamma\right)
\end{bmatrix}
$$

such that $\bar{x}_S$, $\bar{x}_P$, $\bar{x}_{TD_l}$, and $\bar{x}_{TD_m}$ are the steady-state solutions for the S-system and $\bar{x}_{new}$ is the solution of the newly defined variable at steady state. Solving the linear equation gives us

$$
\begin{bmatrix}
\log\left(\bar{x}_S\right) \\
\log\left(\bar{x}_P\right) \\
\log\left(\bar{x}_{TD_l}\right) \\
\log\left(\bar{x}_{TD_m}\right) \\
\log\left(\bar{x}_{new}\right)
\end{bmatrix}
=
\begin{bmatrix}
\log\left(\frac{\bar{p}_0\eta_2}{\gamma}\right) \\
\log\left(\frac{2\bar{p}_0}{\gamma}\right) \\
\log\left(\frac{4\bar{p}_0 q_1\eta_2}{d_l}\right) \\
\log\left(\frac{4\bar{p}_0\eta_2}{d_m}\right) \\
\log\left(2\bar{p}_0\right)
\end{bmatrix}.
$$

We can now construct the boundaries for this S-system. We substitute our steady-state solutions obtained above into the logged inequality constraints from *Appendix 1—table 1*. Thus, the inequalities in log space become

$$0 < \log\left(2\bar{p}_0\right) \tag{19}$$

$$0 > \log\left(2\bar{p}_1\right) \tag{20}$$

$$0 > \log\left(\bar{q}_1\right) \tag{21}$$

Out of the 24 possible S-systems, only S-system 2 has a unique steady state that satisfies the constraints. We can plot the design space by varying $\bar{p}_0$ and $\gamma$ (see *Appendix 1—figure 3*). The design space shows one region where $\bar{p}_0 > 0.5$, a requirement for a positive steady state in the full system. The domain in parameter space corresponds to S-system 2 in *Appendix 1—table 1*.

**Appendix 1—table 1.** A sample of S-systems from *Equations 14–18*.

| | S-system 1 | S-system 2 | S-system 3 | S-system 4 |
|---|---|---|---|---|
| $x_S'$ | $2\bar{p}_0 x_{new}^{-1}\eta_1 x_S - \eta_1 x_S$ | $2\bar{p}_0 x_{new}^{-1}\eta_1 x_S - \eta_1 x_S$ | $2\bar{p}_0 x_{new}^{-1}\eta_1 x_S - \eta_1 x_S$ | $2\bar{p}_0 x_{new}^{-1}\eta_1 x_S - \eta_1 x_S$ |
| $x_P'$ | $2\eta_1 x_S - 2\bar{p}_0 x_{new}^{-1}\eta_1 x_S$ | $2\eta_1 x_S - \eta_2 x_S$ | $2p_1\eta_2 x_S - 2\bar{p}_0 x_{new}^{-1}\eta_1 x_S$ | $2p_1\eta_2 x_S - \eta_2 x_S$ |
| $x_{TD_l}'$ | $2q_1\eta_2 x_P - d_l x_{TD_l}$ | $2q_1\eta_2 x_P - d_l x_{TD_l}$ | $2q_1\eta_2 x_P - d_l x_{TD_l}$ | $2q_1\eta_2 x_P - d_l x_{TD_l}$ |
| $x_{TD_m}'$ | $2\eta_2 x_P - d_m x_{TD_m}$ | $2\eta_2 x_P - d_m x_{TD_m}$ | $2\eta_2 x_P - d_m x_{TD_m}$ | $2\eta_2 x_P - d_m x_{TD_m}$ |
| 0 | $\gamma x_P - x_{new}$ | $\gamma x_P - x_{new}$ | $\gamma x_P - x_{new}$ | $\gamma x_P - x_{new}$ |
| Boundary 1 | $2\eta_1 x_S > 2p_1\eta_2 x_S$ | $2\eta_1 x_S > 2p_1\eta_2 x_S$ | $2\eta_1 x_S < 2p_1\eta_2 x_S$ | $2\eta_1 x_S < 2p_1\eta_2 x_S$ |
| Boundary 2 | $2\bar{p}_0 x_{new}^{-1}\eta_1 x_S > \eta_2 x_S$ | $2\bar{p}_0 x_{new}^{-1}\eta_1 x_S < \eta_2 x_S$ | $2\bar{p}_0 x_{new}^{-1}\eta_1 x_S > \eta_2 x_S$ | $2\bar{p}_0 x_{new}^{-1}\eta_1 x_S < \eta_2 x_S$ |

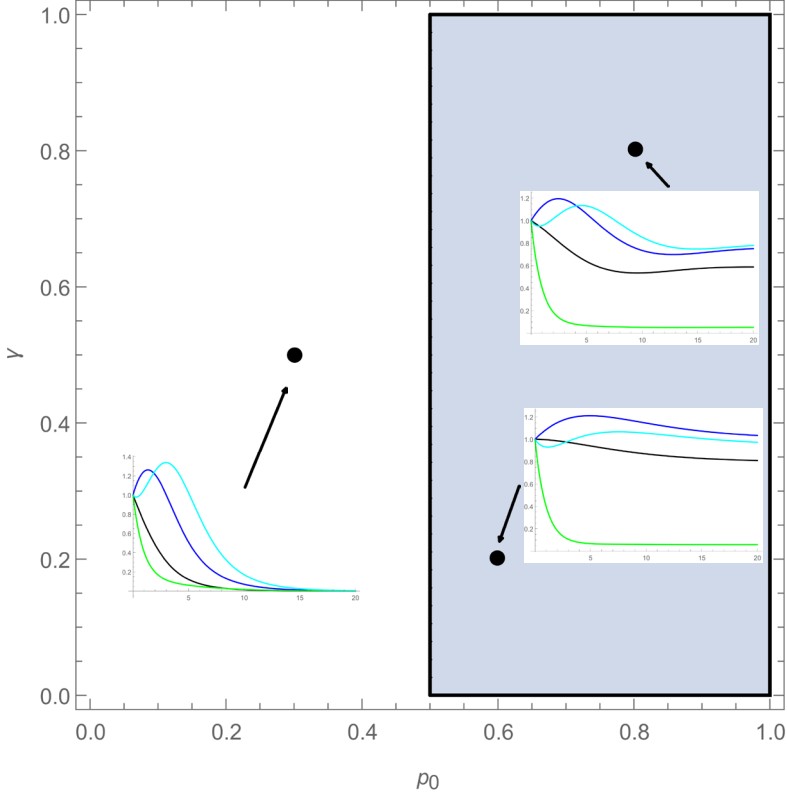

**Appendix 1—figure 3.** Design space for four-cell lineage model. The design space is showing a slice of parameter space varying the max self-renewal probability ($\bar{p}_0$) and feedback gain for stem cells ($\gamma$). If we are to sample parameters sets (shown by points), we observe oscillatory behavior in the full system, as shown by the time evolution plots.

It is possible to relate the S-system back to the true ODE system with classical techniques. For example, it was shown in *Savageau et al., 2009*; *Fasani and Savageau, 2010*; *Lomnitz and Savageau, 2013*; *Lomnitz and Savageau, 2016* that the S-system and full ODE system have the same linear stability behavior in the parameter regime appropriate for the S-system. Thus, parameter sensitivity analyses of the S-system provide insight on the behavior of the full system.

Performing a linear stability analysis on the S-system that corresponds to the region in parameter space shown in *Appendix 1—figure 3*, we obtain the following eigenvalues:

$$\lambda = \left[ -0.5d - \frac{0.125\gamma\sqrt{(d(-64+16d)\bar{p}_0{}^4)/\gamma^2}}{\bar{p}_0{}^2} \right., \tag{22}$$

$$\left. -0.5d + \frac{0.125\gamma\sqrt{(d(-64+16d)\bar{p}_0{}^4)/\gamma^2}}{\bar{p}_0{}^2} \right] \tag{23}$$

which suggests a stable spiral or stable node depending on the value of $d$. We also perform the linear stability analysis on the full system and obtain the following eigenvalues:

$$\lambda = \left[ \frac{-0.25d\bar{p}_0 - 0.125\sqrt{d\bar{p}_0{}^2(4d + (32 - 64p_0)\bar{p}_0)}}{\bar{p}_0{}^2} \right., \tag{24}$$

$$\left. \frac{-0.25d\bar{p}_0 + 0.125\sqrt{d\bar{p}_0{}^2(4d + (32 - 64p_0)\bar{p}_0)}}{\bar{p}_0{}^2} \right] \tag{25}$$

and in the parameter space for the S-system, we obtain the same behavior. Using the full system, we can also plot the dynamics in the domain, which are shown as insets in *Appendix 1—figure 3*. When plotting the full system in *Appendix 1—figure 3*, we observe that the stem and progenitor populations oscillate before reaching steady state, characteristics of a stable spiral. The S-system and the true system eigenvalues are not identical, but using the same parameters in the appropriate S-system parameter space, the systems will have the same behavior. However, outside of the domain of validity of the S-system, we cannot conclude anything about the full system's dynamics from the S-system. In fact, from *Appendix 1—figure 3*, we observe being outside of the domain of validity of S-system 2 (e.g., $\bar{p}_0 < 0.5$) yields to solutions that tend to the zero steady state.

We use DSA to select among all possible ODE models for normal hematopoiesis consistent with the lineage diagram shown in *Figure 1A* in the main text in which each parameter $p_0$, $p_1$, $q_1$, $\eta_1$, and $\eta_2$ is either unregulated (constant) or is subject to positive or negative regulation from most one cell type in the lineage. As described in the main text, there are 59,049 possible models, counting each combination as a model. We implement an automated implementation of DSA that enables an efficient exploration of this large space of models. Eliminating models with no valid S-systems and those with unstable equilibria, we eliminate all but the four model classes shown in *Figure 1B* in the main text.

## 2 Mathematical model

The complete ODE model of normal hematopoiesis is composed of the following equations:

$$x'_S = \left(2\frac{p_{0,max}}{1+\gamma_1 x_P} - 1\right)\frac{\eta_{1,max}}{1+\gamma_2 x_S}x_S \tag{26}$$

$$x'_P = 2\left(1 - \frac{p_{0,max}}{1+\gamma_1 x_P}\right)\frac{\eta_{1,max}}{1+\gamma_2 x_S}x_S + \left(2\frac{p_{1,max}}{1+\gamma_3 x_{TD_m}} - 1\right)\frac{\eta_{2,max}}{1+\gamma_5 x_S}x_P \tag{27}$$

$$x'_{TD_L} = 2\frac{q_{1,max}}{1+\gamma_4 x_{TD_m}}\frac{\eta_{2,max}}{1+\gamma_5 x_S}x_P - d_L x_{TD_L} \tag{28}$$

$$x'_{TD_m} = 2\left(1 - \frac{p_{1,max}}{1+\gamma_3 x_{TD_m}} - \frac{q_{1,max}}{1+\gamma_4 x_{TD_m}}\right)\frac{\eta_{2,max}}{1+\gamma_5 x_S}x_P - d_m x_{TD_m}. \tag{29}$$

The ODE model for CML hematopoiesis tracks the dynamics of both the normal and CML cells (superscript $L$), and assumes that both cell types provide and respond to feedback signaling, although the CML stem cells are slightly less responsive to negative feedback regulation, which gives them a fitness advantage. The complete system is given by

$$x'_S = \left(2\frac{p_{0,max}}{1+\gamma_1\left(x_P + x_{P^L}\right)} - 1\right)\frac{\eta_{1,max}}{1+\gamma_2\left(x_S + x_{S^L}\right)}x_S \tag{30}$$

$$x'_P = 2\left(1 - \frac{p_{0,max}}{1+\gamma_1\left(x_P + x_{P^L}\right)}\right)\frac{\eta_{1,max}}{1+\gamma_2\left(x_S + x_{S^L}\right)}x_S + \left(2\frac{p_{1,max}}{1+\gamma_3\left(x_{TD_m} + x_{TD_m^L}\right)} - 1\right)\frac{\eta_{2,max}}{1+\gamma_5\left(x_S + x_{S^L}\right)}x. \tag{31}$$

$$x'_{TD_L} = 2\frac{q_{1,max}}{1+\gamma_4\left(x_{TD_m} + x_{TD_m^L}\right)}\frac{\eta_{2,max}}{1+\gamma_5\left(x_S + x_{S^L}\right)}x_P - d_L x_{TD_L} \tag{32}$$

$$x'_{TD_m} = 2\left(1 - \frac{p_{1,max}}{1+\gamma_3\left(x_{TD_m} + x_{TD_m^L}\right)} - \frac{q_{1,max}}{1+\gamma_4\left(x_{TD_m} + x_{TD_m^L}\right)}\right)\frac{\eta_{2,max}}{1+\gamma_5\left(x_S + x_{S^L}\right)}x_P - d_m x_{TD_m} \tag{33}$$

$$x'_{S^L} = \left( 2 \frac{p^L_{0,max}}{1 + \gamma^L_1 \left( x_P + x_{P^L} \right)} - 1 \right) \frac{\eta^L_{1,max}}{1 + \gamma^L_2 \left( x_S + x_{S^L} \right)} x_{S^L} \tag{34}$$

$$x'_{P^L} = 2 \left( 1 - \frac{p^L_{0,max}}{1 + \gamma^L_1 \left( x_P + x_{P^L} \right)} \right) \frac{\eta^L_{1,max}}{1 + \gamma^L_2 \left( x_S + x_{S^L} \right)} x_{S^L} + \left( 2 \frac{p^L_{1,max}}{1 + \gamma^L_3 \left( x_{TD_m} + x_{TD^L_m} \right)} - 1 \right) \frac{\eta^L_{2,max}}{1 + \gamma^L_5 \gamma_5 \left( x_S + \jmath \right.} \tag{35}$$

$$x'_{TD^L_L} = 2 \frac{q^L_{1,max}}{1 + \gamma^L_4 \left( x_{TD_m} + x_{TD^L_m} \right)} \frac{\eta^L_{2,max}}{1 + \gamma^L_5 \left( x_S + x_{S^L} \right)} x_{P^L} - d^L_l x_{TD^L_L} \tag{36}$$

$$x'_{TD^L_m} = 2 \left( 1 - \frac{p^L_{1,max}}{1 + \gamma^L_3 \left( x_{TD_m} + x_{TD^L_m} \right)} - \frac{q^L_{1,max}}{1 + \gamma^L_4 \left( x_{TD_m} + x_{TD^L_m} \right)} \right) \frac{\eta^L_{2,max}}{1 + \gamma^L_5 \left( x_S + x_{S^L} \right)} x_{P^L} - d^L_m x_{TD^L_m}. \tag{37}$$

When TKI therapy is applied, *Equations 34 and 35* are replaced with

$$x'_{S^L} = \left( 2 \frac{p^L_{0,max}}{1 + \gamma^L_1 \left( x_P + x_{P^L} \right)} - 1 - TKI_{HSC} \right) \frac{\eta^L_{1,max}}{1 + \gamma^L_2 \left( x_S + x_{S^L} \right)} x_{S^L} \tag{38}$$

$$x'_{P^L} = 2 \left( 1 - \frac{p^L_{0,max}}{1 + \gamma^L_1 \left( x_P + x_{P^L} \right)} \right) \frac{\eta^L_{1,max}}{1 + \gamma^L_2 \left( x_S + x_{S^L} \right)} x_{S^L} + \left( 2 \frac{p^L_{1,max}}{1 + \gamma^L_3 \left( x_{TD_m} + x_{TD^L_m} \right)} - 1 - TKI_{MPP} \right) \frac{}{1 + \gamma} \tag{39}$$

where $TKI_{HSC}$ and $TKI_{MPP}$ denote TKI-induced death rates of the CML stem and MPP cells. With the introduction of differentiation therapy, *Equations 30 and 38* instead become

$$x'_S = \left( 2 \frac{p_{0,max}}{1 + \gamma_1 \left( x_P + x_{P^L} \right) + \Delta} - 1 \right) \frac{\eta_{1,max}}{1 + \gamma_2 \left( x_S + x_{S^L} \right)} x_S \tag{40}$$

$$x'_{S^L} = \left( 2 \frac{p^L_{0,max}}{1 + \gamma^L_1 \left( x_P + x_{P^L} \right) + \Delta} - 1 - TKI_{HSC} \right) \frac{\eta^L_{1,max}}{1 + \gamma^L_2 \left( x_S + x_{S^L} \right)} x_{S^L}. \tag{41}$$

## 3 Parameter estimation

The parameters for the model of normal hematopoiesis, and their descriptions, are listed in *Appendix 1—table 2*. The additional parameters needed to model the CML cell population, and the application of TKI therapy are given in *Appendix 1—table 3*.

**Appendix 1—table 2.** Parameter values used to model the normal hematopoietic system.

| Parameter | Description |
| --- | --- |
| $p_{0,max}$ | Maximal self-renewal fraction of HSC |
| $p_{1,max}$ | Maximal self-renewal fraction of MPP |
| $q_{1,max}$ | Maximum branching fraction from MPP to $TD_L$ |

*Appendix 1—table 2 Continued on next page*

*Appendix 1—table 2 Continued*

| Parameter | Description |
|---|---|
| $\eta_{1,max}$ | Maximum HSC proliferation rate |
| $\eta_{2,max}$ | Maximum MPP proliferation rate |
| $\gamma_1$ | Feedback gain on HSC self-renewal fraction |
| $\gamma_2$ | Feedback gain on HSC proliferation rate (from HSC) |
| $\gamma_3$ | Feedback gain on MPP self-renewal fraction |
| $\gamma_4$ | Feedback gain on MPP branching fraction |
| $\gamma_5$ | Feedforward gain on MPP proliferation rate |
| $d_L$ | Death rate of $TD_L$ |
| $d_m$ | Death rate of $TD_M$ |

HSC: hematopoietic stem cell; MPP: multipotential progenitor.

**Appendix 1—table 3.** Additional parameter values used to model the chronic myeloid leukemia (CML) cell population dynamics and effect of tyrosine kinase inhibitor (TKI) therapy.

| Parameter | Description |
|---|---|
| $p_{0,max}^L$ | Maximal self-renewal fraction of HSC L |
| $p_{1,max}^L$ | Maximal self-renewal fraction of MPP L |
| $q_{1,max}^L$ | Maximum branching fraction from MPP L to $TD_L^L$ |
| $\eta_{1,max}^L$ | Maximum HSC L proliferation rate |
| $\eta_{2,max}^L$ | Maximum MPP L proliferation rate |
| $\gamma_1^L$ | Feedback gain on HSC L self-renewal fraction |
| $\gamma_2^L$ | Feedback gain on HSC L proliferation rate (from HSC L) |
| $\gamma_3^L$ | Feedback gain on MPP L self-renewal fraction |
| $\gamma_4^L$ | Feedback gain on MPP L branching fraction |
| $\gamma_5^L$ | Feedforward gain on MPP L proliferation rate |
| $d_L^L$ | Death rate of $TD_L^L$ |
| $d_m^L$ | Death rate of $TD_M^L$ |
| $TKI_{HSC}$ | Death rate of HSC L due to TKI therapy |
| $TKI_{MPP}$ | Death rate of MPP L due to TKI therapy |

HSC: hematopoietic stem cell; MPP: multipotential progenitor.

## 3.1 Parameter distributions

A grid-search algorithm was used to find parameter values that demonstrate steady-state cell counts consistent with *Manesso et al., 2013* and time of recovery to steady state from cellular perturbations consistent with *Reynaud et al., 2011*. The resulting distributions of the 1493 parameters are displayed along the diagonal of *Appendix 1—figure 4*. In the distributions, the feedback gain of the feedforward regulation is skewed toward smaller values across all parameter sets. To investigate the observed feedforward loop, we selected only the parameter sets with sufficiently large feedforward gain $\gamma_5 > 0.01$. This resulted in a reduced distribution of 563 parameter sets and is shown in *Appendix 1—figure 5*. The parameter ranges for the gridsearch were found using the method shown in the following pseudo-Python.

```
for i in range(1e6):

param_gridsearch_dist = dict(p0max = np.random.uniform(0.5, 1.0),
p1max = np.random.uniform(0.0,0.5), q1max = np.abs(np.random.uniform(0.0,0.
49)),
eta1max = np.random.uniform(0,0.5), eta2max = 10**np.random.uniform(-
2,1.5),gam2=10**np.random.uniform(-6,0), gam3=10**np.random.uniform(-6,0),
gam4=10**np.random.uniform(-6,0), gam5=10**np.random.uniform(-6,0),dL =
10**np.random.uniform(-4,1), dM = 10**np.random.uniform(-4,1))
y=hematopoiesis(param_gridsearch_dist)

if y[0,-1]>.01 and y[0,-1]<y[1,-1]<y[2,-1]<y[3,-1]:master_params.append(
param_gridsearch_dist)
```

The specific parameters used to model the normal hematopoietic system in *Figures 3–7A, C and D* in the main text are given in *Appendix 1—table 4*. When CML cells are introduced, the additional parameters of a representative responder associated with the leukemic cell ODEs are given in *Appendix 1—table 5*. A representative parameter set for a nonresponder is shown in *Appendix 1—table 6*.

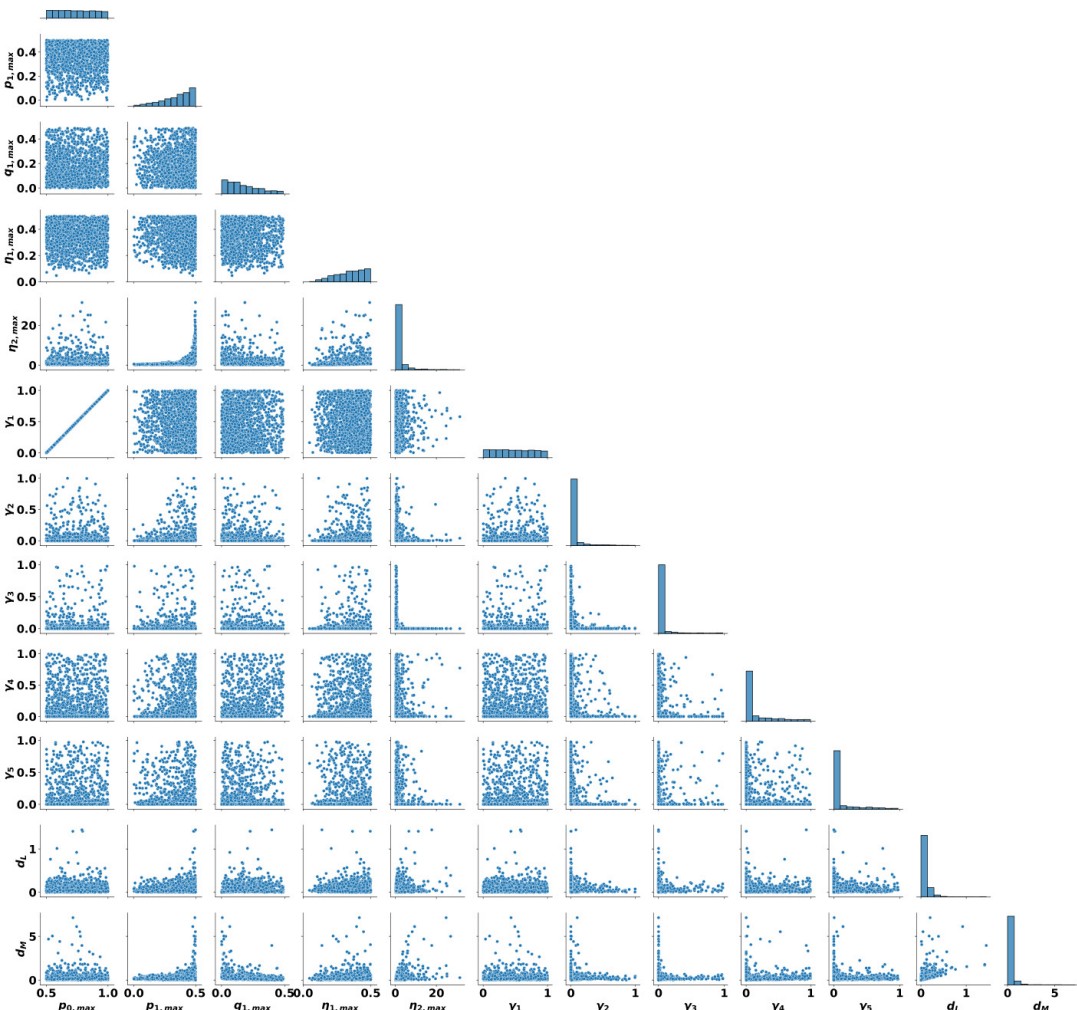

**Appendix 1—figure 4.** Pairwise parameter distributions for all 1493 parameter sets found from the gridsearch. The overall distributions for each parameter are shown along the diagonal.

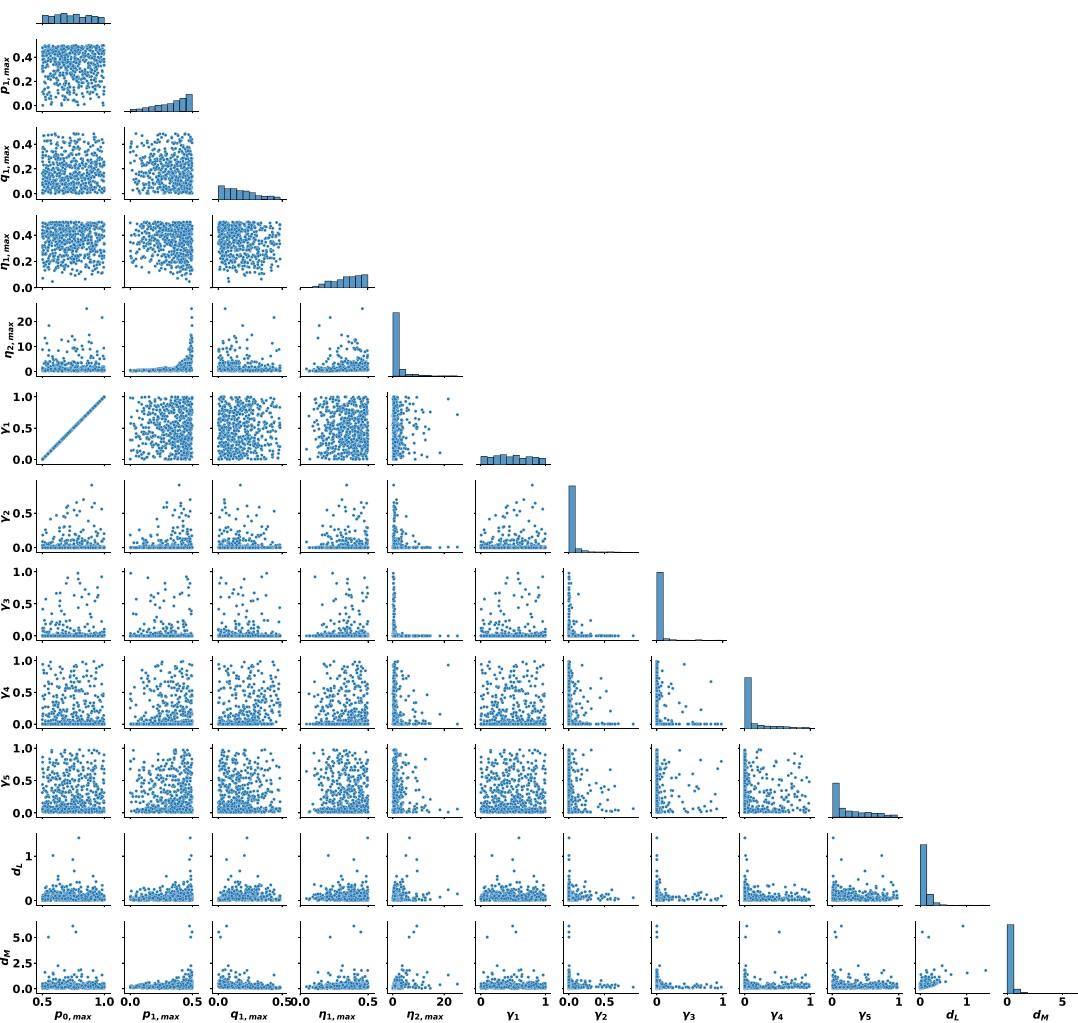

**Appendix 1—figure 5.** Pairwise parameter distributions for the 563 parameter sets from *Appendix 1—figure 4* that have feedforward gain $\gamma_5 > 0.01$. The overall distributions for each parameter are shown along the diagonal.

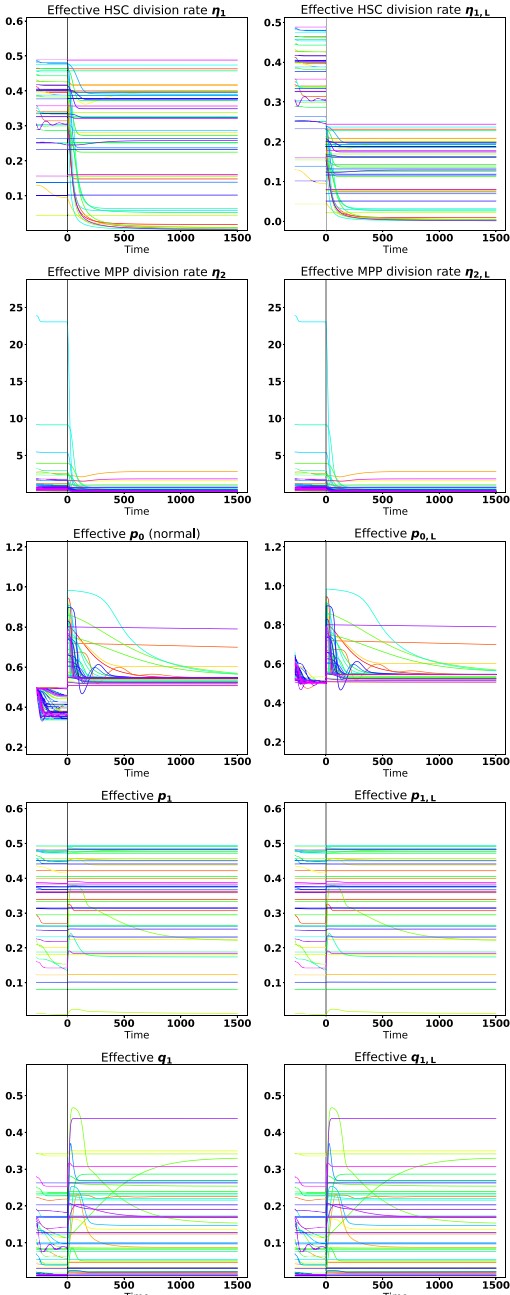

**Appendix 1—figure 6.** A spaghetti plot showing the dynamics of the effective proliferation, self-renewal, and branching parameters for 50 parameter sets under tyrosine kinase inhibitor (TKI) treatment started at early times.

**Appendix 1—table 4.** Parameter values used to model the normal hematopoietic system in *Figures 3–7A, C and D* in the main text.

| Parameter | Value |
|---|---|
| $p_{0,max}$ | 0.756641 |
| $p_{1,max}$ | 0.357913 |
| $q_{1,max}$ | 0.032241 |
| $\eta_{1,max}$ | 0.197639 |

*Appendix 1—table 4 Continued on next page*

*Appendix 1—table 4 Continued*

| Parameter | Value |
| --- | --- |
| $\eta_{2,max}$ | 0.47088 |
| $\gamma_1$ | 0.513281 |
| $\gamma_2$ | 0.197639 |
| $\gamma_3$ | 0.47088 |
| $\gamma_4$ | 0.513281 |
| $\gamma_5$ | 0.543987 |
| $d_L$ | 0.000165 |
| $d_m$ | 0.000428 |

**Appendix 1—table 5.** Additional parameter values used to model the chronic myeloid leukemia (CML) cell population dynamics in *Figures 4A ,B, 5–7A, C and D* in the main text.

| Parameter | Value |
| --- | --- |
| $p_{0,max}^L$ | 0.756641 |
| $p_{1,max}^L$ | 0.357913 |
| $q_{1,max}^L$ | 0.032241 |
| $\eta_{1,max}^L$ | 0.197639 |
| $\eta_{2,max}^L$ | 0.47088 |
| $\gamma_1^L$ | 0.2566405 |
| $\gamma_2^L$ | 0.197639 |
| $\gamma_3^L$ | 0.47088 |
| $\gamma_4^L$ | 0.2566405 |
| $\gamma_5^L$ | 0.543987 |
| $d_L^L$ | 0.000165 |
| $d_m^L$ | 0.000428 |
| $TKI_{HSC}$ | 0.201311 |
| $TKI_{MPP}$ | 0.024757 |

**Appendix 1—table 6.** Secondary parameter values used as a representative case to model nonresponsive chronic myeloid leukemia (CML) cell population dynamics in *Figures 7C, D and 8* in the main text.

| Parameter | Value |
| --- | --- |
| $p_{0,max}$ | 0.838481 |
| $p_{1,max}$ | 0.009776 |
| $q_{1,max}$ | 0.418627 |
| $\eta_{1,max}$ | 0.226025 |
| $\eta_{2,max}$ | 0.247591 |

*Appendix 1—table 6 Continued on next page*

*Appendix 1—table 6 Continued*

| Parameter | Value |
|---|---|
| $\gamma_1$ | 0.676962 |
| $\gamma_2$ | 0.000219 |
| $\gamma_3$ | 0.000326 |
| $\gamma_4$ | 0.367564 |
| $\gamma_5$ | 0.168132 |
| $d_L$ | 0.071273 |
| $d_m$ | 0.155649 |
| $p_{0,max}^L$ | 0.838481 |
| $p_{1,max}^L$ | 0.009776 |
| $q_{1,max}^L$ | 0.418627 |
| $\eta_{1,max}^L$ | 0.226025 |
| $\eta_{2,max}^L$ | 0.247591 |
| $\gamma_1^L$ | 0.338481 |
| $\gamma_2^L$ | 0.000219 |
| $\gamma_3^L$ | 0.000326 |
| $\gamma_4^L$ | 0.367564 |
| $\gamma_5^L$ | 0.168132 |
| $d_L^L$ | 0.071273 |
| $d_m^L$ | 0.155649 |
| $TKI_{HSC}$ | 0.25 |
| $TKI_{MPP}$ | 30 |

## 4 Simulations of transplant experiment

As described in the main text, we simulated a transplant experiment in a transgenic mouse model of CML performed in *Reynaud et al., 2011*. In this experiment, either leukemic stem HSC L or leukemic MPP L cells were implanted into sublethally irradiated mice. Transplantation of HSC L enables engraftment and myeloid cell production that leads to CML. On the other hand, transplanting MPP L cells does not allow for long-term engraftment but results in a larger fraction of donor-derived lymphoid cells after 35 d. We modeled this experiment by reducing the number of cells in equilibrium to mimic the effects of sublethal radiation (see 'Methods'). Here, we present results of a range of possible reductions of HSC L and MPP L cells, and tracked the outcomes when 4000 HSC L or MPP L were introduced after the decrements from equilibrium. We then determine which of our parameter sets are consistent with the experimental outcomes found in *Reynaud et al., 2011* using a simple majority of myeloid cells for HSC L transplant and a simple majority of lymphoid cells for MPP L transplant as consistency criteria. The results are summarized in *Figure 5—figure supplements 1–2*. The pairwise parameter distributions of the 478 remaining parameter sets are shown in *Figure 5—figure supplement 3*.

## 5 Effective parameters

Here, we present the effective proliferation rates and self-renewal and branching factors—that is, the values of these parameters that takes the feedback regulation into account. That is, the effective stem cell proliferation rate $\eta_1 = \eta_{1,max} / (1 + \gamma_2 x_S)$ in normal hematopoiesis and

$\eta_1 = \eta_{1,max}/\left(1 + \gamma_2\left(x_S + x_{S^L}\right)\right)$ when CML stem cells are present. The other effective parameters are defined analogously.

In *Figure 3—figure supplement 2*, the effective HSC and MPP proliferation rates ($\eta_1$, $\eta_2$), the effective HSC and MPP self-renewal fractions ($p_0$, $p_1$) and branching fraction $q_1$ are shown. The corresponding effective parameters are shown in *Figure 6—figure supplement 1* when CML cells are introduced when the normal hematopoietic model system is at steady state and in response to treatment by TKIs (starts at the time labeled $t = 0$, as indicated by the vertical line) for the cases shown in *Figure 6* in the main text. In *Appendix 1—figure 6*, we plot the effective proliferation, self-renewal, and branching parameters for 50 parameter sets under treatment at early times.

## 6 Distributions of model parameters grouped by response to TKI treatment using synthetic data (478 parameter sets)

In *Figure 7—figure supplement 1*, we plot the distributions of all the unregulated proliferation, self-renewal, and branching parameters, as well as the feedback gains, grouped by response to TKI therapy. In particular, the blue color indicates achievement of MR3 by 50 mo (termed as responders) while orange indicates that MR3 is not achieved within 50 mo (termed as nonresponders). The only parameters that clearly delineate the responders from nonresponders are $p_{0,max}$ and $\gamma_1$, with responders occurring at the lower values and nonresponders at the higher values.

## 7 Comparisons of prognostic criteria for predicting response to TKI therapy

### 7.1 Performance of prognostic criteria using synthetic data (478 parameter sets)

Using data generated from our 478 parameter sets, we tested whether prognostic criteria could correctly identify patients who achieve MR3 within 18 mo after therapy starts. We tested the performance of our prognostic criterion (relative change of transcript levels) against two existing clinical prognostics: halving time (the time it takes for the BCR-ABL1 transcripts to reach one-half of their pretreatment value) and early molecular response (EMR) in which the fraction of BCR-ABL1 transcripts are 10% or less after 3 mo of treatment. We also tested a prognostic criterion based on the ratio of transcript levels. These different prognostic criteria were calculated for both the first and second 3 mo after the start of therapy (e.g., 0–3 mo and 3–6 mo). We calculated the corresponding receiving operating characteristic (ROC) curves and found the optimal threshold by maximizing the difference between true and false positive rates. The results are presented in *Appendix 1—figure 7* and reveal that generally the ratio and relative change prognostics offer similar performance, but that both demonstrate somewhat better performance compared to the traditional prognostics. In addition, all the prognostic criteria are more accurate when applied 3–6 mo after the start of therapy than when applied during the 0–3-month period. We did not calculate the ROC curves for EMR but rather we only plotted the point that corresponds to 10% transcript levels at 3 mo (black circle) and 1% at 6 mo (open black circle). The EMR prognostic criterion has fewer false positives but also fewer true positives than the other prognostics.

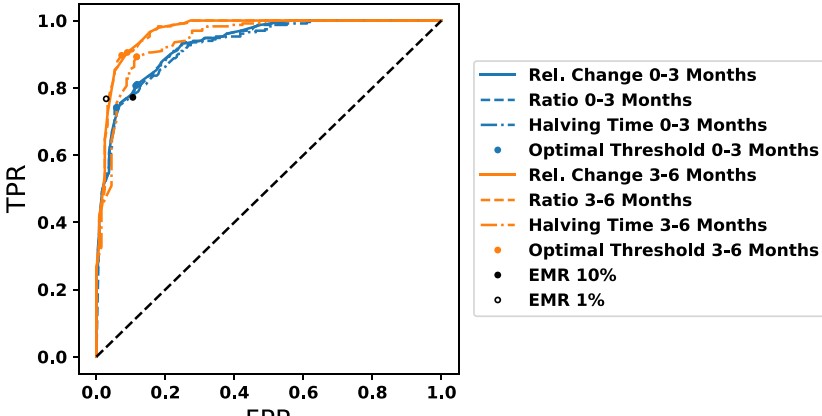

**Appendix 1—figure 7.** Comparison between our prognostic and alternate prognostics, as labeled, at the first and second 3 mo after the start of therapy.

## 7.2 Performance of prognostic criteria using patient data

The prognostic criteria were tested on anonymized patient data obtained from Dr. Van Etten's clinical practice, again asking whether MR3 at 18 mo after treatment could be correctly predicted. We used data in which patients were kept on the same therapy for 6 mo either from the start of therapy or after a change of therapy. For patients that achieved MR3 within 3 mo, we did not include their data at 6 mo. In *Appendix 1—figure 8*, the first two columns correspond to results when the same therapy is applied for 6 mo after patient diagnoses. The last two columns correspond to patients who have had a change of therapy, but the new therapy is maintained for 6 mo. We do not use the EMR as a prognostic in the cases when therapy is changed. The figure demonstrates that the prognostics are more accurate for the 3–6-month period, as predicted from the synthetic data. Although the numbers of patients are small, the relative ratio prognostic criterion is at least as accurate as, or more accurate than, the other criteria. In *Appendix 1—figure 9*, the predictions of the prognostic criteria are grouped by whether the patients achieve or do not achieve MR3 by 18 mo and by time period. In *Appendix 1—figure 10*, the prognostic criteria data are aggregated into 3-month windows, where 0–3 mo contains both 0–3 mo after start of therapy and after a therapy change. The 3–6-month data is similarly aggregated. It is clear that the predictions using the 3–6-month data are more accurate than those using the 0–3-month window.

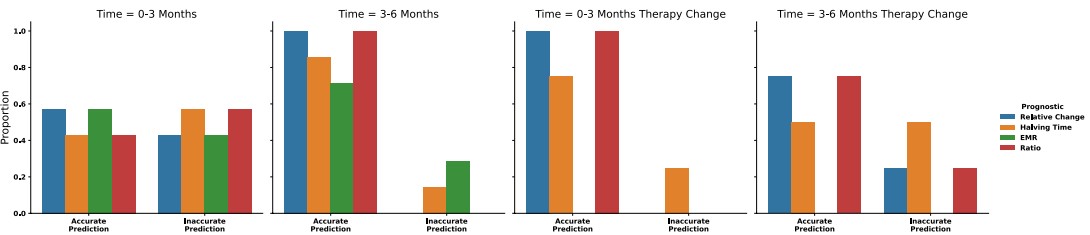

**Appendix 1—figure 8.** The ability of prognostic criteria to predict MR3 by 18 mo is evaluated using anonymized patient data (n = 11) in which patients received the same therapy for 6 mo either from the start of therapy or after a change in therapy. The first two columns correspond to results when the same therapy is applied for 6 mo after patient diagnoses. The last two columns correspond to patients who have had a change of therapy, but the new therapy is maintained for 6 mo. The Early molecular response (EMR) prognostic criterion is not used when the patients have had a therapy change.

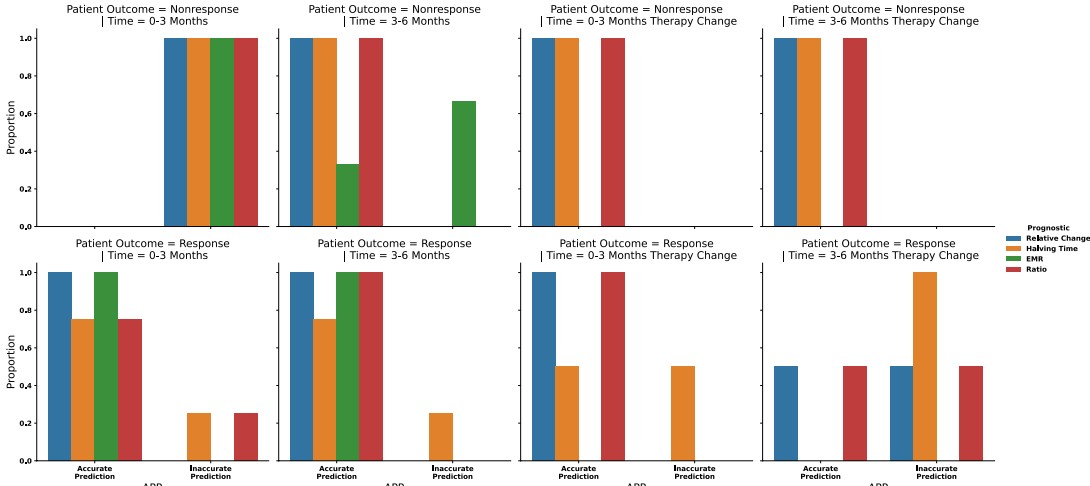

**Appendix 1—figure 9.** The predictive ability of each prognostic criterion from *Appendix 1—figure 8* but grouped based upon patient (n = 11) outcome (blue, responder; and yellow, nonresponder) and the time frame for prediction.

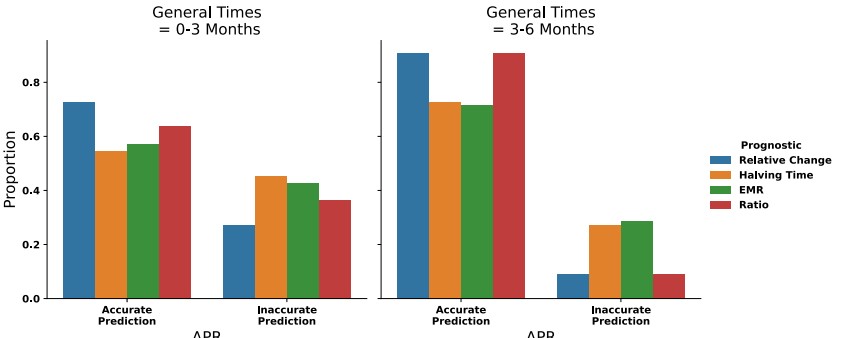

**Appendix 1—figure 10.** Aggregating the accuracy of prognostic criteria predictions with patient data (n = 11) from *Appendix 1—figure 8* where a general time of 0–3 mo contains both 0–3 mo after start of therapy and after therapy change. The 3–6-month data is aggregated similarly.

## 8 Combination therapy parameter distributions by response

Finally, in *Figure 8—figure supplement 1*, we show how the distributions of model parameters from *Figure 7—figure supplement 1* change when TKI therapy is combined with a differentiation promoter. The blue hatching indicates a nonresponder that becomes a responder while the yellow hatching indicates that a responder becomes a nonresponder. Here, response is defined as achieving MR3 in 50 mo. We observe that differentiation therapy is very effective in driving nonresponders at large $p_{0,max}$ and $\gamma_1$ to become responders, but also drives parameter sets that responded to TKI monotherapy at small values of $p_{0,max}$ and $\gamma_1$ to no longer achieve MR3 at 50 mo (nonresponders).

## 9 Impact of intrinsic differences between normal and leukemic cells

In *Figure 4—figure supplements 2–4*, we explored perturbations ranging from 90 to 110% of the original parameter values from *Appendix 1—tables 4 and 5* to explore the effect of intrinsic differences between normal and leukemic cells. One exception is $p_{0,max}^L$, which is limited to a perturbation range of 90–100% due to biological constraints as described in the main text. For the single parameter set for normal cells from *Appendix 1—table 4*, perturbations in most leukemic parameters yielded insignificant differences at the cellular and response dynamics levels. The three parameters that did see significant sensitivity to perturbation were the leukemic stem cell-specific parameters $p_{0,max}^L$, $\frac{\gamma_1^L}{\gamma_1}$, and $TKI_{HSC}$. To determine whether this sensitivity applies to the entire population of parameter sets, we explored sweeps of the newly added leukemic parameters and their associated feedback gain shown in the heat maps of *Appendix 1—figure 11* and *Figure 4—figure supplement 6*. For a parameter combination to be considered useful, there must be a region of the left plots that is of a lower value, such that in a majority of cases leukemic cells can dominate the system. Along the bottom right plot, the region should be neither fully light or fully dark to give regions where there are both responsive parameter sets and nonresponsive parameter sets. Through these parameter combination studies, we find that even on the broader parameter set population $p_{0,max}^L$ and $\frac{\gamma_1^L}{\gamma_1}$ are the only cases with significant differences in overall qualitative outcomes. Additionally, we find that the domains of $p_{0,max}^L$ and $\frac{\gamma_1^L}{\gamma_1}$ are relatively restricted with the only viable values of $p_{0,max}^L$ being roughly equivalent with $p_{0,max}$ with necessary decreases in $\frac{\gamma_1^L}{\gamma_1}$ for lower $\frac{p_{0,max}^L}{p_{0,max}}$ values to ensure parameter combination viability.

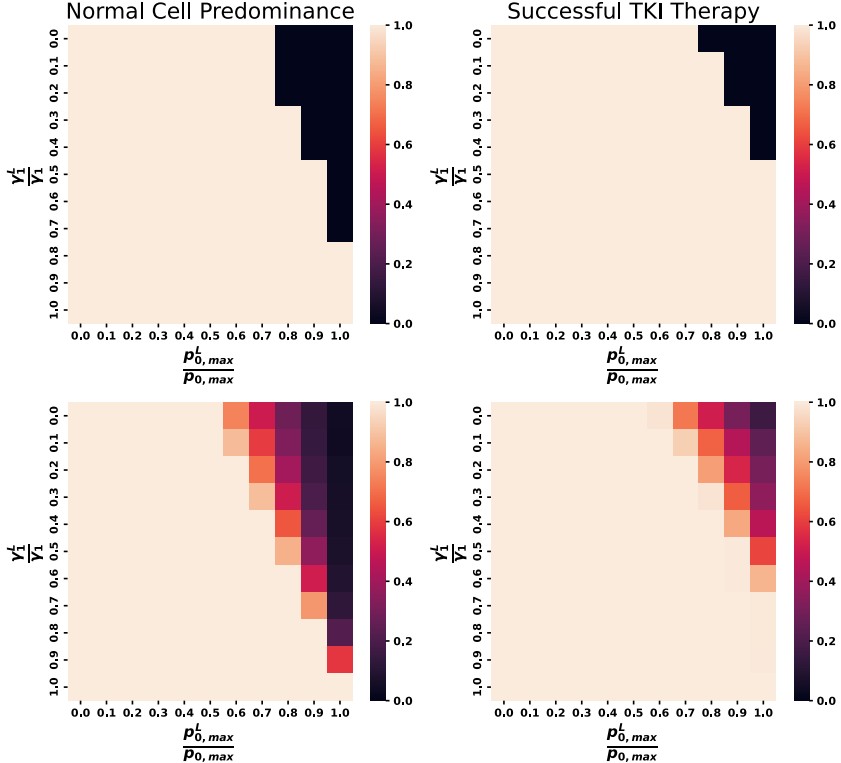

**Appendix 1—figure 11.** Variations in $\frac{p_{0,max}^L}{p_{0,max}}$ and $\frac{\gamma_1^L}{\gamma_1}$ for the individual parameter set from *Figure 6* (top) and all eligible parameter sets (bottom). Left heat maps display the proportion of parameter sets that maintain dominance of normal cells in the system in the lighter regions and the proportion that possess leukemic cell dominance in the darker regions. The right heat maps indicate proportion of cases where parameter sets achieve MR3 within 50 mo with lighter values associated with a higher proportion of response. Biologically relevant parameter combinations exist within darker regions on the bottom left and regions within orange to purple on the bottom-right plot.

## 10 Effect of leukemic stem cell parameters on response to TKI therapy

By finding similar qualitative regions of the leukemic stem cell self-renewal heat map in *Appendix 1—figure 11* (lower right), we examined the distributions of response for a few combinations of parameters shown in *Appendix 1—figure 12*. We found that it is possible to achieve similar distributions with different combinations of parameters. When $p_{0,max}^L/p_{0,max} = 0.8$ and $\gamma_1^L/\gamma_1 = 0.2$, in 33% of the cases the treatment is unsuccessful in achieving MR3 in 50 mo. When $p_{0,max}^L/p_{0,max} = 0.8$ and $\gamma_1^L/\gamma_1 = 0.1$, in 20% of the cases the treatment is unsuccessful in achieving MR3 in 50 mo. When $p_{0,max}^L/p_{0,max} = 0.9$ and $\gamma_1^L/\gamma_1 = 0.3$, in 30% of the cases the treatment is unsuccessful in achieving MR3 in 50 mo. When $p_{0,max}^L/p_{0,max} = 1.0$ and $\gamma_1^L/\gamma_1 = 0.5$, in 39% of the cases the treatment is unsuccessful in achieving MR3 in 50 mo.

When we analyzed the role of the pretreatment leukemic stem cell proportion on response to TKI therapy (*Figure 7A*), we find the results to agree qualitatively across the leukemic parameter combinations (*Appendix 1—figure 13*). From this, to determine whether the $p_0$ or $p_0^L$ is the true predictor of response to TKI therapy $p_0$, we first calculated a quantity we termed a characteristic effective self-renewal fraction to attempt to group these combinations by similarity. The characteristic self-renewal fractions for leukemic and normal stem cells are defined as $\bar{p}_0^L = p_{0,max}^L / \left(1 + \gamma_1^L \bar{N}\right)$ and $\bar{p}_0 = p_{0,max} / \left(1 + \gamma_1 \bar{N}\right)$. We take $\bar{N} = 10^5$ to be a characteristic value of the size of the MPP population. We then analyzed the behavior as a function of the maximal HSC self-renewal fraction $p_{0,max}$ and $\bar{p}_{0,max}^L/p_{0,max}$. The results are shown in *Appendix 1—figure 14* where all 478 parameter sets representing the states of the normal system are considered and the leukemic parameters $p_{0,max}^L/p_{0,max}$ and $\gamma_1^L/\gamma_1$ are varied from 0.6 to 1.0 and 0.1–0.6, respectively, using blue and yellow colors to denote responders and nonresponders. We observe that when the fitness of the $HSC^L$ (as measured by $\bar{p}_{0,max}^L/p_{0,max}$) is sufficiently low (e.g., $\bar{p}_{0,max}^L/p_{0,max} < 0.65$), all the systems respond to TKI therapy. When the $HSC^L$ increase in fitness, the number of nonresponders increases but

nonresponders are only observed when $p_{0,max}$ is above a critical threshold, which depends on $\gamma_1^L$ through $\bar{p}_{0,max}^L/p_{0,max}$ . Note that the fitness of the HSC$^L$ can be increased either by increasing $p_{0,max}^L$ or $\gamma_1^L$ or both.

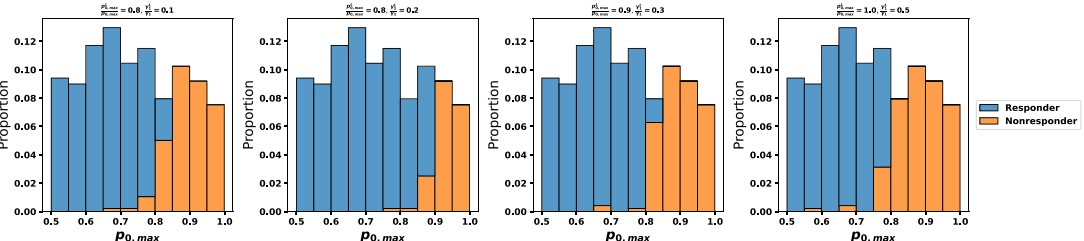

**Appendix 1—figure 12.** Combinations of leukemic values with similar overall response rates to tyrosine kinase inhibitor (TKI) therapy (see text) yield similar distributions of response.

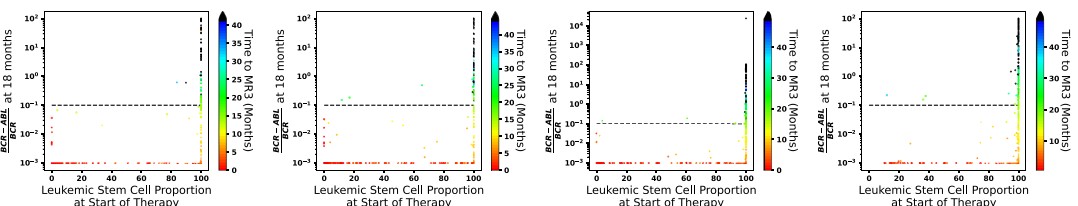

**Appendix 1—figure 13.** Simulated distributions of response to tyrosine kinase inhibitor (TKI) therapy as a function of initial proportion of leukemic stem cells is qualitatively similar across the parameter combinations from *Appendix 1—figure 12* (from left to right): $\frac{p_{0,max}^L}{p_{0,max}} = 0.8$ and $\frac{\gamma_1^L}{\gamma_1} = 0.1$, $\frac{p_{0,max}^L}{p_{0,max}} = 0.8$ and $\frac{\gamma_1^L}{\gamma_1} = 0.2$, $\frac{p_{0,max}^L}{p_{0,max}} = 0.9$ and $\frac{\gamma_1^L}{\gamma_1} = 0.3$, and $\frac{p_{0,max}^L}{p_{0,max}} = 1$ and $\frac{\gamma_1^L}{\gamma_1} = 0.5$.

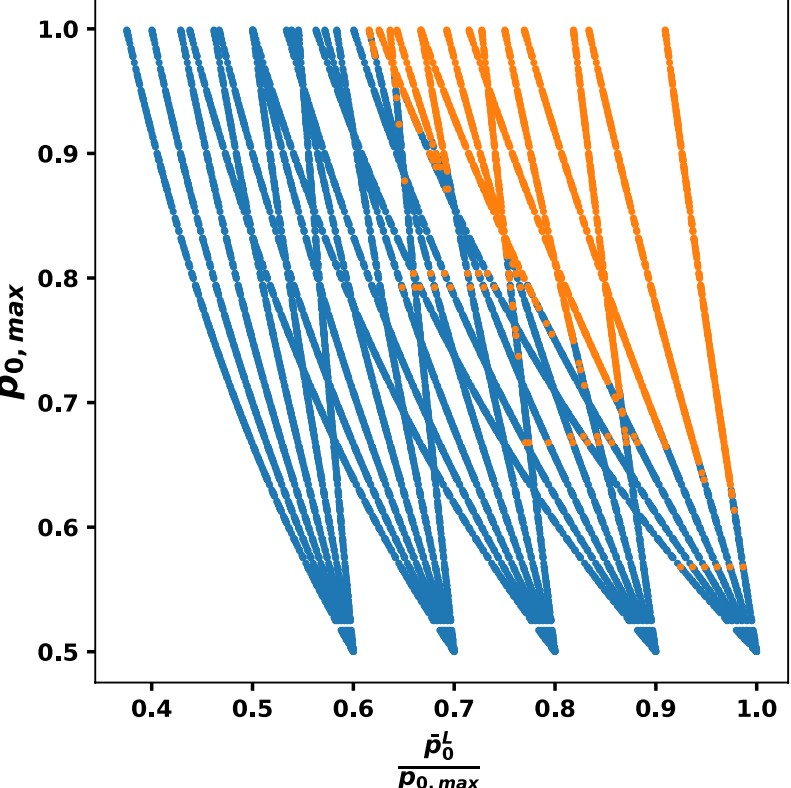

**Appendix 1—figure 14.** Comparison of trajectories of response as functions of $\bar{p}_0^L/p_{0,max}$ and $p_{0,max}$. For each value of $\frac{p_{0,max}^L}{p_{0,max}}$, there is one tree starting from $p_{0,max} = 0.5$, then for each $\frac{\gamma_1^L}{\gamma_1}$ up to $\frac{\gamma_1^L}{\gamma_1} = 0.6$ the tree splits into six branches. Dot color denotes whether a parameter set responds (blue) or does not respond (orange).

The overall fitness of the leukemic stem cells relative to that of the normal cells determines whether CML will develop and whether treatments will succeed or fail. This is shown in **Figure 7—figure supplement 2**. The relative fitness of the CML cells is measured by the ratios of characteristic values of the $HSC^L$ and $HSC$ self-renewal fractions: $\bar{p}_0^L/\bar{p}_0$. Here, all 478 parameter sets representing the states of the normal system are considered and the leukemic parameters $p_{0,max}^L/p_{0,max}$ and $\gamma_1^L/\gamma_1$ are varied from 0.6 to 1.0 and 0.1–0.6, respectively. The larger the relative fitness, the more likely that CML will develop and take

over the system (**Figure 4D**) and that the system will be refractory to TKI treatment (**Figure 7—figure supplement 2**).

As we described in the main text, we estimated the relative fitness of $HSC^L$ as $\bar{p}_0^L/\bar{p}_0 \approx 0.70$. In **Appendix 1—figures 15–16** and **Figure 7B**, we plot the bivariate histogram for response to TKI treatment for combinations of leukemic stem cell parameters such that $\bar{p}_0^L/\bar{p}_0 \geq 0.50$ (**Appendix 1—figure 15**), $\bar{p}_0^L/\bar{p}_0 \geq 0.60$ (**Appendix 1—figure 16**), and $\bar{p}_0^L/\bar{p}_0 \geq 0.7$ (**Figure 7B**). Through exploration of bivariate and marginal distributions, we find that $p_{0,max}$ is capable of separating response, while the fitness $\bar{p}_0^L/p_{0,max}$ does not have a clear delineation (**Appendix 1—figures 15–16** and **Figure 7B**). Additionally, in our virtual patient population we consider variations of $p_{0,max}$ for individual biological variation with CML cells operating in a similar capacity across virtual patients to be more meaningful.

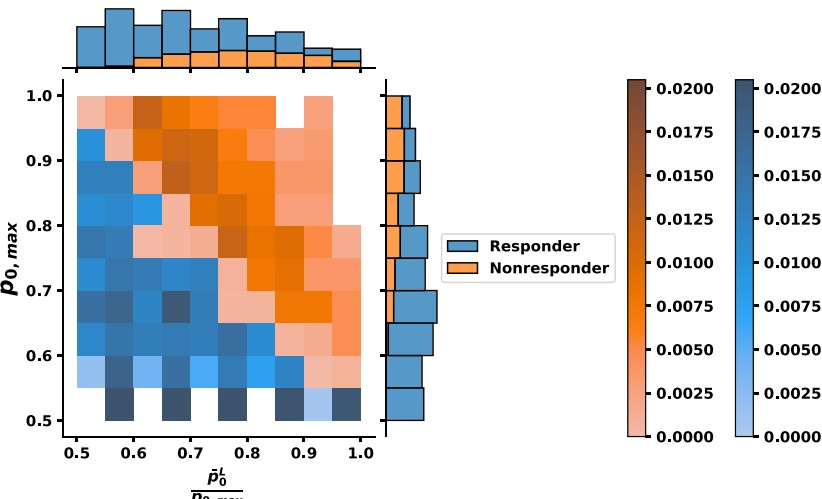

**Appendix 1—figure 15.** Bivariate histogram for response to tyrosine kinase inhibitor (TKI) therapy where the relative fitness of $HSC^L$ is $\bar{p}_0^L/\bar{p}_0 \geq 0.50$. Individual variable marginal distributions are shown along the sub-axes.

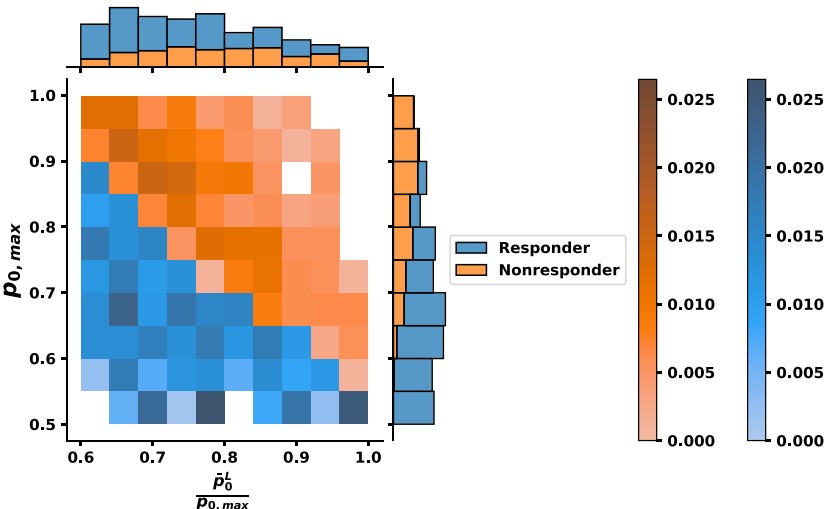

**Appendix 1—figure 16.** Bivariate histogram for response to tyrosine kinase inhibitor (TKI) therapy where the relative fitness of $HSC^L$ is $\bar{p}_0^L/\bar{p}_0 \geq 0.60$. Individual variable marginal distributions are shown along the sub-axes.

## 11 The effect of leukemic stem cell parameters on combination therapy and prognostic criterion

We checked the combinations of $\frac{p_{0,max}^L}{p_{0,max}}$ and $\frac{\gamma_1^L}{\gamma_1}$ to ensure that the effectiveness of combination therapy and the accuracy of our prognostic criterion are largely unchanged by varying the leukemic stem cell parameters. In **Figure 8—figure supplement 2**, we see combination therapy is still successful at improving the proportion of responders. We find that the optimal values from **Figure 7E**, where a single set of leukemic parameters was used, need to be modified when all the leukemic parameter combinations that have significant takeover proportions (> 60%) are considered. Nevertheless, using synthetic data we find that the 3–6-month time frame has a better predictivity of TKI response than does the 0–3-month time frame (**Appendix 1—figure 17**).

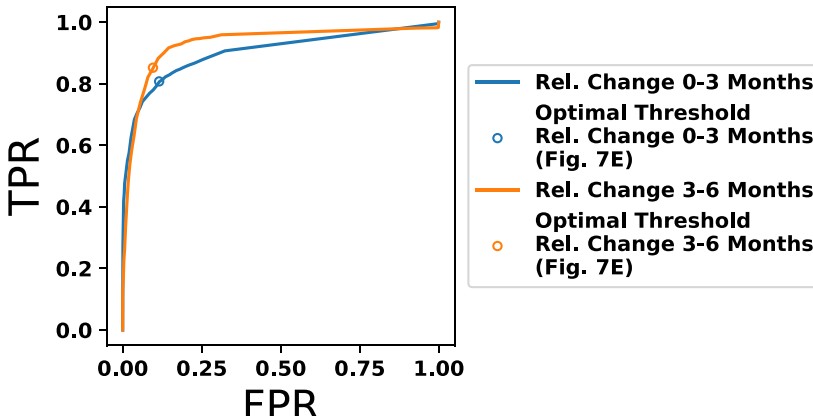

**Appendix 1—figure 17.** Prognostic value sweep applied to combinations of $\frac{p^L_{0,max}}{p_{0,max}}$ and $\frac{\gamma^L_1}{\gamma_1}$ with leukemic predominance > 60%. Optimal values from *Figure 7E* are no longer the optimal across all combinations, but the 3–6-month time frame still outcompetes 0–3-month time frame.

