## [Editor Report]

This is an important study that investigates the impact of tyrosine kinase inhibitors (TKIs) in chronic myeloid leukemia. Through a combination of preclinical in vivo measurements, clinical data, and computational modeling, the authors present solid evidence regarding the heterogeneous effects of TKIs in patients and how the response to treatment may be improved. This study is of interest to those working in the fields of mathematical oncology and cancer biology.

---

## [Decision Letter]

**Decision letter after peer review:**

Thank you for submitting your article "Predictive nonlinear modeling of malignant myelopoiesis and tyrosine kinase inhibitor therapy" for consideration by *eLife*. Your article has been reviewed by 3 peer reviewers, and the evaluation has been overseen by a Reviewing Editor and Aleksandra Walczak as the Senior Editor. The following individual involved in the review of your submission has agreed to reveal their identity: Jesse A Sharp (Reviewer #3).

Essential revisions:

1) Account for intrinsic differences between leukemic and normal cells. This aspect of the manuscript is poorly justified and weakens the study.

*Reviewer #1 (Recommendations for the authors):*

A) Several assumptions and hypotheses require a more detailed justification or a more careful discussion, e.g.,

– "For simplicity, we assumed the only difference between the two lineages is a decrease in the feedback strength for leukemic HSC (HSC L ; S L ), as indicated by p 0L in the schematic in Figure 4A." This choice requires a more detailed justification. Why are potential differences in other parameters not considered?

– "TKI treatment was initiated at three different times to achieve varying leukemic stem cell load (6, 18, and 36 months) and was simulated by introducing a death rate of HSC L and MPP L proportional to their proliferation rates, with the HSC L proliferation rate lower than that of normal HSC (Jørgensen et al., 2006)" Several researchers in the field held the strong opinion that TKI do not affect the CML stem cell population in humans (Corbin et al., cited by the authors). Also, the work of Reynaud (cited by the authors) discusses this issue. Please provide a more detailed discussion of the assumption that TKI kills leukemic stem cells.

– Along the lines of point (2) above: "The model further suggests that a key predictor of refractory response to TKI treatment is an increased probability of self-renewal of normal hematopoietic stem cells." and related statements: Please rephrase the statements more carefully pointing out the meaning of the parameter p_{0,max} and the model assumptions underlying this conclusion.

– Along the lines of point (2) above: To support the conclusion that normal and not leukemic stem cell self-renewal matters for TKI response, please consider analyzing a version of the model allowing for different p_{0,max} values for HSC and HSCL. In such a model: What is the predictor of poor TKI response, the p_{0,max} for HSC or the p_{0,max} for HSCL?

– "This is consistent with clinical data that suggest that CML patients with pre-existing mutations in genes such as TET2 and ASXL1, which are known to increase stem cell self-renewal (Steensma, 2018), tend to have inferior outcomes under TKI therapy (Kim et al., 2017; Marum et al., 2017)." In my opinion, the provided references argue for the concept that a simultaneous increase of self-renewal in both leukemic and normal stem cells leads to a low TKI efficacy. The CHIP mutations analyzed in the cited studies most probably affect normal AND leukemic stem cells. Marum et al. study germ-line mutations that naturally affect normal and leukemic cells at the same time.

– "Although our data were derived from mice, we hypothesize that similar cell-cell

signaling occurs in humans." This hypothesis should be discussed in more detail. What are the potential differences between mice and humans that could lead to problems when applying the proposed model to patients?

– "only those with sufficiently large feedforward gains on the MPP division rate (γ 5 >0.01)" Please justify the cutoff of 0.01, how sensitive are the presented results with respect to this choice?

– "We used one eligible parameter set (see Supplemental Tables S4, S5), which is capable of characterizing the normal state of our simplified model of the hematopoietic system." Do other parameter sets lead to comparable results?

B) Several passages of the main text are difficult to follow since important arguments have been moved to the Methods Section at the very end or to the Supplement. Some conclusions merit a more detailed explanation. E.g.

– "Taking all these results into consideration, we arrive at the feedback-feedforward model shown in Figure 2F." This statement should be explained in more detail in the main text. Have the authors performed a systematic search of the literature to search for evidence of such mechanisms, can they name potential factors mediating them?

– "These time points and the number of mice analyzed at each time point were informed by a Bayesian hierarchical framework for optimal experimental design of mathematical models of hematopoiesis (Lomeli et al., 2021)"

Please provide details.

*Reviewer #2 (Recommendations for the authors):*

Very nicely done work and I appreciate the efforts to validate the model and parameter estimates using sophisticated computational techniques as well as animal data.

A well-written manuscript and the right amount of detail.

*Reviewer #3 (Recommendations for the authors):*

I commend the authors on a well-written manuscript with important results and clear methodology.

Here are some recommendations which I hope may improve the manuscript. I will preface these recommendations by noting that I have limited experience in a laboratory setting and as such do not have the expertise required to make any comments about the experimental component of this work.

– The authors could provide a brief justification for a grid search as the mechanism for identifying biologically relevant parameters.

– In modelling CML development, there is an assumption that the only difference between normal and leukemic cell lineages is the feedback strength for leukemic HSC self-renewal. Could the validity of this assumption be explored, perhaps via supporting evidence in the literature, or through briefly considering the sensitivity of the model predictions to this assumption?

– In Figure 1, concerning the process of identifying candidate model classes, I am not sure whether it is not present or simply hard to make out in the figure, but it appears to me that the downregulation of q_1 by TD_m marked in blue in Figure 1C is not a possible a factor in any of the four candidate model classes identified through design space analysis (Figure 1B). This is in contrast to the bottom left model of Figure 1B, where the downregulation of p_0 by p (also marked with blue in Figure 1C) is visible as a possibility.

---

## [Author Response]

Essential revisions:1) Account for intrinsic differences between leukemic and normal cells. This aspect of the manuscript is poorly justified and weakens the study.

We have now performed a more thorough investigation of the leukemic parameters and their effect on CML hematopoiesis and response to TKI therapy. We describe the results below in our detailed responses to the Reviewers. We have also modified the paper to reflect this investigation in both the main manuscript (see modifications in the Results, Discussion and Methods sections) and in Appendix 1, where we introduce new sections 9-11, describing the effect of the leukemic parameters on the results (see revised Figures 4C,D and Figure 7B in the main manuscript, Figure 4—figure supplement 6, Figure 6—figure supplements 2-4, Figure 7—figure supplement 2, Figure 8—figure supplement 2, and Appendix 1–figures 11-17).

Our results can be summarized as follows. We found that only the leukemic cell parameters for the leukemic stem cells (HSC^L^)—the maximal HSC^L^ self-renewal fraction p0,maxL, the feedback gain γ1L on the HSC^L^ self-renewal fraction , and the TKI-induced HSC^L^ death rate TKIHSCL—have the potential to significantly influence the results. The results are insensitive to changes in the other leukemic cell parameters (see Appendix 1–table 3 for a list of the parameters, Appendix 1–table 3 for the parameters used in Figures 4-7, Appendix 1 Sec. 9 for sensitivity studies of the CML parameters, Figure 6—figure supplements 2-4, and Appendix 1–figure 11). A careful investigation indicates that these leukemic stem cell parameters influence the fitness of the leukemic cells and this can also be used to predict outcomes of CML hematopoeisis and TKI therapy. Nevertheless we find that for sufficiently fit leukemic cells, which we define more precisely and estimate below (see Figure 4, Figure 7—figure supplement 2, Figure 7B and Appendix 1-figure 14 and the associated text), the main conclusions of the paper still hold. In particular, the maximum probability of self-renewal of the normal hematopoetic cells is a key predictor of refactory response to TKI treatment, the new clinical prognostic criterion is still predictive of outcomes, and differentiation therapy can significantly improve treatment outcomes.

The biological reason for this is as follows. If the CML stem cells are not sufficiently fit relative to the normal stem cells, these *BCR-ABL1*-expressing stem cells may not be able to produce clinical disease (e.g., come to predominate in the bone marrow and blood). Further, even if the CML stem cells are fit enough to drive the development of leukemia, since TKI therapy decreases their effective fitness, an even higher level of fitness of the CML stem cells is required for the disease to be refractory to therapy. Otherwise, the CML will always respond to therapy. See Appendix 1-figure 11, Figure 4 and Figure 7—figure supplement 2. On the other hand, if the CML cells are sufficiently fit relative to the normal stem cells, then increasing the self-renewal fraction of the normal stem cells has the effect of also increasing the fitness of the CML stem cells. Thus, we find that the self-renewal fraction of the normal stem cells is still predictive of outcomes to TKI therapy.

Reviewer #1 (Recommendations for the authors):A) Several assumptions and hypotheses require a more detailed justification or a more careful discussion, e.g.,1. "For simplicity, we assumed the only difference between the two lineages is a decrease in the feedback strength for leukemic HSC (HSC L ; S L ), as indicated by p 0L in the schematic in Figure 4A." This choice requires a more detailed justification. Why are potential differences in other parameters not considered?

We thank the reviewer for making this point. We have now performed a more thorough investigation of the leukemic parameters and their effect on CML hematopoiesis and response to TKI therapy. We have carefully investigated the effect of changes in the other leukemic cell parameters on the results. We describe these in detail below. We have incorporated these results into the main text, supplemental figures, and Appendix 1.

a). Only the leukemic stem cell (HSC^L^) parameters are found to have a significant impact on the results. All other leukemic cell parameters have a negligible effect. To illustrate this, we present a sensitivity analysis of the simulation in Figure 6A (early time) where we perturbed each of the leukemic parameters within 10% of the values used in that figure. However, we constrained the maximal HSC^L^ self-renewal fraction p0,maxL to be less than or equal to p0,max, the maximal self-renewal fraction of the normal stem cells (HSC). Although leukemic stem cells predominate the BM in most patients diagnosed with CML (*Thielen et al., 2016*), there is little evidence that p0,maxL could be larger than p0,max. On the contrary, in vitro and in vivo studies suggest the opposite. For example, CML long-term culture initiating cells (LTC-IC; thought to be biologically similar to stem cells) decrease significantly in in vitro cultures while the number of normal LTC-IC is unchanged, consistent with a relative decrease in self-renewal probability of the CML cells (*Udomsakdi et al., 1992*). in vivo evidence for a lower intrinsic self-renewal comes from studies that attempted to engraft NOD-SCID mice with human CML bone marrow (BM). These mice can be efficiently engrafted with normal human BM, and also engraft with human AML samples, which leads to AML-like disease, morbidity and death. By contrast, attempts to engraft NOD-SCID mice with CML BM samples led to preferential engraftment by normal HSC. Although PH^+^ cells did engraft, engraftment was highly variable and none of the mice developed a CML-like syndrome (*Wang et al., 1998*). We have revised the text in section “Extension of the hematopoiesis model to CML” to include this discussion and created a new section Appendix 1 Sec. 9 to describe these results.

The results for the HSC^L^ parameters—the maximal self-renewal fraction p0,maxL, the feedback gain γ1L, and TKI-induced HSC^L^ death rate TKIHSCL -- are shown in Figure 6—figure supplement 2. To illustrate the insensitivity of the results to the other parameters, we present results for the maximal self-renewal fraction of the leukemic multipotent progenitor cells (MPP^L^) p1,maxL in Figure 6—figure supplement 2. The effect of 10% variations in all of the leukemic cell parameters are shown in the revised Figure 6—figure supplement 2-4 and details in Appendix 1 Sec. 9. These results are characteristic of even larger changes in the base parameters. In Figure 6—figure supplement 2, the stem and progenitor cell numbers are in the first column, the terminally differentiated cells are in the second and the third column shows the simulated molecular responses to TKI treatment. The shaded regions denote the range of the dynamics while the while the curves denote the original dynamics of the base case.

As seen in Figure 6—figure supplement 2, perturbations in p0,maxL have the largest influence on the results, although in this case the leukemic stem and progenitor cells are decreased and the system responds better to therapy because only values of p0,maxL that are less than or equal to p0,max are considered. When γ1L, and TKIHSCL are perturbed, the variation is smaller but the leukemic cell populations can increase or decrease and the treatment outcomes can be improved or worsened.

As indicated by the very narrow shaded region in Figure 6—figure supplement 2, the results are insensitive to perturbations to p1,maxL.

b). In order for CML hematopoiesis to dominate the system and cause therapy to fail, the CML stem cells should have p0,maxL sufficiently close to p0,max and γ1L should be sufficiently small. We considered the influence of a wider range of leukemic stem cell parameters and normal stem cell parameters and investigate the fitness of the leukemic stem cells, relative to the normal stem cells, and the response of the system to therapy, focusing on the effects of γ1L and p0,maxL. The results are presented in Appendix 1-figure 11. We have also placed Appendix 1-figure 11(C) in the main text as a new Figure 4(C) and appropriately modified the text in section “Extension of the hematopoiesis model to CML” to describe the new figure.

The heat maps in Appendix 1-figure 11(A,C) show the outcomes after CML stem cells are introduced and how the system responds to TKI therapy (B,D) as functions of the ratios p0,maxL/p0,max and γ1L/γ1. Panels A and B show results where all the other parameters correspond to those used in the base case from Figure 6—figure supplement 2. We observe that CML cells predominate hematopoiesis when p0,maxL/p0,max is close to 1 (dark regions in A). Here, “predominate” is defined as greater than 50% of the terminal cells being leukemic. As the ratio p0,maxL/p0,max decreases from 1, takeover by CML cells requires smaller feedback gains γ1L to compensate. Further, there are threshold values of the parameters required for takeover. Namely, the system is dominated by normal cells (CML cells do not take over) when p0,maxL/p0,max is sufficiently large or when γ1L/γ1 is sufficiently small.

In Appendix 1-figure 11C-D we consider the behavior of the system when all 478 parameter sets are considered. In panel C, the heat map shows the proportion of the cases in which the system is dominated by normal cells after CML is introduced. The results are similar to that obtained for the base case. In order for the CML cells to take over a large proportion of the cases, the ratio p0,maxL/p0,max should be close to 1 and if this ratio decreased, takeover by CML cells requires a smaller relative feedback gain ratio γ1L/γ1 (i.e., a smaller effect of negative feedback on leukemic stem cells relative to normal stem cells) to compensate. Since experimental evidence (clinical and from mouse models) suggests that CML cells generally dominate the hematopoietic system, this places biologically-relevant constraints on the values of p0,maxL and γ1L. (see our response to Reviewer 1 point 2 for estimates).

In panel D, the heat map shows the proportion of cases in which the system achieves MR3 under TKI therapy within 50 months (successful therapy). In the range of the biologically-relevant self-renewal ratios and feedback strengths, we observe that a proportion of the cases do not achieve MR3 after TKI treatment (darker regions). The proportion of unsuccessful treatments increases when the ratio p0,maxL/p0,max increases and/or γ1L decreases. The region of unsuccessful TKI treatments is a subset of the takeover region because TKI-induced cell death decreases the fitness of HSC^L^, and thus to survive treatment, the HSC^L^ must be fitter than would be required to dominate hematopoiesis in the absence of treatment.

c). When there is a significant proportion of cases in which MR3 is not achieved, the self-renewal fraction of the normal stem cells p0,max still predicts refractory response to TKI treatment. This is illustrated in Appendix 1-figure 12 using the combinations (A) p0,maxL/p0,max=0.8 and γ1Lγ1=0.1 where in 33% of the cases the treatment is unsuccessful; (B) p0,maxL/p0,max=0.8 and γ1Lγ1=0.2 where in 20% of the cases the treatment is unsuccessful; (C) p0,maxL/p0,max=0.9 and γ1Lγ1=0.3 where 33% of treatments are unsuccessful and (D) p0,maxL/p0,max=1.0 and γ1Lγ1=0.5 where 39% of treatments are unsuccessful (note that D is the original parameter set shown in the manuscript).

Summarizing, the only parameters that have a significant effect on CML hematopoiesis and response to TKI treatment are p0,maxL/p0,max, γ1L/γ1, and TKIHSCL. Further, in order for CML to dominate the hematopoietic system and be refractory to treatment, p0,maxL/p0,max should be sufficiently close to 1 and γ1L/γ1 should be sufficiently small. The value of TKIHSCL determines the level of potency at which therapy will fail to achieve MR3. For choices of leukemic stem cell parameters for which there is a significant proportion of non-responders, p0,max still predicts refractory response to TKI treatment as presented in the original manuscript.

2. "TKI treatment was initiated at three different times to achieve varying leukemic stem cell load (6, 18, and 36 months) and was simulated by introducing a death rate of HSC L and MPP L proportional to their proliferation rates, with the HSC L proliferation rate lower than that of normal HSC (Jørgensen et al., 2006)" Several researchers in the field held the strong opinion that TKI do not affect the CML stem cell population in humans (Corbin et al., cited by the authors). Also, the work of Reynaud (cited by the authors) discusses this issue. Please provide a more detailed discussion of the assumption that TKI kills leukemic stem cells.

We recognize that several publications have concluded, based on in vitro studies, that CML stem cells are refractory to TKI killing. Originally, Graham et al. (Blood 2002; 99:319-325), which we also cite, demonstrated that a quiescent CD34+CD38– CML progenitor population was resistant to imatinib in vitro. Subsequently, Corbin et al. showed that primitive CML cells were resistant to short-term (72h) TKI treatment despite substantial inhibition of BCR-ABL1 kinase activity. Reynaud et al. cited Corbin as a source for the statement that CML stem cells are TKI-resistant but did not carry out any experiments directly addressing this. However, clinical studies argue that long-term TKI treatment can decrement the CML stem cell population, at least in some patients. The only consistent factor that predicts whether a patient can remain in deep molecular remission (*BCR-ABL1* transcripts undetectable) following TKI withdrawal is the length of time spent in deep remission while on TKI therapy (*Etienne et al., 2017*).; Chen et al., Front Oncol 2019;9:372, suggesting that TKI treatment can gradually decrease CML stem cell frequency or function over time. This hypothesis is further supported by mathematical modeling of patient *BCR-ABL1* transcript data (Tang et al., Blood 2011; 118:1622-1631).

This remains an unsettled question in the CML field, but we have added a brief mention of these studies to the text in section “Leukemic stem cell load influences TKI therapy outcomes.”

3. Along the lines of point (2) above: "The model further suggests that a key predictor of refractory response to TKI treatment is an increased probability of self-renewal of normal hematopoietic stem cells." and related statements: Please rephrase the statements more carefully pointing out the meaning of the parameter p_{0,max} and the model assumptions underlying this conclusion.

The reviewer raises an excellent point. The reviewer is correct that it is the overall fitness of the leukemic stem cells relative to that of the normal cells that determines whether CML hematopoiesis will predominate and whether TKI treatment will succeed or fail. This is shown Fig 4 where the relative fitness of the CML cells is measured by the ratios of characteristic effective values of the HSC^L^ and HSC self-renewal fractions: p¯0L/p¯0. Here, all 478 parameter sets representing the states of the normal system are considered and the leukemic parameters p0,maxL/p0,max and γ1L/γ1 are varied from 0.6-1.0 and 0.1-0.6, respectively. The characteristic effective self-renewal fractions for leukemic and normal cells are defined as γ1L/γ1 and p¯0=p0,max/(1+γ1N¯), where N¯=105, a characteristic value of the size of the MPP population based on MPP steady state values (*Manesso et al., 2013*). The larger the relative fitness, the more likely that leukemic hematopoiesis will predominate the system Figure 4(D) and that the system will be refractory to TKI treatment Figure 7—figure supplement 2.

We have added accompanying text in section “Extension of the hematpoiesis model to CML”. Further analysis of the leukemic parameter combinations for CML hematopoiesis and under treatment has been added in Appendix 1 Secs. 9 and 10 in addition to Figure 4—figure supplement 6.

However, we have shown in Appendix 1-figure 12, using 3 examples where the leukemic parameters are varied and there is a significant proportion of non-responders (e.g., systems that do not achieve MR3), the normal stem cell self-renewal fraction p0,max does predict refractory response to TKI treatment.

We next considered all the combinations of p0,maxL/p0,max and γ1L/γ1 described above for each of the 478 data sets and plotted the outcomes of TKI therapy as a function of p0,max and p¯0L/p¯0 in Appendix 1-figure 14 using blue and orange colors to denote responders and non-responders, respectively. We observe that when the fitness of the HSC^L^ (as defined by p¯0Lp0,max) is sufficiently low (e.g., p¯0Lp0,max<0.65), all the systems respond to therapy. When the HSC^L^ increase in fitness the number of non-responders increases but non-responders are only observed when p0,max is above a critical threshold, which depends on γ1L through p¯0Lp0,max. Note that the fitness of the HSC^L^ can be increased either by increasing p0,maxL or decreasing γ1L or both.

We expect the HSC^L^ to be sufficiently fit for the following reasons. First, there is a significant proportion of patients (10-15%) who do not respond adequately to initial TKI treatment (Hanfstein et al., 2012; Marin et al., 2012) and BCR-ABL1 mutations are generally not present in this group of patients (Zhang et al., 2009; Pietarinen et al., 2016). From Appendix 1-figure 11(C)-(D) this suggests that the maximal self-renewal fraction p0,maxL of the CML stem cells should be close to that of the normal stem cells p0,max. Second, several studies suggest that CML stem cells are at least 5-10 times less sensitive to MIP-1a/CCL3-regulated self-renewal inhibition (Eaves et al., 1993; Chasty et al., 1995; Wark et al., 1998; Durig et al., 1999). That is, we expect γ1Lγ1 should be at less than 0.2 or 0.1. Our results in Appendix 1-figure 12(D) suggest that to be consistent with all this data, we should expect p0,maxL/p0,max≈0.8. As shown in Appendix 1-figure 12(A) and (C) the combination of p0,maxL/p0,max=0.8 and γ1Lγ1=0.1 can be qualitatively reconstructed with the values used in the main text. As a result of the expected values, we expect p¯0Lp0,max should be greater than 0.67, corresponding to p0,maxL/p0,max=0.8 and γ1Lγ1 <0.2 , or greater than 0.73, corresponding to p0,maxL/p0,max=0.8 and γ1Lγ1 <0.1. Therefore, taking p¯0Lp0,max≥0.7 as an example, we can replot the results in Appendix 1-figure 14 using a heat map, as seen in Figure 7(B), which shows the proportion of responders and non-responders as a joint distribution of the parameters p0,max and p¯0L/p0,max. Summing the results over p0,max results in the marginal distribution of p¯0L/p0,max shown at the top of the plot, which indicates that the fitness parameter p¯0Lp0,max does not distinguish between responders and non-responders. However, summing the results over p¯0Lp0,max gives the marginal distribution of p0,max on the right hand side of the plot and confirms that p0,max can be used to predict refractory response for a wide range of HSC^L^ parameters, as suggested by Appendix 1-figure 12. The biological reason for this is as follows. If the CML cells are sufficiently fit relative to the normal stem cells, then increasing the self-renewal fraction of the normal stem cells has the effect of also increasing the fitness of the CML stem cells, thereby making refractory outcomes to treatment more likely.

These figures, and associated text, have been incorporated in Appendix 1 Sec. 10 in the (see Appendix 1–figures 13, 15-16) where we also present heatmaps corresponding to different minimum fitness thresholds of p¯0Lp0,max. As seen in Appendix 1, when the minimum threshold is decreased, the trend remains: larger p0,max correlates with non-response although the correlation becomes weaker because the fitness of the HSC^L^ decreases. For example, as seen from Appendix 1-figure 14 for p¯0Lp0,max < 0.65, all the systems respond to TKI therapy regardless of the value of p0,max.

We also modified the “Discussion” section summarizing our findings for the leukemic parameters, the efficacy of maximum stem cell self-renewal as a predictor, and that our prognostic criterion and theoretical combination therapy are robust to modifications of the leukemic stem cell parameters within the fitness thresholds identified above.

4. Along the lines of point (2) above: To support the conclusion that normal and not leukemic stem cell self-renewal matters for TKI response, please consider analyzing a version of the model allowing for different p_{0,max} values for HSC and HSCL. In such a model: What is the predictor of poor TKI response, the p_{0,max} for HSC or the p_{0,max} for HSCL?

We agree with the reviewer that it is important to distinguish whether the self-renewal fractions of the HSC^L^ and HSC matter for TKI response. While the reviewer is correct that the overall fitness of the HSC^L^ relative to the normal HSC is key for predicting refractory response, when the HSCL are sufficiently fit (see our response to Reviewer 1 point 2), the maximal self-renewal probability of normal stem cells p_[0,max] is still predictive of refractory response to TKI treatment. Please see our response to Reviewer 1 point 2 for a detailed explanation.

5. "This is consistent with clinical data that suggest that CML patients with pre-existing mutations in genes such as TET2 and ASXL1, which are known to increase stem cell self-renewal (Steensma, 2018), tend to have inferior outcomes under TKI therapy (Kim et al., 2017; Marum et al., 2017)." In my opinion, the provided references argue for the concept that a simultaneous increase of self-renewal in both leukemic and normal stem cells leads to a low TKI efficacy. The CHIP mutations analyzed in the cited studies most probably affect normal AND leukemic stem cells. Marum et al. study germ-line mutations that naturally affect normal and leukemic cells at the same time.

We agree with the reviewer, which is why we used the term “pre-existing mutations”. In the study by Kim et al., some of the patients with suboptimal response to TKI therapy had mutations in *TET2* or *ASXL1* that were present in both T cells and myeloid cells at diagnosis, indeed suggesting the existence of a pre-leukemic mutant clone, such as occurs in CHIP. Other refractory patients had these mutations only in the myeloid cells after TKI therapy (average treatment duration 12 months), suggesting they were selected for during TKI treatment. In our model, the self-renewal frequencies of normal and leukemic stem cells, while not required to be identical (see response to previous comments), are linked, so that the presence of a *TET2* or *ASXL1* mutation in both normal and leukemic stem cells that led to a proportional increase in self-renewal in both populations would tend to cause resistance to TKI therapy, provided that the HSC^L^ are sufficiently fit in the presence of the mutations, which we would expect. We have reworded this section in the Results and Discussion to better clarify the discussion of this issue.

6. "Although our data were derived from mice, we hypothesize that similar cell-cellsignaling occurs in humans." This hypothesis should be discussed in more detail. What are the potential differences between mice and humans that could lead to problems when applying the proposed model to patients?

While there are some documented differences between mouse and human hematopoiesis (reviewed in (*Parekh and Crooks, 2013*)), many of the proposed cell-cell signaling pathways have already been validated in both mice and humans. For example, CCL3 was shown to selectively inhibit normal HSC over CML HSC (*Eaves et al., 1993*), and this has been reproduced in a mouse CML model (*Baba et al., 2013*), while IL-6 has been shown to regulate lymphoid differentiation in both mice and humans. It remains important, however, to validate results from mice in human studies, whenever possible. We have added this information to the Discussion.

7. "only those with sufficiently large feedforward gains on the MPP division rate (γ 5 >0.01)" Please justify the cutoff of 0.01, how sensitive are the presented results with respect to this choice?

This cutoff was selected to further explore the novel feedforward loop we identified; however, the other parameter sets still yielded consistent results as they obeyed the same criteria. The corresponding sentence is updated to, “We further restricted the candidate parameter sets by considering only those with sufficiently large feedforward gains on the MPP division rate (γ_5_>0.01) in order to focus on the novel feedforward dynamics.” We added this comment in the section “Parameter estimation for feedback-feedforward model of hematopoiesis”.

8. "We used one eligible parameter set (see Supplemental Tables S4, S5), which is capable of characterizing the normal state of our simplified model of the hematopoietic system." Do other parameter sets lead to comparable results?

The answer is yes. The other parameter sets were selected based on qualitative and quantitative similarities, so for normal dynamics they are very comparable. Under the development of CML all systems exhibit the takeover by leukemic cells but differ in the timescale that the takeover occurs. This is shown in Figure 7A for one choice of leukemic stem cell parameters and in Appendix 1–figure 13 for several additional choices of leukemic stem cell parameters. The text in section “Leukemic stem cell load influences TKI therapy outcomes” has been updated and a new section Appendix 1 Sec. 10 has been added.

B) Several passages of the main text are difficult to follow since important arguments have been moved to the Methods Section at the very end or to the Supplement. Some conclusions merit a more detailed explanation. E.g.1. "Taking all these results into consideration, we arrive at the feedback-feedforward model shown in Figure 2F." This statement should be explained in more detail in the main text. Have the authors performed a systematic search of the literature to search for evidence of such mechanisms, can they name potential factors mediating them?

We thank the reviewer for this comment. In the section “Model of Normal Hematopoiesis”, we discussed IL-6 and CCL3 as candidate factors mediating negative feedback of differentiated myeloid cells on lymphoid differentiation and of myeloid progenitors on stem cell self-renewal, respectively. In the discussion, we now also mention TGF-β as a candidate factor mediating negative feedback of HSC onto their own division rate and that of the MPPs (*Naka and Hirao, 2017*), and IL-6 as a candidate factor inhibiting MPP self-renewal (*Zhao et al., 2014*).

2. "These time points and the number of mice analyzed at each time point were informed by a Bayesian hierarchical framework for optimal experimental design of mathematical models of hematopoiesis (Lomeli et al., 2021)"Please provide details.

We agree with the Reviewer that more detail would be helpful. We have modified the text in the section “Model of normal hematopoiesis” to read: These time points and the number of mice analyzed at each time point were informed by a Bayesian hierarchical framework for optimal experimental design of mathematical models of hematopoiesis. In particular, the Bayesian framework suggests combining early time points (soon after radiation was applied) with late time points because the early time points provide more information about division rates, while the late time points provide more information about the feedback parameters.

Reviewer #3 (Recommendations for the authors):I commend the authors on a well-written manuscript with important results and clear methodology.Here are some recommendations which I hope may improve the manuscript. I will preface these recommendations by noting that I have limited experience in a laboratory setting and as such do not have the expertise required to make any comments about the experimental component of this work.1. The authors could provide a brief justification for a grid search as the mechanism for identifying biologically relevant parameters.

While we could have used other approaches (*Read et al., 2018*), such as the Latin hypercube algorithm, to sample our multidimensional parameter space we chose to perform a grid search because of the ease of implementation and the fact that our goal was not to exhaustively search the full parameter space but rather obtain a set of biologically relevant parameter values consistent with normal homeostasis. This enabled us to create a cohort of virtual patients to investigate the variable dynamics of CML hematopoiesis and response to TKI treatment. We have updated the text in the Discussion to make this point.

2. In modelling CML development, there is an assumption that the only difference between normal and leukemic cell lineages is the feedback strength for leukemic HSC self-renewal. Could the validity of this assumption be explored, perhaps via supporting evidence in the literature, or through briefly considering the sensitivity of the model predictions to this assumption?

We agree with the Reviewer that this requires further investigation. Please see our responses to Reviewer 1 point (1) and (3), which address this point.

3. In Figure 1, concerning the process of identifying candidate model classes, I am not sure whether it is not present or simply hard to make out in the figure, but it appears to me that the downregulation of q_1 by TD_m marked in blue in Figure 1C is not a possible a factor in any of the four candidate model classes identified through design space analysis (Figure 1B). This is in contrast to the bottom left model of Figure 1B, where the downregulation of p_0 by p (also marked with blue in Figure 1C) is visible as a possibility.

We are sorry for the confusion. While this possibility is shown in both 1A and 1B as the dotted line from TD_m_ to q_1_ (see Figure 1A), but it is more difficult to see in Figure 1B. We have modified the figure to make this potential regulation more apparent.

References

Agathocleous M, Meacham CE, Burgess RJ, Piskounova E, Zhao Z, Crane GM, Cowin BL, Bruner E, Murphy MM, Chen W, Spangrude GJ, Hu Z, DeBerardinis RJ, Morrison SJ. 2017. Ascorbate regulates haematopoietic stem cell function and leukaemogenesis. *Nature* 549(7673):476-481. doi: 10.1038/nature23876, PMID: PMC5910063

Baba T, Naka K, Morishita S, Komatsu N, Hirao A, Mukaida N. 2013. MIP-1α/CCL3-mediated maintenance of leukemia-initiating cells in the initiation process of chronic myeloid leukemia. *J. Exp. Med.* 210(12):2661-2673. doi: 10.1084/jem.20130112, PMID: PMC3832924

Chasty RC, Lucas GS, Owen-Lynch PJ, Pierce A, Whetton AD. 1995. Macrophage inflammatory protein-1 α receptors are present on cells enriched for CD34 expression from patients with chronic myeloid leukemia. *Blood* 86(11):4270-4277. doi: PMID:

Cimmino L, Dolgalev I, Wang Y, Yoshimi A, Martin GH, Wang J, Ng V, Xia B, Witkowski MT, Mitchell-Flack M, Grillo I, Bakogianni S, Ndiaye-Lobry D, Martín MT, Guillamot M, Banh RS, Xu M, Figueroa ME, Dickins RA, Abdel-Wahab O, Park CY, Tsirigos A, Neel BG, Aifantis I. 2017. Restoration of TET2 Function Blocks Aberrant Self-Renewal and Leukemia Progression. *Cell* 170(6):1079-1095.e1020. doi: 10.1016/j.cell.2017.07.032, PMID: PMC5755977

Durig J, Testa NG, Lord BI, Kasper C, Chang J, Telford N, Dexter TM, Heyworth CM. 1999. Characterisation of the differential response of normal and CML haemopoietic progenitor cells to macrophage inflammatory protein-1alpha. *Leukemia* 13(12):2012-2022. doi: 10.1038/sj.leu.2401610, PMID:

Eaves CJ, Cashman JD, Wolpe SD, Eaves AC. 1993. Unresponsiveness of primitive chronic myeloid leukemia cells to macrophage inflammatory protein 1 α, an inhibitor of primitive normal hematopoietic cells. *Proc. Nat.l Acad. Sci. USA* 90(24):12015-12019. doi: PMID: Pmc48116

Eaves CJ, Cashman JD, Wolpe SD, Eaves AC. 1993. Unresponsiveness of primitive chronic myeloid leukemia cells to macrophage inflammatory protein 1 α, an inhibitor of primitive normal hematopoietic cells. *Proc Natl Acad Sci U S A* 90(24):12015-12019. doi: 10.1073/pnas.90.24.12015, PMID: PMC48116

Etienne G, Guilhot J, Rea D, Rigal-Huguet F, Nicolini F, Charbonnier A, Guerci-Bresler A, Legros L, Varet B, Gardembas M, Dubruille V, Tulliez M, Noel MP, Ianotto JC, Villemagne B, Carre M, Guilhot F, Rousselot P, Mahon FX. 2017. Long-Term Follow-Up of the French Stop Imatinib (STIM1) Study in Patients With Chronic Myeloid Leukemia. *J Clin Oncol* 35(3):298-305. doi: 10.1200/JCO.2016.68.2914, PMID:

Hanfstein B, Muller MC, Hehlmann R, Erben P, Lauseker M, Fabarius A, Schnittger S, Haferlach C, Gohring G, Proetel U, Kolb HJ, Krause SW, Hofmann WK, Schubert J, Einsele H, Dengler J, Hanel M, Falge C, Kanz L, Neubauer A, Kneba M, Stegelmann F, Pfreundschuh M, Waller CF, Branford S, Hughes TP, Spiekermann K, Baerlocher GM, Pfirrmann M, Hasford J, Saussele S, Hochhaus A. 2012. Early molecular and cytogenetic response is predictive for long-term progression-free and overall survival in chronic myeloid leukemia (CML). *Leukemia* 26(9):2096-2102. doi: 10.1038/leu.2012.85, PMID:

Manesso E, Teles J, Bryder D, Peterson C. 2013. Dynamical modelling of haematopoiesis: an integrated view over the system in homeostasis and under perturbation. *J. R. Soc. Interface* 10(80):20120817. doi: 10.1098/rsif.2012.0817, PMID: Pmc3565732

Marin D, Hedgley C, Clark RE, Apperley J, Foroni L, Milojkovic D, Pocock C, Goldman JM, O'Brien S. 2012. Predictive value of early molecular response in patients with chronic myeloid leukemia treated with first-line dasatinib. *Blood* 120(2):291-294. doi: 10.1182/blood-2012-01-407486, PMID:

Naka K, Hirao A. 2017. Regulation of Hematopoiesis and Hematological Disease by TGF-β Family Signaling Molecules. *Cold Spring Harb Perspect Biol* 9(9). doi: 10.1101/cshperspect.a027987, PMID: PMC5585852

Parekh C, Crooks GM. 2013. Critical differences in hematopoiesis and lymphoid development between humans and mice. *J. Clin. Immunol.* 33(4):711-715. doi: 10.1007/s10875-012-9844-3, PMID: PMC3633618

Pietarinen P, Koskenvesa P, Klievink J, Porkka K, Niemi M, Mustjoki S. 2016. CML Patients with Primary Resistance or Suboptimal Response to TKI Therapy Have Variants in Genes Affecting Drug Absorption and Metabolism. *Blood* 128(22):a3017. doi: PMID:

Read MN, Alden K, Timmis J, Andrews PS. 2018. Strategies for calibrating models of biology. *Briefings in Bioinformatics* 21(1):24-35. doi: PMID:

Reynaud D, Pietras E, Barry-Holson K, Mir A, Binnewies M, Jeanne M, Sala-Torra O, Radich JP, Passegue E. 2011. IL-6 controls leukemic multipotent progenitor cell fate and contributes to chronic myelogenous leukemia development. *Cancer Cell* 20(5):661-673. doi: 10.1016/j.ccr.2011.10.012, PMID: Pmc3220886

Thielen N, Richter J, Baldauf M, Barbany G, Fioretos T, Giles F, Gjertsen BT, Hochhaus A, Schuurhuis GJ, Sopper S, Stenke L, Thunberg S, Wolf D, Ossenkoppele G, Porkka K, Janssen J, Mustjoki S. 2016. Leukemic Stem Cell Quantification in Newly Diagnosed Patients With Chronic Myeloid Leukemia Predicts Response to Nilotinib Therapy. *Clin. Cancer Res.* 22(16):4030-4038. doi: 10.1158/1078-0432.Ccr-15-2791, PMID:

Udomsakdi C, Eaves CJ, Swolin B, Reid DS, Barnett MJ, Eaves AC. 1992. Rapid decline of chronic myeloid leukemic cells in long-term culture due to a defect at the leukemic stem cell level. *Proc Natl Acad Sci U S A* 89(13):6192-6196. doi: 10.1073/pnas.89.13.6192, PMID: PMC402148

Wang JC, Lapidot T, Cashman JD, Doedens M, Addy L, Sutherland DR, Nayar R, Laraya P, Minden M, Keating A, Eaves AC, Eaves CJ, Dick JE. 1998. High level engraftment of NOD/SCID mice by primitive normal and leukemic hematopoietic cells from patients with chronic myeloid leukemia in chronic phase. *Blood* 91(7):2406-2414. doi: PMID:

Wark G, Heyworth CM, Spooncer E, Czaplewski L, Francis JM, Dexter TM, Whetton AD. 1998. Abl protein kinase abrogates the response of multipotent haemopoietic cells to the growth inhibitor macrophage inflammatory protein-1 α. *Oncogene* 16(10):1319-1324. doi: 10.1038/sj.onc.1201914, PMID:

Zhang WW, Cortes JE, Yao H, Zhang L, Reddy NG, Jabbour E, Kantarjian HM, Jones D. 2009. Predictors of primary imatinib resistance in chronic myelogenous leukemia are distinct from those in secondary imatinib resistance. *J. Clin. Oncol.* 27(22):3642-3649. doi: 10.1200/jco.2008.19.4076, PMID: PMC2799062 are found at the end of this article.

Zhao JL, Ma C, O'Connell RM, Mehta A, DiLoreto R, Heath JR, Baltimore D. 2014. Conversion of danger signals into cytokine signals by hematopoietic stem and progenitor cells for regulation of stress-induced hematopoiesis. *Cell Stem Cell* 14(4):445-459. doi: 10.1016/j.stem.2014.01.007, PMID: PMC4119790